# Deconstruction of rheumatoid arthritis synovium defines inflammatory subtypes

Fan Zhang[1,2,3,4,5,6,39], Anna Helena Jonsson[1,38,39], Aparna Nathan[1,2,3,4,5,39], Nghia Millard[1,2,3,4,5,39], Michelle Curtis[1,2,3,4,5], Qian Xiao[1,2,3,4,5], Maria Gutierrez-Arcelus[1,2,3,4,5,7], William Apruzzese[8], Gerald F. M. Watts[1], Dana Weisenfeld[1], Saba Nayar[9,10], Javier Rangel-Moreno[11], Nida Meednu[11], Kathryne E. Marks[1], Ian Mantel[12,13], Joyce B. Kang[1,2,3,4,5], Laurie Rumker[1,2,3,4,5], Joseph Mears[1,2,3,4,5], Kamil Slowikowski[4,5,14], Kathryn Weinand[1,2,3,4,5], Dana E. Orange[12,15], Laura Geraldino-Pardilla[16], Kevin D. Deane[17], Darren Tabechian[11], Arnoldas Ceponis[18], Gary S. Firestein[18], Mark Maybury[9,19], Ilfita Sahbudin[9,19], Ami Ben-Artzi[20], Arthur M. Mandelin II[21], Alessandra Nerviani[22,23], Myles J. Lewis[22,23], Felice Rivellese[22,23], Costantino Pitzalis[22,23,24], Laura B. Hughes[25], Diane Horowitz[26], Edward DiCarlo[27], Ellen M. Gravallese[1], Brendan F. Boyce[28], Accelerating Medicines Partnership: RA/SLE Network*, Larry W. Moreland[17,29], Susan M. Goodman[12,13], Harris Perlman[21], V. Michael Holers[17], Katherine P. Liao[1,4], Andrew Filer[9,10,19], Vivian P. Bykerk[12,13], Kevin Wei[1,40], Deepak A. Rao[1,40], Laura T. Donlin[12,13,40], Jennifer H. Anolik[11,40], Michael B. Brenner[1,40] & Soumya Raychaudhuri[1,2,3,4,5,40] ✉

Rheumatoid arthritis is a prototypical autoimmune disease that causes joint inflammation and destruction[1]. There is currently no cure for rheumatoid arthritis, and the effectiveness of treatments varies across patients, suggesting an undefined pathogenic diversity[1,2]. Here, to deconstruct the cell states and pathways that characterize this pathogenic heterogeneity, we profiled the full spectrum of cells in inflamed synovium from patients with rheumatoid arthritis. We used multi-modal single-cell RNA-sequencing and surface protein data coupled with histology of synovial tissue from 79 donors to build single-cell atlas of rheumatoid arthritis synovial tissue that includes more than 314,000 cells. We stratified tissues into six groups, referred to as cell-type abundance phenotypes (CTAPs), each characterized by selectively enriched cell states. These CTAPs demonstrate the diversity of synovial inflammation in rheumatoid arthritis, ranging from samples enriched for T and B cells to those largely lacking lymphocytes. Disease-relevant cell states, cytokines, risk genes, histology and serology metrics are associated with particular CTAPs. CTAPs are dynamic and can predict treatment response, highlighting the clinical utility of classifying rheumatoid arthritis synovial phenotypes. This comprehensive atlas and molecular, tissue-based stratification of rheumatoid arthritis synovial tissue reveal new insights into rheumatoid arthritis pathology and heterogeneity that could inform novel targeted treatments.

Rheumatoid arthritis is a systemic autoimmune disease that affects up to 1% of the population[3]. It is characterized by inflammation of synovial joint tissue and extra-articular manifestations that lead to pain, joint damage and disability[1]. The clinical course of rheumatoid arthritis has been transformed by targeted therapies, including those aimed at TNF, IL-6, B cells, T cell co-stimulation and the JAK–STAT pathway[1]. However, many patients are refractory to these therapies and do not achieve remission[2]. Thus, there is a clinical need for new treatment targets and for predictors of patient-specific responses to treatment. Genetic diversity and variable responses to targeted therapies suggest that rheumatoid arthritis is a heterogeneous disease[4]. However, genetic and clinical differences in disease duration or activity do not reliably predict the treatment response or druggable targets[1,5].

A more granular understanding of cell states and synovial phenotypes in inflamed joints could inform prognosis and therapeutic targets. Encouragingly, clinical trials using histologic or bulk RNA-sequencing (RNA-seq) analysis of synovial tissue suggest that treatment response may depend on synovial cellular composition[6,7]. Previous studies have identified effector cell states in rheumatoid arthritis pathophysiology that represent promising treatment targets, including *HBEGF*+*IL1B*+ macrophages, SLAMF7+ super-activated macrophages, *MERTK*+ macrophages, CD11c+ autoimmune-associated B cells (ABCs), PD-1hi T peripheral helper (T_PH) cells, granzyme K+CD8+ T cells and *NOTCH3*+ synovial fibroblasts[8–16]. To determine whether some states are enriched only in specific subsets of patients, we analysed cell-state composition in a clinically diverse set of patients with active rheumatoid arthritis.

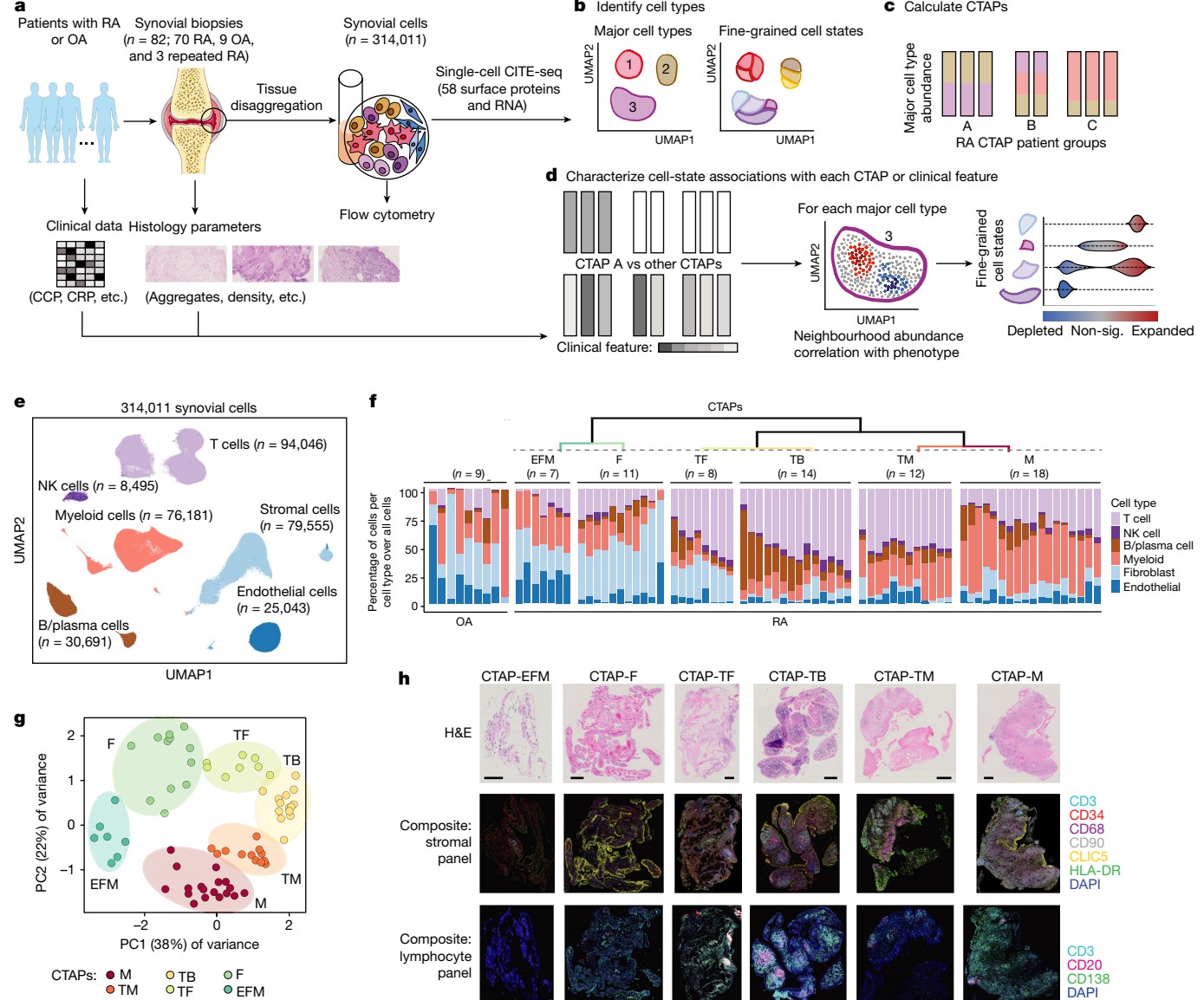

**Fig. 1 | Overview of the multi-modal single-cell synovial tissue pipeline and cell-type abundance analysis that reveals distinct rheumatoid arthritis CTAPs. a–d**, Description (**a**) of the patient recruitment, clinical and histologic metrics, synovial sample processing pipeline and computational analysis strategy, including identification of major cell types and fine-grained cell states (**b**), definition of distinct rheumatoid arthritis CTAPs (**c**), and cell neighbourhood associations with each CTAP or with clinical or histologic parameters for each major cell type (**d**). OA, osteoarthritis; RA, rheumatoid arthritis; sig., significant. **e**, Integrative uniform manifold approximation and projection (UMAP) based on mRNA and protein discriminated major cell types. **f**, Hierarchical clustering of cell-type abundances captures six rheumatoid arthritis subgroups, referred to as CTAPs. The nine osteoarthritis samples are shown as a comparison. Each bar represents one synovial sample, coloured by the proportion of each major cell type. **g**, PCA of major cell-type abundances. Each dot represents a sample, plotted based on its PC1 and PC2 projections and coloured by CTAPs. **h**, Representative synovial tissue fragments from each of the CTAPs. Top row, haematoxylin and eosin (H&E) staining. Middle row, immunofluorescence microscopy for CD3, CD34, CD68, CD90, CLIC5 and HLA-DR. Bottom row, immunofluorescence microscopy for CD3, CD20 and CD138. Scale bars: 100 μm (CTAP-EFM) and 250 μm (all other images). Single-colour images are presented in Supplementary Fig. 4. A total of 150 fragments from 36 donors were stained in batches and analysed as a single cohort. Parts of Fig. 1a were generated using Servier Medical Art, provided by Servier, licensed under a Creative Commons Attribution 3.0 unported license.

As rheumatoid arthritis shares disease-associated tissue cell states and genetic risk loci with other autoimmune diseases[17,18], these analyses may offer insights into other diseases that feature tissue inflammation.

## Recruitment and multi-modal analysis of tissue

We obtained a total of 82 synovial tissue samples from patients exhibiting moderate to high disease activity (clinical disease activity index (CDAI) ≥ 10). To capture a clinical spectrum of rheumatoid arthritis, we collected biopsies from treatment-naive patients (n = 28) early in their disease course, methotrexate (MTX)-inadequate responders (n = 27), and anti-TNF agent-inadequate responders (n = 15) as well as from patients with osteoarthritis (n = 9) (Fig. 1a–d, Supplementary Table 1).

We simultaneously characterized the transcriptome and surface expression of 58 proteins (Supplementary Table 2) in a total of 314,011 cells (more than 3,800 cells per sample) after quality control (Supplementary Fig. 1). We integrated surface marker and RNA data using canonical correlation analysis, corrected batch effects and defined six major cell types: T, B and plasma (B/plasma), natural killer (NK),

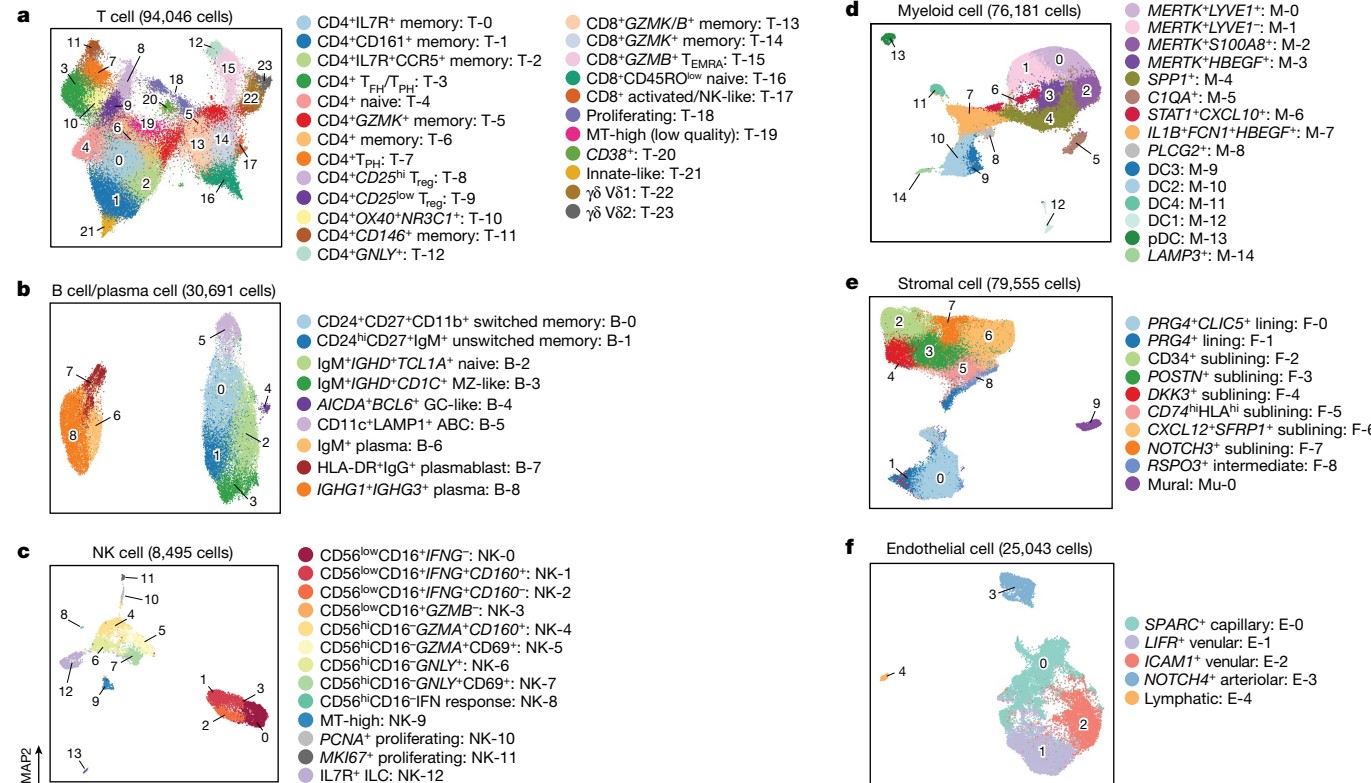

**Fig. 2 | Cell-type-specific single-cell analysis captures 77 distinct cell states in rheumatoid arthritis synovium. a–f,** Cell-type-specific reference UMAPs for T cells (**a**) B/plasma cells (**b**), NK cells (**c**), myeloid cells (**d**), stromal cells (**e**) and endothelial cells (**f**), coloured by fine-grained cell-state clusters. MT, mitochondrial; MZ, marginal zone; pDC, plasmacytoid dendritic cell.

## Stratifying synovium by cell-type abundance

To define potentially distinct tissue inflammatory phenotypes, we hierarchically clustered synovial samples on the basis of the frequency of the six major cell lineages (Fig. 1f,g). On the basis of in-group similarity with bootstrapping, we arrived at six different categories that we call CTAPs, which are largely robust to adjustment for treatment and disease duration (Extended Data Fig. 1b–e). We named the CTAPs on the basis of relatively enriched cell type(s): (1) endothelial, fibroblast and myeloid cells (EFM); (2) fibroblasts (F); (3) T cells and fibroblasts (TF); (4) T and B cells (TB); (5) T and myeloid cells (TM); and (6) myeloid cells (M) (Extended Data Fig. 1d and Supplementary Table 4). Alternative clustering schemes using highly variable genes, all transcriptional states, or separating plasma cells from non-plasma B cells led to similar results (Supplementary Fig. 3). Post hoc mapping of the osteoarthritis samples demonstrates that they most resemble CTAP-EFM and CTAP-F (Extended Data Fig. 1f). Categorization by effector functions using pseudo-bulk expression of 55 cytokines, chemokines and growth factors was similar to the cell lineage-based CTAP categorization (Extended Data Fig. 1g,h).

## CTAP patterns are consistent across fragments

To examine the robustness of CTAPs across paired biopsy fragments from the same joint, we performed immunofluorescence microscopy staining on synovial tissue fragments from a subset of patients ($n = 36$) (Fig. 1h and Supplementary Fig. 4). We compared cell-type proportions in individual high-density biopsy fragments with the disaggregated

cellular indexing of transcriptomes and epitopes (CITE-seq)-based cell frequencies (Extended Data Fig. 1i,j). The proportions of cell types followed the patterns predicted by the CITE-seq-based CTAP assignment. For example, CD20+ (that is, non-plasma) B cells were most frequent in CTAP-TB, whereas CD68+ myeloid cells were most frequent in CTAP-M and CTAP-TM. As the histology analysis was performed on synovial tissue fragments separate from those used for CITE-seq, these findings support the consistency of CTAP assignments across a joint.

## A rheumatoid arthritis synovial cell-state atlas

We defined finer-grained cell states and quantified cluster abundances within cell types (Fig. 2 and Extended Data Fig. 2) using canonical variates from canonical correlation analysis reflecting both RNA and protein for T and B cells and mRNA principal components for myeloid, stromal and endothelial cell states (Supplementary Figs. 5 and 6 and Supplementary Table 3). In total we defined 77 cell states: 24 T cell clusters ($n = 94,046$ cells), 9 B/plasma cell clusters ($n = 30,691$), 14 NK clusters ($n = 8,495$), 15 myeloid clusters ($n = 76,181$), 5 endothelial clusters ($n = 25,043$) and 10 stromal clusters ($n = 79,555$) (Fig. 2 and Supplementary Table 5). Cell states associated with rheumatoid arthritis versus osteoarthritis in a previous study of more than 5,000 synovial cells were also associated with rheumatoid arthritis in this dataset (Supplementary Fig. 7 and Supplementary Table 6).

The 24 T cell clusters spanned innate-like states and CD4+ and CD8+ adaptive lineages, including states implicated in autoimmunity, such as regulatory CD4+ T cells ($T_{reg}$) (T-8 and T-9) and *CXCL13*- and *IL21*-expressing T follicular helper ($T_{FH}$) and $T_{PH}$ cells[17,19] (T-3 and T-7) (Fig. 2a and Extended Data Figs. 2 and 3). T-7 exclusively comprised $T_{PH}$ cells and expressed more *ICOS*, *IFNG* and *GZMA*, whereas T-3 contained $T_{FH}$ and $T_{PH}$ ($T_{FH}/T_{PH}$) cells expressing the lymphoid homing marker gene

*CCR7*. CD8[+] subsets expressed different combinations of *GZMB* and *GZMK*, reflecting differential cytotoxic potential. Using cell surface protein data, we resolved T cell clusters that were not observed in our earlier study[8], including CD4[+]*GNLY*[+] (T-12), double-negative (CD4[−]CD8[−]) γδ T cells expressing *TRDC* (T-22 and T-23) and double-negative and CD8[+] T cells expressing *ZBTB16* (which encodes PLZF) that resemble NK T cells and mucosal-associated innate T (MAIT) cells (T-21).

CD20 (encoded by *MS4A1*)-expressing B cells comprised six clusters, including IgM[+]*IGHD*[+]*TCL1A*[+] naive (B-2), CD24[hi]CD27[+]IgM[+] unswitched memory (B-1) and CD24[+]CD27[+]CD11b[+] (CD11b is also known as ITGAM) switched memory (B-0) B cells (Fig. 2b and Extended Data Figs. 2 and 4). CD11c[+]*CXCR5*[low] (CD11c is also known as *ITGAX*) ABCs (B-5) expressed LAMP1, HLA-DR and *CIITA*, indicating B cell antigen presentation[20–22]. Unexpectedly, we observed *CD1C*[+] B cells (B-3) with CD27 and *IGHD* expression, consistent with recirculating extrasplenic marginal zone B cells. These and other non-plasma B cells expressed *IL6* and *TNF* (Extended Data Fig. 4d). We identified *AICDA*[+]*BCL6*[+] germinal centre-like B cells (B-4), consistent with ectopic germinal centre formation in synovium[23]. Plasma cell populations included HLA-DR[+]IgG[+] plasmablasts (B-7) expressing *MKI67*, IgM[+] plasma cells (B-6) and mature *IGHG1*[+]*IGHG3*[+] plasma cells (B-8), possibly reflecting both in situ generation and recruitment from the circulation.

We also captured innate lymphocytes, including CD56[hi]CD16[−] NK (eight clusters), CD56[low]CD16[+] NK (four clusters) and CD56[low]CD16[−]IL7R[+] innate lymphoid cells (ILCs) (two clusters) (Fig. 2c and Extended Data Figs. 2 and 5). CD56[hi]CD16[−] NK cells were more abundant (mean 48% per donor) than CD56[low]CD16[+] NK cells (36%) and ILCs (13%). CD56[hi]CD16[−] NK clusters expressed *GZMK*, with variable expression of cytotoxicity genes such as *GZMB* and *GNLY*. CD56[low]CD16[+] NK cells exhibited universally high expression of *GZMB*, *GNLY* and *PRF1*. Several NK cell clusters highly expressed *IFNG* (Extended Data Fig. 5d). ILCs, identified by the absence of CD56 and CD16 with high CD127 (also known as IL-7Rα) protein, included group 3 ILCs (*RORC*[+] NK-12) and group 2 ILCs[24] (CD161[+]*GATA3*[+] NK-13).

We identified 15 myeloid clusters (Fig. 2d). CD68 and CCR2 discriminated tissue macrophages from infiltrating monocytes (Extended Data Figs. 2 and 6). Three tissue macrophage clusters (M-0, M-1 and M-2) were abundant in both osteoarthritis and rheumatoid arthritis synovium and expressed the phagocytic factors CD206 (also known as macrophage mannose receptor (MMR)) and CD163 and *MERTK* (Extended Data Fig. 6b–d), suggesting a homeostatic debris-clearing function[25,26]. *LYVE1* expression (M-0) is likely to indicate a perivascular function[12,27]. Infiltrating monocytes included a previously described *IL1B*[+]*FCN1*[+]*HBEGF*[+] pro-inflammatory subset (M-7), probably derived from classical CD14[hi] monocytes[8,12] and a *STAT1*[+]*CXCL10*[+] subset (M-6) that expresses interferon-response genes. *MERTK*[+]*HBEGF*[+] (M-3) and *SPP1*[+] (M-4) subsets expressed *SPP1* (osteopontin) and other factors consistent with wound-healing responses[28,29]. Four dendritic cell (DC) populations corresponded to subsets described by Villani et al.[30]. *CLEC10A*[hi] DC2 and DC3 (M-9 and M-10) and *CLEC9A*[+]*THBD*[+] DC1 (M-12) are likely to activate CD4[+] and CD8[+] T cells, respectively, whereas DC4 (M-11) expressed CD16[+] monocyte factors and an interferon signature (Extended Data Fig. 6d). A fifth DC subset (M-14) highly expressed the endosomal marker *LAMP3*[31].

Fibroblasts segregated broadly into lining (*PRG4*[hi]) and sublining (*THY1*[+]*PRG4*[low]) subsets and *NOTCH3*[+]*MCAM*[+] (CD146) mural cells (Fig. 2e and Extended Data Figs. 2 and 7a–f). As previously described, lining fibroblasts (F-0 and F-1) were depleted in rheumatoid arthritis relative to osteoarthritis and subdivided into *PRG4*[+]*CLIC5*[+] (F-0), *PRG4*[+] (F-1) and *RSPO3*[+] (F-8) populations, the last exhibiting an intermediate lining–sublining phenotype. Sublining fibroblasts separated into *HLA-DRA*[+], CD34[+] and *DKK3*[+] groups[8,32,33]. The CD34[+] sublining fibroblast cluster (F-2) highly expressed *PI16* and *DPP4* (CD26), suggesting an undifferentiated, progenitor-like state[34]. *CXCL12*[+] fibroblasts included an inflammatory *CD74*[hi]*HLA*[hi] cluster (F-5) and a *CXCL12*[+]*SFRP1*[+] cluster

(F-6) with the highest levels of *IL6*, which encodes a proven drug target in rheumatoid arthritis.

Synovial endothelial cells separated into lymphatic endothelial cells and blood endothelial cells. Lymphatic endothelial cells (E-4), identified on the basis of high expression of the lymphatic markers *LYVE1* and *PROX1*, exhibited high expression of *CCL21* and *FLT4*[35,36] (Fig. 2f and Extended Data Figs. 2 and 7g,k). Among blood endothelial cells, we observed several clusters along an arterial-to-venous axis, including *NOTCH4*[+] arteriolar (E-3), *SPARC*[+] capillary (E-0) and *CLU*[+] venular (E-1 and E-2) cells. Arteriolar cells expressed high levels of *CXCL12*, *LTBP4*, *NOTCH4* and the NOTCH ligand *DLL4*. *SPARC*[+] capillary cells expressed collagen and extracellular matrix genes. Venular cells further subdivided into *LIFR*[+] (E-1) and *ICAM1*[+] (E-2) and had high expression of inflammatory genes such as *IL6* and HLA genes, along with genes that facilitate leukocyte transmigration, such as *ICAM1* and *SELE* (E-selectin) (Extended Data Fig. 7i).

## CTAPs are defined by specific cell states

We used co-varying neighbourhood analysis (CNA) to identify single-cell-resolution 'neighbourhoods' associated with individual CTAPs. We use 'expanded' and 'depleted' to refer to differences in relative abundance within a cell type, accounting for age, sex and cell count per sample. Of note, this may not reflect a difference relative to total synovial cells. We tested each cell type for associations with all CTAPs, recognizing that even less enriched cell types may contain critical subsets.

We observed skewed T and B cell neighbourhoods in CTAP-TB (permutation $P = 0.046$ and $0.03$, respectively) (Fig. 3a, Extended Data Fig. 3e, Supplementary Tables 7 and 8). T cell neighbourhoods among CD4[+] T$_{FH}$/T$_{PH}$ (T-3) and CD4[+] T$_{PH}$ (T-7) cells were expanded, whereas neighbourhoods among cytotoxic CD4[+]*GNLY*[+] (T-12) and CD8[+]*GZMB*[+] cells (T-15) were depleted. Among B cells, we observed expanded neighbourhoods in memory B (B-0 and B-1) and ABC (B-5) clusters, whereas IgG1[+]IgG3[+] and IgM[+] plasma cells (B-8 and B-6) were relatively depleted (Fig. 3b and Extended Data Fig. 4e). We note that although plasma cells are depleted among B/plasma cells in CTAP-TB, plasma cells are enriched among total cells in CTAP-TB (4.1% compared with 0.6–3.1% in other CTAPs) (Extended Data Fig. 4e,f). Although T$_{PH}$ (T-7), T$_{FH}$/T$_{PH}$ (T-3) and ABC (B-5) cells are enriched in CTAP-TB, they are present in all six CTAPs (Extended Data Figs. 3e and 4e). By contrast, germinal centre cells (B-4) were almost exclusively found in CTAP-TB (Extended Data Fig. 4e). Consistent with a role for T$_{FH}$/T$_{PH}$ and IL-21 in ABC generation[37], the frequency of ABCs (B-5) amongst B/plasma cells correlated with the proportion of T$_{PH}$ (T-7) and T$_{FH}$/T$_{PH}$ (T-3) among T cells (Pearson $r = 0.50$, $P = 3.7 \times 10^{-6}$ and Pearson $r = 0.24$, $P = 0.034$, respectively) (Fig. 3c and Extended Data Fig. 4g).

We hypothesized that the preferential enrichment of T$_{PH}$ and T$_{FH}$ cells in CTAP-TB reflected the ability of these subsets to sustain and activate B cells. To test this hypothesis, we sorted T$_{PH}$ and T$_{FH}$ cells and other memory CD4[+] T cells, as well as CD45RA[+] effector memory CD8[+] T (T$_{EMRA}$) cells and CD45RO[+] memory CD8[+] T cells, which are enriched for GZMB[+] and GZMK[+] CD8[+] T cells, respectively[16] from blood and co-cultured them with B cells and staphylococcal enterotoxin B superantigen (Fig. 3d, Extended Data Fig. 4h and Supplementary Fig. 8). T$_{PH}$ and T$_{FH}$ cells efficiently induced B cell differentiation into plasmablast and ABC phenotypes. Notably, non-T$_{FH}$/T$_{PH}$ memory CD4[+] T cells were also able to induce ABC differentiation, but not plasmablast differentiation. CD8[+] T cells did not induce B cell differentiation despite being functionally potent in cytotoxicity assays.

T cell neighbourhoods enriched in CTAP-TF (permutation $P = 0.036$) consisted mainly of cytotoxic CD4[+]*GNLY*[+] (T-12) and CD8[+]*GZMB*[+] cells (T-15) as well as naive CD4[+] and CD8[+] T cells (T-4 and T-16) (Fig. 3a, Extended Data Fig. 3e and Supplementary Tables 7 and 8). *GZMB*-expressing CD56[low]CD16[+] NK cells (NK-0–3) were also enriched in

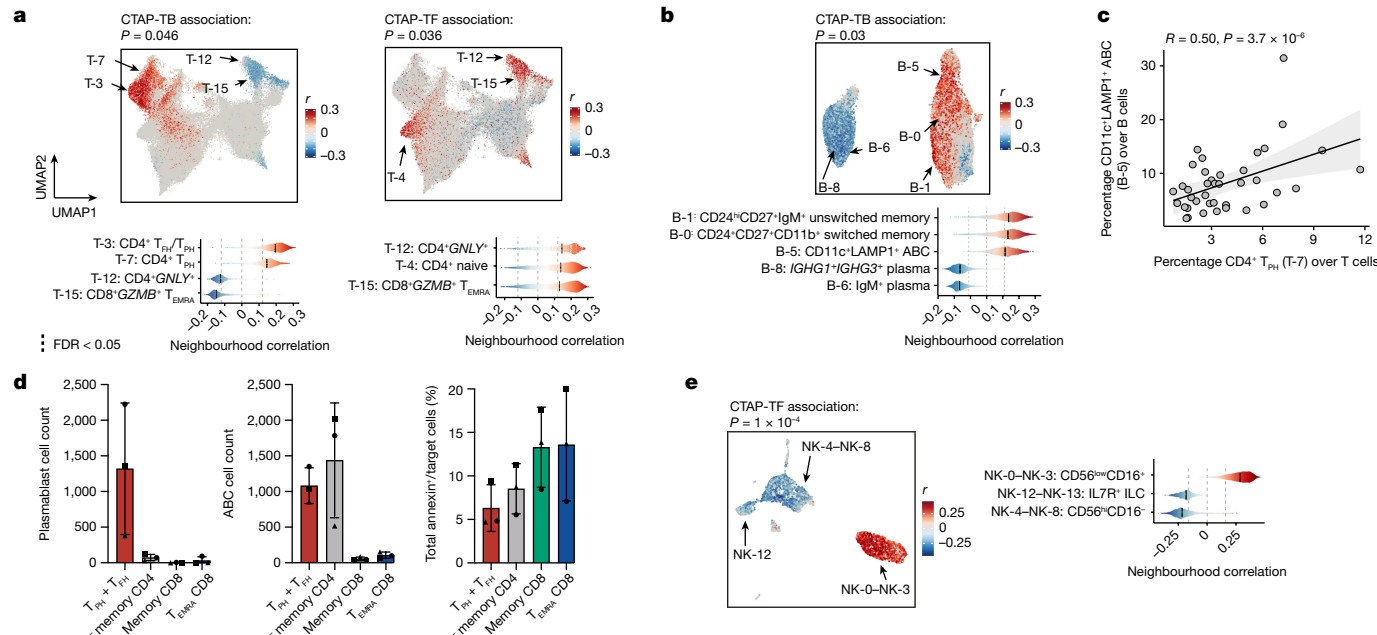

**Fig. 3 | Different T cell, B cell and NK cell populations are associated with rheumatoid arthritis CTAPs. a**, Associations of T cell neighbourhoods with CTAP-TB and CTAP-TF. *P* values are from the CNA test for each CTAP within T cells. **b**, Associations of B/plasma cell neighbourhoods with CTAP-TB. **c**, Percentage of T$_{PH}$ (T-7) as a proportion of T cells and CD11c⁺ LAMP1⁺ ABCs (B-5) as a proportion of B/plasma cells for each donor sample. *R* and *P* values are calculated from Pearson correlation and two-sided *t*-tests, respectively. The shaded region represents 95% confidence interval. **d**, Plasmablast count (left), ABC count (centre) or percentage of annexin⁺ cells (right) stratified by

co-cultured T cell subset. Points represent samples and shapes correspond to samples from the same donor, which were tested in independent experiments (*n* = 3). Data are mean ± s.d. **e**, Associations of NK cell neighbourhoods with CTAP-TF. **a**,**b**,**d**, For all CNA results, cells in UMAPs are coloured red (positive) or blue (negative) if their neighbourhood is significantly associated with the CTAP (false discovery rate (FDR) < 0.05), and grey otherwise. Distributions of neighbourhood correlations are shown for clusters with more than 50% of neighbourhoods correlated with the CTAP at FDR < 0.05. Global *P* values were obtained based on permutation testing from the CNA package.

CTAP-TF, and the proportion of *GZMB*⁺ NK cells (NK-0–3) correlated with the proportion of *GZMB*⁺ T cells (T-15) (Pearson *r* = 0.63, *P* = 4.87 × 10⁻¹⁰; Fig. 3e and Extended Data Fig. 5g). Conversely, *GZMK*⁺ CD8⁺ T cells (T-13 and T-14) correlated with *GZMK*⁺ NK cells (NK-4–8, Pearson *r* = 0.51, *P* = 1.41 × 10⁻⁶), suggesting that GZMB- and *GZMK*-expressing CD8⁺ T and NK cells share a transcriptional programme influenced by their tissue environments.

CTAP-TF also exhibited specific expansion among *CXCL12*⁺*SFRP1*⁺ sublining fibroblasts (F-6), which expressed *IL6* but not HLA-DR genes (Fig. 4a and Extended Data Fig. 7c). By contrast, CTAP-M demonstrated enrichment of *CD74*ʰⁱ*HLA*ʰⁱ sublining fibroblast neighbourhoods (F-5) among stromal cells (permutation *P* = 10⁻³). We also observed that *SPARC*⁺ capillary cells (E-0) were expanded among endothelial cells in CTAP-M (permutation *P* = 7 × 10⁻³; Extended Data Fig. 7l).

Among myeloid populations, cell neighbourhoods within *SPP1*⁺ (M-4) and *MERTK*⁺*HBEGF*⁺ (M-3) macrophages were enriched in CTAP-M, suggesting recruitment of inflammatory monocytes and transition to macrophage function (Fig. 4b). Pro-inflammatory *IL1B*⁺ macrophages (M-7), known to be expanded in patients with rheumatoid arthritis in general[8], were less frequent in CTAP-EFM relative to other CTAPs.

Of note, CTAP-M and CTAP-F exhibited contrasting cell enrichments and depletions across three cell types. (Fig. 4a,b and Extended Data Fig. 7l). Specifically, lining (F-0 and F-1) and CD34⁺ sublining (F-2) fibroblasts (permutation *P* = 3 × 10⁻³), *MERTK*⁺*LYVE1*⁺ (M-0) and *MERTK*⁺*S100A8*⁺ (M-2) macrophages (permutation *P* = 10⁻³), and *LIFR*⁺ venular (E-1) and *ICAM1*⁺ venular (E-2) endothelial cells were expanded in CTAP-F (permutation *P* = 3 × 10⁻³) and depleted in CTAP-M.

Given their high plasticity, we hypothesized that monocytes entering synovial tissue are shaped by the network of cell types and soluble factors associated with each CTAP. We tested this concept for CTAP-M

and CTAP-TM by exposing human blood CD14⁺ monocytes to factors enriched in these tissues and then examining which CTAP-associated myeloid state these cells resembled (Extended Data Fig. 6g). We found that activated CD8⁺ T cell factors that mark CTAP-TM induced a set of genes that mark the *STAT1*⁺*CXCL10*⁺ macrophage state that is enriched in CTAP-TM (Extended Data Fig. 6h,i). Conversely, factors enriched in CTAP-M, including M-CSF, TGFβ and fibroblasts, drove monocytes towards the *MERTK*⁺*HBEGF*⁺ phenotype that is enriched in CTAP-M.

## Cell states are associated with histology

We used CNA to test for cell neighbourhoods associated with histologic features of rheumatoid arthritis synovium, including Krenn scores and discrete histologic cell density and aggregate scores reflecting inflammatory cell infiltration and organization (Fig. 5a, Supplementary Fig. 9a and Methods). Several T cell states were associated with aggregate scores (permutation *P* = 0.0088), including neighbourhoods among CD4⁺ T$_{FH}$/T$_{PH}$ (T-3), *GZMK*⁺CD8⁺ T cells, and some memory CD4⁺ T cells (Fig. 5a, Supplementary Fig. 9b and Supplementary Table 7). A *GZMK*⁺ NK cell cluster, NK-4, was associated with both density and aggregate scores (permutation *P* = 3 × 10⁻⁴ and 10⁻⁴, respectively) (Supplementary Fig. 9b). Neighbourhoods within *STAT1*⁺*CXCL10*⁺ (M-6), *SPP1*⁺ (M-4) and inflammatory DC3 (M-9) (Fig. 5a and Supplementary Fig. 9b) were associated with both aggregate and density scores (permutation *P* = 0.006 and *P* = 0.005, respectively). Among B cells, IgM⁺ plasma cells (B-6), plasmablasts (B-7) and ABCs (B-5) were associated with aggregate scores (permutation *P* = 0.007) (Fig. 5a and Supplementary Fig. 9b). These disparate cell-state associations with aggregate scores probably reflect the diverse composition of aggregates, which can be T cell-dominant, plasma cell-dominant or T and B cell follicles[38,39].

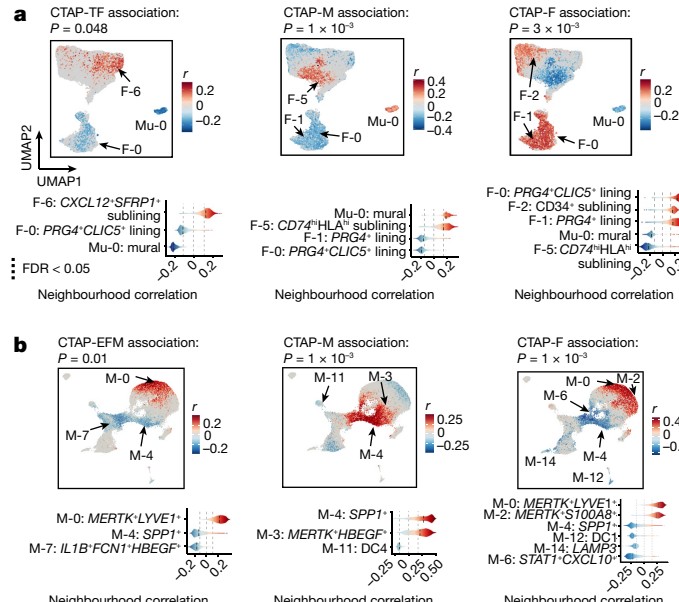

**Fig. 4 | Different stromal, myeloid and endothelial cell populations are associated with rheumatoid arthritis CTAPs. a**, Association of stromal cell neighbourhoods with CTAP-TF, CTAP-M and CTAP-F. **b**, Association of myeloid cell neighbourhoods with CTAP-EFM, CTAP-M and CTAP-F for all CNA results. Cells in UMAPs are coloured red (positive) or blue (negative) if their neighbourhood is significantly associated with the CTAP (FDR < 0.05), and grey otherwise. Distributions of neighbourhood correlations are shown for clusters with more than 50% of neighbourhoods correlated with the CTAP at FDR < 0.05. Global *P* values were obtained based on the permutation testing from the CNA package.

After accounting for age, sex, cell count and clinical collection site (Methods), we found that CTAPs account for 18% of variance of histologic density (*P* = 0.0035) and 18% of variance for aggregates (*P* = 0.0059), with CTAP-TB and CTAP-TF having the highest scores for both (Extended Data Fig. 8a,b). Consistent with these observations, CTAPs are associated with Krenn inflammation scores (*P* = 4 × 10⁻⁴), but not with Krenn lining scores (*P* = 0.11) (Extended Data Fig. 8a,b). Ultrasound measurements in the biopsied joint did not vary by CTAP (Extended Data Fig. 8b). In our dataset, we observed no association between Krenn inflammation and power doppler scores, consistent with some previous studies[40–42] (Extended Data Fig. 8c).

## CTAPs are largely independent of clinical metrics

Cyclic citrullinated peptide (CCP) autoantibodies are known to confer a higher risk of severe disease and radiographic progression[43]. CCP titre values differed across CTAPs (*P* = 0.023, 18% variance), with CTAP-M having the lowest CCP titres, even after restricting the analysis to seropositive patients (*P* = 0.0047) (Extended Data Fig. 8a,d). *HLA-DRB1* is the strongest genetic rheumatoid arthritis risk factor for seropositive disease, yet we did not find that *HLA-DRB1* risk alleles were associated with a particular CTAP, although there was a trend toward association with CTAP-TB (Extended Data Fig. 8e and Methods).

We did not find a significant association between CTAPs and disease activity score-28 for rheumatoid arthritis with C-reactive protein (DAS28-CRP) or CDAI (Extended Data Fig. 8b), although our patient cohort is not ideal for testing such associations because it only includes patients with high disease activity. CTAPs were also independent of other clinical factors, smoking history and sex, and mostly independent of anatomic category and clinical site (Extended Data Fig. 8b,f–l and Supplementary Table 9). Patients with CTAP-EFM had statistically

nonsignificant trends to be older, have longer-standing rheumatoid arthritis and be inadequate responders to TNF inhibitors (Extended Data Fig. 8m–p).

## CTAPs have disease-relevant cytokine profiles

We next analysed transcript levels of cytokines, chemokines, and their receptors, recognizing that these transcripts are often sparse in single-cell RNA-seq data (Supplementary Fig. 10). Most cytokines and chemokines are detected predominantly in one cell type, although some key cytokines were produced by multiple cell types (Extended Data Fig. 9a,b). For example, we detected *TNF* in roughly equal numbers of T cells and myeloid cells, whereas fibroblasts, endothelial cells and B cells dominated among cells with detectable *IL6*.

Next, we correlated CTAP neighbourhood association scores with the expression of key cytokines and receptors to identify soluble factors produced by CTAP-associated cell states. For example, as predicted, CTAP-TB, enriched for $T_{FH}/T_{PH}$ cell states, had T cell neighbourhood association scores that correlated with expression of the $T_{FH}/T_{PH}$ marker *CXCL13* (Fig. 3a and Extended Data Fig. 9c). By contrast, CTAP-TF-associated *GZMB*⁺ T and NK cell neighbourhoods had association scores correlating with the expression of *IFNG* and *TNF* (Fig. 3a,e and Extended Data Fig. 9c), suggesting that these cytokines may be key molecular drivers of CTAP-TF.

In some CTAPs, this analysis revealed potential cytokine networks. For example, in CTAP-M, myeloid neighbourhood association scores correlated with expression of angiogenic factor *VEGFA*, whereas endothelial cell neighbourhood association scores correlated with expression of *KDR* (also known as *VEGFR2*), potentially explaining the observed enrichment of capillaries in this CTAP (Extended Data Figs. 7l and 9c). By contrast, in CTAP-F, enriched *LIFR*⁺ and *ICAM1*⁺ venular endothelial cell neighbourhoods expressed high levels of *CCL14*, whose cognate receptor *CCR1* was highly expressed by *MERTK*⁺ macrophage neighbourhoods, which are also enriched in CTAP-F (Fig. 4b and Extended Data Fig. 7l and Fig. 9c). Cell–cell communication analysis confirmed these putative interactions (Supplementary Fig. 11).

Our study included three patients with replicate biopsies obtained from the same joint 98 to 190 days after the initial biopsy. Cell-type composition of repeat biopsies was similar to the initial biopsy (permutation *P* = 0.073) (Supplementary Fig. 12a,b), but more samples are needed to understand how dynamic CTAPs are.

## Mapping CTAPs to other patient cohort data

To enable investigation of these and other CTAP-related questions in larger studies, we examined whether samples can be classified into CTAPs using lower-resolution technologies such as flow cytometry and bulk tissue RNA-seq. We first built a nearest-neighbour classifier for flow cytometry data and were able to accurately replicate CITE-seq-based CTAP assignments (accuracy = 87%; Extended Data Fig. 9d, Supplementary Fig. 12c,d and Supplementary Table 10).

We next developed a method to classify CTAPs using bulk RNA-seq data of intact synovial tissue from a recent clinical trial[6]. CTAP classification based on bulk RNA-seq agreed with the CITE-seq-based CTAP assignment for 6 out of 7 samples in the present study that were also analysed with bulk RNA-seq (Extended Data Fig. 10a).

We applied our CTAP classification algorithm to bulk RNA-seq profiles from the R4RA clinical trial comparing rituximab and tocilizumab for the treatment of patients with rheumatoid arthritis with inadequate response to TNF inhibitor therapy[44] (*n* = 133). The distribution of CTAPs differs between these datasets, probably reflecting differences in cohort recruitment criteria (Extended Data Fig. 10b). As in our cohort, we found no association between CTAP assignment and disease activity or between treatment response and disease activity (Extended Data Fig. 10c,d), supporting our hypothesis that CTAPs reflect distinct

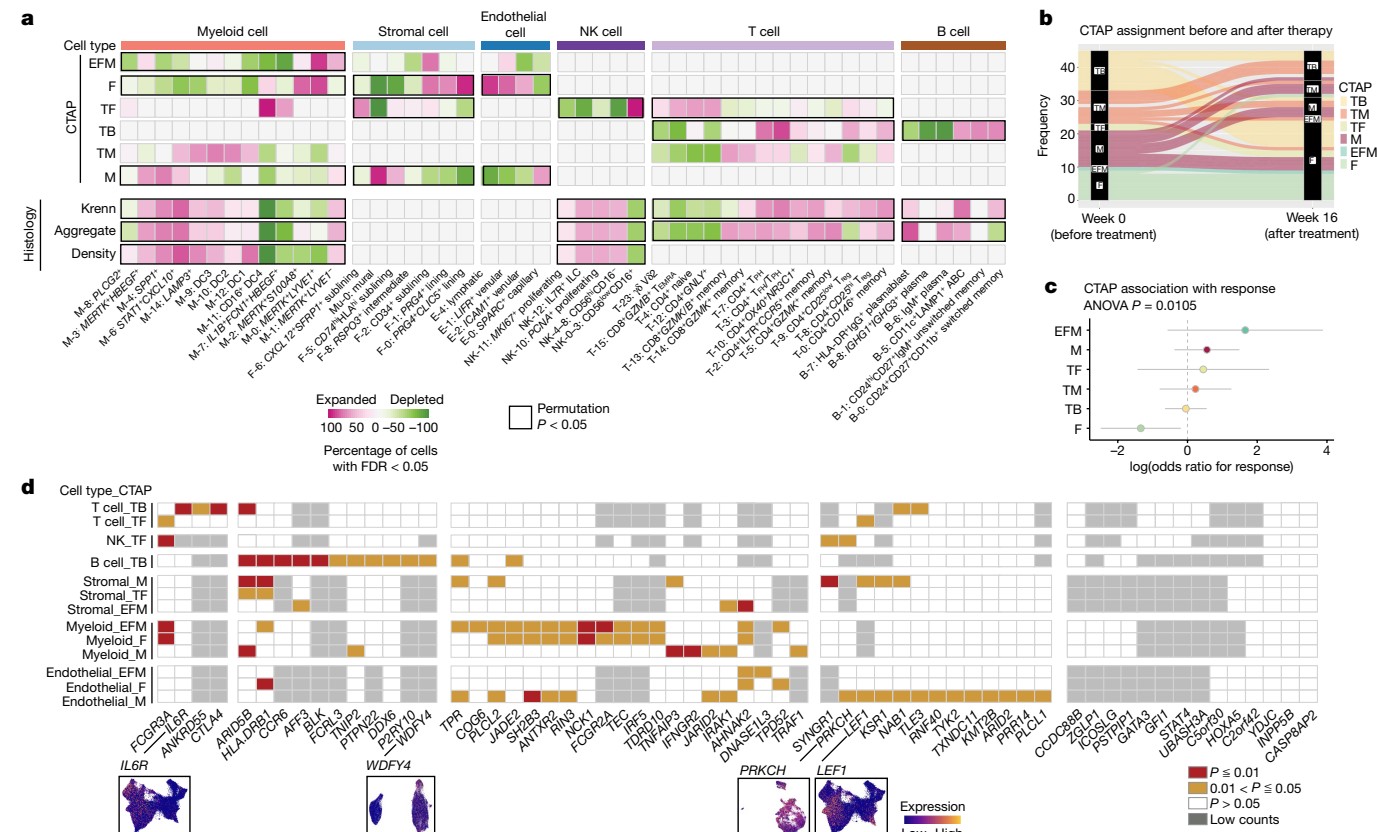

**Fig. 5 | Single-cell CNA reveals significant association of cell states with disease indicators, genetic factors and treatment response. a**, Heat map of CNA associations of specific cell states with each rheumatoid arthritis CTAP. Colours represent the percentage of cell neighbourhoods from each cell state with local (neighbourhood-level) phenotype correlations passing FDR < 0.05 significance from white to pink (expanded) or green (depleted). Cell types significantly associated globally (at cell-type level) with a phenotype at permutation $P < 0.05$ are boxed in black. **b**, Alluvial plot showing CTAP classification of samples prior to and at week 16 after starting treatment with either tocilizumab or rituximab ($n = 45$). **c**, Associations between clinical response and CTAPs after correcting for sex, age, treatment and CCP status in

the baseline (week 0) samples from the R4RA study ($n = 133$). The percentage of variance explained by CTAPs alone and $P$ value are calculated with ANOVA tests. Dots represent odds ratios and bars represent 95% confidence intervals. **d**, Significance of correlations between rheumatoid arthritis risk gene expression and CTAP-associated cells. Significance levels are shown in red ($P < 0.01$), yellow ($0.01 < P < 0.05$), and white ($P > 0.05$). Genes with low counts (more than one unique molecular identifier among less than 5% of cells with a given cell type) were not analysed in that cell type (grey boxes). Bottom, UMAPs displaying normalized expression levels of selected genes in T cells (*IL6R* and *LEF1*), B cells (*WDFY4*) and endothelial cells (*PRKCH*).

inflammatory phenotypes driving arthritis rather than differences in clinical disease activity.

To investigate whether CTAPs change over time, we applied our CTAP classification algorithm to 45 patients from the R4RA trial who had synovial tissue biopsies before and 16 weeks after starting treatment. CTAPs were dynamic during this period, with 30 out of 45 (67%) patients changing to a different CTAP (Fig. 5b and Extended Data Fig. 10e). Patients in the tocilizumab and rituximab treatment arms exhibited similar frequencies of CTAP change (20 out of 29 (69%) and 10 out of 16 (63%) patients, respectively) (Extended Data Fig. 10f–i). Among patients who changed CTAPs, CTAP-F was the most common CTAP at week 16 (16 out of 30 (53%)), consistent with rituximab and tocilizumab targeting inflammatory cells and pathways.

## Response to biologic therapy varies by CTAP

To determine whether CTAPs can predict the response to these treatments, we used our algorithm to determine the CTAPs of pre-treatment bulk RNA-seq for R4RA samples ($n = 133$). We then compared the frequencies of responders (defined as at least 50% improvement in CDAI) versus non-responders among the CTAPs (Extended Data Fig. 10j,k). We found that responses varied by CTAP ($P = 0.0105$), with CTAP-F having

the poorest response to both treatments, even after controlling for covariates (odds ratio = 0.2619, $P = 0.0403$; Fig. 5c).

## CTAP-enriched cell states express risk genes

We next tested whether genes implicated by recent multi-ancestry rheumatoid arthritis genetic studies are preferentially expressed by cell states associated with specific CTAPs[45,46]. We identified 71 genes that were likely to be causal, all of which were detected in one or more cell types in our dataset (Methods, Supplementary Fig. 13a and Supplementary Table 11).

We identified 48 genes with expression that was significantly positively correlated with CNA loadings for one or more CTAPs for a cell type ($P < 0.05$, controlling for expression level), indicating that cell states expanded in that CTAP specifically express the rheumatoid arthritis risk gene (Fig. 5d). This is significantly higher than predicted by chance (median = 34, permutation $P < 0.01$; Supplementary Fig. 13b,c). Some cell types expressed different rheumatoid arthritis genes in different subsets of cells (for example, *LEF1* in CTAP-TF-associated naive states and *IL6R* in CTAP-TB-associated $T_{FH}/T_{PH}$ states). *HLA-DRB1* expression was correlated with CTAP-associated cell states in several cell types (Fig. 5d). CTAP-associated rheumatoid arthritis risk genes may also be

expressed agnostic of CTAP in a given cell type, such as *IL6R* in myeloid cells (Supplementary Fig. 13d).

Some genes point to signalling pathways that may be important in a specific CTAP, such as VEGF in CTAP-M (Extended Data Fig. 9c). *PRKCH*—which encodes protein kinase C (PKC)-η, a mediator of VEGF-induced endothelial cell differentiation[47]—is highly expressed in endothelial cell states expanded in CTAP-M, which has high expression of VEGF receptor genes *KDR* and *FLT1* among expanded endothelial cell states and *VEGFA* among expanded myeloid cell states (Fig. 5d and Supplementary Fig. 13e–g).

## Discussion

We constructed a comprehensive rheumatoid arthritis synovial tissue reference of more than 314,000 single cells which revealed diverse cellular composition that we characterized into six CTAPs. Previously identified pathogenic cell states in rheumatoid arthritis are expanded in specific CTAPs. For example, CD4$^+$ T$_{FH}$ and T$_{PH}$ cells, which are enriched among T cells in rheumatoid arthritis compared with osteoarthritis[11], are present in synovium of all CTAPs but are most expanded in CTAP-TB. Our work also suggests the presence of extra-follicular activation pathways, especially in CTAP-TB, given the rarity of germinal centre dark-zone B cells and abundance of ABCs. Our study also provided more granular insights into previously identified pathogenic cells. For example, inflammatory sublining fibroblast subsets *CXCL12*$^+$ and *CD74*$^{hi}$*HLA*$^{hi}$ cells were enriched in CTAP-TF and CTAP-M, respectively. *MERTK*$^+$*HBEGF*$^+$ and *SPP1*$^+$ macrophages were also enriched in CTAP-M, probably reflecting different inflammatory axes. These and other instances of co-enriched populations (for example, *GZMK*$^+$ versus *GZMB*$^+$CD8$^+$ T and NK cells) inspire new questions about cell–cell interactions underlying inflammatory phenotypes in rheumatoid arthritis and other tissues and diseases.

We found that CTAPs are associated with histologic and serologic (CCP) parameters, in line with studies[48] that report increased lymphocyte infiltration (suggesting CTAP-TB, CTAP-TF or CTAP-TM) in CCP-positive synovium compared with CCP-negative synovium. Our finding that CTAP-M, and not CTAP-F or CTAP-EFM, was associated with CCP-negative status warrants further investigation in future studies.

CTAPs can be inferred from single-cell RNA-seq, bulk RNA-seq or flow cytometry data to provide cellular and molecular insights in clinical trials. Even within the more limited clinical diversity of the R4RA cohort[44], we found that CTAPs can change over time with treatment, and that CTAP-F was associated with poor clinical response. The dynamic heterogeneity of rheumatoid arthritis synovitis may explain the observation that clinical measures of patients treated with TNF inhibitors do not fall into a bimodal distribution of responders and non-responders[49]. It is possible that specific CTAPs are more likely to respond to specific therapies that preferentially target infiltrating cell types and relevant pathways. We anticipate that future longitudinal studies will investigate the association of CTAP changes with treatment effects across a larger array of treatments.

The CTAP paradigm provides a tissue classification system that captures coarse cell-type and fine cell-state heterogeneity. This model has the potential to serve as a powerful prototype to classify other types of tissue inflammation, including other immune-mediated diseases. A deeper understanding of the heterogeneity of tissue inflammation in rheumatoid arthritis and other autoimmune diseases may provide new insights into disease pathogenesis and reveal new treatment targets, and key elements of precision medicine.

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

¹Division of Rheumatology, Inflammation and Immunity, Department of Medicine, Brigham and Women's Hospital and Harvard Medical School, Boston, MA, USA. ²Center for Data Sciences, Brigham and Women's Hospital, Boston, MA, USA. ³Division of Genetics, Department of Medicine, Brigham and Women's Hospital and Harvard Medical School, Boston, MA, USA. ⁴Department of Biomedical Informatics, Harvard Medical School, Boston, MA, USA. ⁵Broad Institute of MIT and Harvard, Cambridge, MA, USA. ⁶Division of Rheumatology and the Center for Health Artificial Intelligence, University of Colorado School of Medicine, Aurora, CO, USA. ⁷Division of Immunology, Department of Pediatrics, Boston Children's Hospital and Harvard Medical School, Boston, MA, USA. ⁸Accelerating Medicines Partnership Program: Rheumatoid Arthritis and Systemic Lupus Erythematosus (AMP RA/SLE) Network, Bethesda, MD, USA. ⁹Rheumatology Research Group, Institute for Inflammation and Ageing, University of Birmingham, Birmingham, UK. ¹⁰Birmingham Tissue Analytics, Institute of Translational Medicine, University of Birmingham, Birmingham, UK. ¹¹Division of Allergy, Immunology and Rheumatology, Department of Medicine, University of Rochester Medical Center, Rochester, NY, USA. ¹²Hospital for Special Surgery, New York, NY, USA. ¹³Weill Cornell Medicine, New York, NY, USA. ¹⁴Center for Immunology and Inflammatory Diseases, Department of Medicine, Massachusetts General Hospital (MGH), Boston, MA, USA. ¹⁵Laboratory of Molecular Neuro-Oncology, The Rockefeller University, New York, NY, USA. ¹⁶Division of Rheumatology, Columbia University College of Physicians and Surgeons, New York, NY, USA. ¹⁷Division of Rheumatology, University of Colorado School of Medicine, Aurora, CO, USA. ¹⁸Division of Rheumatology, Allergy and Immunology, University of California, San Diego, La Jolla, CA, USA. ¹⁹NIHR Birmingham Biomedical Research Center and Clinical Research Facility, University of Birmingham, Queen Elizabeth Hospital, Birmingham, UK. ²⁰Division of Rheumatology, Cedars-Sinai Medical Center, Los Angeles, CA, USA. ²¹Division of Rheumatology, Department of Medicine, Northwestern University Feinberg School of Medicine, Chicago, IL, USA. ²²Centre for Experimental Medicine and Rheumatology, EULAR Centre of Excellence, William Harvey Research Institute, Queen Mary University of London, London, UK. ²³Barts Health NHS Trust, Barts Biomedical Research Centre (BRC), National Institute for Health and Care Research (NIHR), London, UK. ²⁴Department of Biomedical Sciences, Humanitas University and Humanitas Research Hospital, Milan, Italy. ²⁵Division of Clinical Immunology and Rheumatology, Department of Medicine, University of Alabama at Birmingham, Birmingham, AL, USA. ²⁶Feinstein Institute for Medical Research, Northwell Health, Manhasset, New York, NY, USA. ²⁷Department of Pathology and Laboratory Medicine, Hospital for Special Surgery, New York, NY, USA. ²⁸Department of Pathology and Laboratory Medicine, University of Rochester Medical Center, Rochester, NY, USA. ²⁹Division of Rheumatology and Clinical Immunology, University of Pittsburgh School of Medicine, Pittsburgh, PA, USA. ³⁸Present address: Division of Rheumatology, University of Colorado School of Medicine, Aurora, CO, USA. ³⁹These authors contributed equally: Fan Zhang, Anna Helena Jonsson, Aparna Nathan, Nghia Millard. ⁴⁰These authors jointly supervised this work: Kevin Wei, Deepak A. Rao, Laura T. Donlin, Jennifer H. Anolik, Michael B. Brenner, Soumya Raychaudhuri. *A list of authors and their affiliations appears at the end of the paper. ✉e-mail: soumya@broadinstitute.org

---

**Accelerating Medicines Partnership: RA/SLE Network**

Jennifer Albrecht¹¹, Jennifer L. Barnas¹¹, Joan M. Bathon¹⁶, David L. Boyle¹⁸, S. Louis Bridges Jr¹²,¹³, Debbie Campbell¹¹, Hayley L. Carr⁹,¹⁹, Adam Chicoine¹, Andrew Cordle³⁰, Patrick Dunn³¹,³², Lindsy Forbess²⁰, Peter K. Gregersen²⁶, Joel M. Guthridge³³, Lionel B. Ivashkiv¹²,¹³, Kazuyoshi Ishigaki¹,²,³,⁴,⁵,³⁴, Judith A. James³³, Gregory Keras¹, Ilya Korsunsky¹,²,³,⁴,⁵, Amit Lakhanpal¹²,¹³, James A. Lederer³⁵, Zhihan J. Li¹, Yuhong Li¹, Andrew McDavid³⁶, Mandy J. McGeachy²⁹, Karim Raza⁹,¹⁹, Yakir Reshef¹,²,³,⁴,⁵, Christopher Ritchlin¹¹, William H. Robinson³⁷, Saori Sakaue¹,²,³,⁴,⁵, Jennifer A. Seifert¹⁷, Anvita Singaraju¹²,¹³, Melanie H. Smith¹², Dagmar Scheel-Toellner⁹,¹⁹, Paul J. Utz³⁷, Michael H. Weisman²⁰,³⁷, Aaron Wyse³⁰ & Zhu Zhu¹

³⁰Department of Radiology, University of Pittsburgh Medical Center, Pittsburgh, PA, USA. ³¹Division of Allergy, Immunology and Transplantation, National Institute of Allergy and Infectious Diseases, National Institutes of Health, Bethesda, MD, USA. ³²Northrop Grumman Health Solutions, Rockville, MD, USA. ³³Department of Arthritis and Clinical Immunology, Oklahoma Medical Research Foundation, Oklahoma City, OK, USA. ³⁴Laboratory for Human Immunogenetics, RIKEN Center for Integrative Medical Sciences, Yokohama, Japan. ³⁵Department of Surgery, Brigham and Women's Hospital and Harvard Medical School, Boston, MA, USA. ³⁶Department of Biostatistics and Computational Biology, University of Rochester School of Medicine and Dentistry, Rochester, NY, USA. ³⁷Division of Immunology and Rheumatology, Institute for Immunity, Transplantation and Infection, Stanford University School of Medicine, Stanford, CA, USA.

## Reporting summary

Further information on research design is available in the Nature Portfolio Reporting Summary linked to this article.

## Data availability

CITE-seq single-cell expression matrices and sequencing and bulk expression matrices are available on Synapse (https://doi.org/10.7303/syn52297840). Associated genotype and clinical data are available through the Arthritis and Autoimmune and Related Diseases Knowledge Portal (ARK Portal, https://arkportal.synapse.org/Explore/Datasets/DetailsPage?id=syn52297840). A cell browser website https://immunogenomics.io/ampra2/ is available to visualize our data and results. AMP Phase 1 single-cell data from ref. 8 are available on Immport (SDY998). PEAC clinical trial RNA-seq data from ref. 6 are available on ArrayExpress (E-MTAB-6141). R4RA clinical trial RNA-seq data from ref. 44 are available on ArrayExpress (E-MTAB-11611). Single-cell and bulk RNA-seq data were aligned to GRCh38 (Ensembl 93), available as part of Cell Ranger v. 3.1.0.

## Code availability

The source code for the analyses is available at https://github.com/Immunogenomics/RA_Atlas_CITEseq/ and https://zenodo.org/record/8118599.

**Acknowledgements** This work was supported by the Accelerating Medicines Partnership® Rheumatoid Arthritis and Systemic Lupus Erythematosus (AMP® RA/SLE) Network. AMP is a public-private partnership (AbbVie, Arthritis Foundation, Bristol-Myers Squibb Company, Foundation for the National Institutes of Health, GlaxoSmithKline, Janssen Research and Development, LLC, Lupus Foundation of America, Lupus Research Alliance, Merck & Co., Inc., National Institute of Allergy and Infectious Diseases, National Institute of Arthritis and Musculoskeletal and Skin Diseases, Pfizer, Inc., Rheumatology Research Foundation, Sanofi and Takeda Pharmaceuticals International, Inc.) created to develop new ways of identifying and validating promising biological targets for diagnostics and drug development. Funding was provided through grants from the National Institutes of Health (UH2-AR067676, UH2-AR067677, UH2-AR067679, UH2-AR067681, UH2-AR067685, UH2- AR067688, UH2-AR067689, UH2-AR067690, UH2-AR067691, UH2-AR067694 and UM2- AR067678). Accelerating Medicines Partnership and AMP are registered service marks of the US Department of Health and Human Services. This work was also supported by the PhRMA Foundation and the Arthritis National Research Foundation Award (to F.Z.); NIH NIAMS K08AR081412, Rheumatology Research Foundation Investigator Award, and Arthritis National Research Foundation Award (to A.H.J.); NIH NHGRI T32HG002295 and NIAMS T32AR007530 (to A. Nathan); NIH NIAMS K08AR077037, Rheumatology Research Foundation Innovative Research Award, and Burroughs Wellcome Fund Career Award in Medical Sciences (to K. Wei); NIH NIGMS T32GM007753 (to J.B.K.); NIH NIAMS R01078268 and NCATS UL1 TR001866 (to D.E.O.); NIH NIAID T32AR007258 (to K.S.); Research into Inflammatory Arthritis Centre Versus Arthritis (22072), IMI-RTCure (777357) and the NIHR Birmingham Biomedical Research Centre (BRC-1215-20009) (to A.F. and D.S.T.); NIH NIAMS K08AR072791 and Burroughs Wellcome Fund Career Award in Medical Sciences (to D.A.R.); NIH NIAID R01AI148435 (to L.T.D.); NIH NIAMS R21AR071670 and P30 AR069655 (to J.H.A.); NIH NIAMS R01AR073833 and R01AR073290 (to M.B.B.); NIH NHGRI U01HG009379 and NIAMS R01AR063759 (to S.R.). We especially acknowledge people in the AMP RA/SLE Network: A. Arazi, C. Berthier, J. Buyon, M. Dall'Era, A. Davidson, B. Diamond, A. Fava, J. Grossman, N. Hacohen, D. Hildeman, J. Hodgin, T. Hwang, M. Ishimori, K. Kalunian, D. Kamen, M. Kretzler, H. Maecker, R. Mao, M. McMahon, F. Payan-Schober, M. Petri, C. Putterman, D. Simmons, T. Tuschl, D. Wofsy and Steve Woodle.

**Author contributions** L.G.-P., K.D.D., D.T., A.C., G.S.F., M.M., I.S., A.B.-A., A.M.M., A. Nerviani, F.R., C.P., L.B.H. and D.H., recruited patients and obtained synovial tissues. L.W.M., S.M.G., H.P., V.M.H., A.F., V.P.B. and J.H.A. contributed to the procurement and processing of samples and design of the AMP study. E.D., E.M.G. and B.F.B. performed histologic assessment of tissues. D.W., K.P.L., A.F. and V.P.B. curated and analysed histologic and clinical data. W.A. provided project management and curated histologic and clinical data. K. Wei, A.H.J., G.F.M.W., A. Nathan and M.B.B. designed and implemented the tissue disaggregation, cell sorting and single-cell sequencing pipeline. A.H.J., K. Wei and G.F.M.W. supervised and executed the tissue disaggregation pipeline. S.N., J.R.-M. and N. Meednu. performed immunofluorescence microscopy and analysed these data together with M.C. and A.H.J. K.E.M. and I.M. performed and analysed functional cellular assays. M.J.L., F.R. and C.P. contributed unpublished data from clinical trials. F.Z., A. Nathan, N. Millard, M.C., Q.X., M.G.-A., J.B.K., K. Weinand, J.M., L.R. and S.R. conducted computational and statistical analysis. A.H.J., K. Wei, M.B.B., J.H.A., L.T.D., D.A.R., F.Z., A. Nathan, S.R., D.E.O., J.R.-M. and A.F. provided input on cellular analysis and interpretation. D.E.O., J.R.-M., A.F. and J.H.A. provided input on histologic analyses. N. Millard and K.S. implemented the website. S.R., M.B.B., J.H.A., L.T.D. and D.A.R. supervised the research. F.Z., A.H.J., A. Nathan, N. Millard, Q.X. and S.R. wrote the initial draft. F.Z., A.H.J., A. Nathan, K. Wei, N. Millard, D.A.R, L.T.D., J.H.A, M.B.B. and S.R. edited the draft. AMP RA/SLE Network members contributed to this work by managing patient recruitment, curating clinical data, obtaining and processing synovial tissue samples, managing biorepositories, conducting histologic or computational analysis, providing software code, providing website support and/or providing input on data analysis and interpretation. All authors participated in editing the final manuscript.

**Competing interests** A.H.J. reports research support from Amgen outside the submitted work. K. Wei is a consultant for Mestag Therapeutics and Gilead Sciences and reports grant support from Gilead Sciences. S.M.G. reports research support from Novartis and is a consultant for UCB, outside the submitted work. V.M.H. is a co-founder of Q32 Bio and has previously received sponsored research from Janssen and been a consultant for Celgene and BMS, outside the submitted work. A.F. reports personal fees from Abbvie, Roche and Janssen and grant support from Roche, UCB, Nascient, Mestag, GlaxoSmithKline and Janssen, outside the submitted work. D.A.R. reports personal fees from Pfizer, Janssen, Merck, Scipher Medicine, GlaxoSmithKline and Bristol-Myers Squibb and grant support from Janssen and Bristol-Myers Squibb, outside the submitted work. In addition, D.A.R. is a co-inventor on a patent submitted on T peripheral helper cells. M.B.B. is a founder for Mestag Therapeutics and a consultant for GlaxoSmithKline, 4FO Ventures and Scailyte AG. S.R. is a founder for Mestag Therapeutics, a scientific advisor for Janssen and Pfizer, and a consultant for Gilead and Rheos Medicines.

**Additional information**
**Correspondence and requests for materials** should be addressed to Soumya Raychaudhuri.

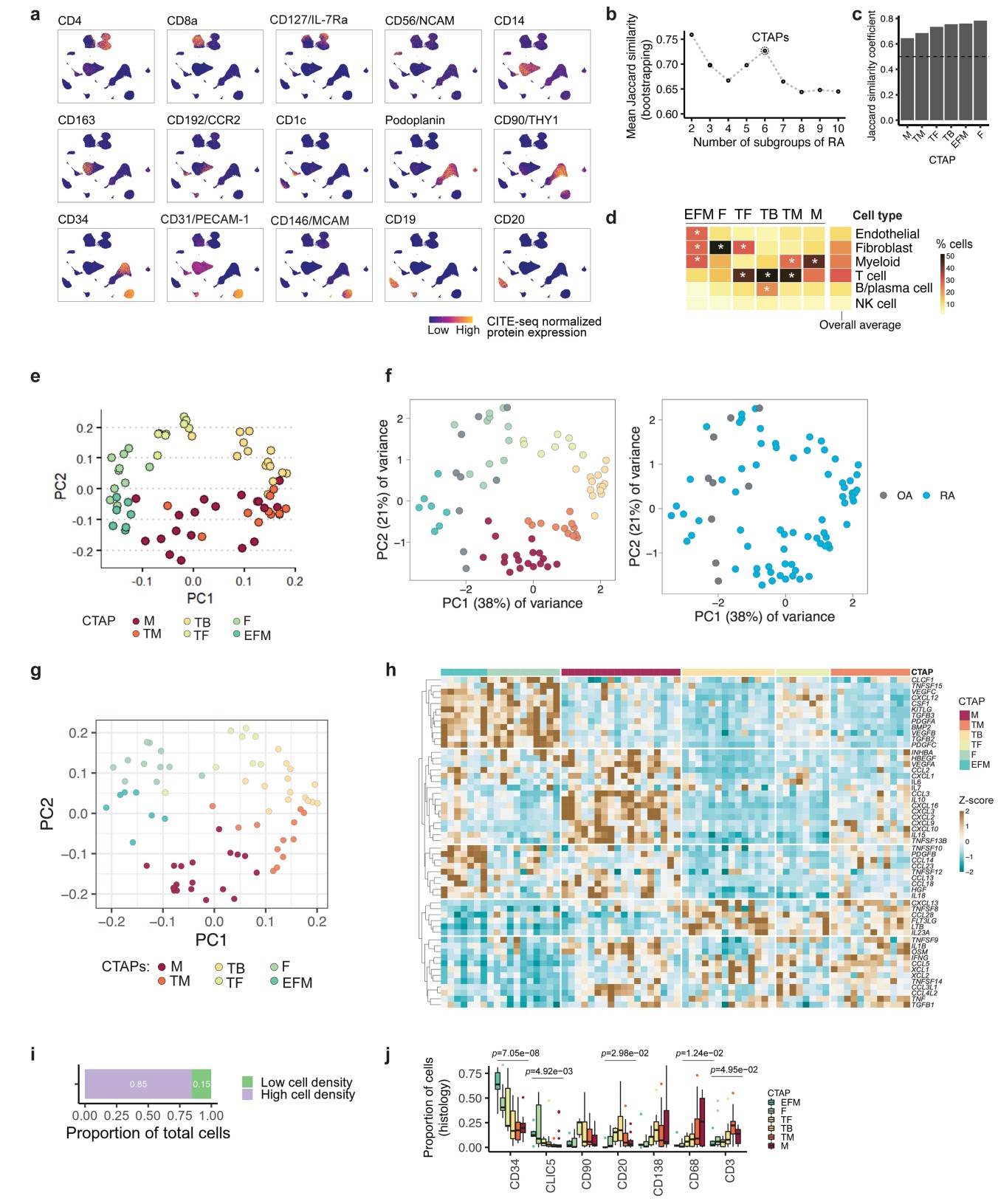

**Extended Data Fig. 1** | See next page for caption.

**Extended Data Fig. 1 | Robust CTAP definition and quantitative cellular histology analysis. a**, UMAPs of CITE-seq antibody-based expression of cell-type lineage protein markers. Cells are colored based on expression from blue (low) to yellow (high). **b**, Mean Jaccard similarity coefficient to test CTAP stability by bootstrapping 10,000 times for each tested number of patient subgroups ranging from 2 to 10. **c**, Mean Jaccard similarity coefficient for each CTAP, comparing full clustering and 10,000 bootstrapped datasets. **d**, Average proportions of each major cell type among samples in each CTAP. Overall average proportions across all the samples are shown as a comparator. Asterisk represents the proportion that is greater than the overall average for that cell type, **e**, PCA of samples based on cell-type abundances, adjusting for disease duration and treatment. Each dot represents a sample, plotted based on its PC1 and PC2 projections and colored by CTAPs. **f**, Projection of OA samples onto PCA of samples based on cell-type abundances from Fig. 1j. OA samples are marked with gray points; RA samples are colored based on CTAP (left) or in blue (right). **g**, PCA of samples based on pseudo-bulk gene expression of 55 soluble immune mediators. Each dot represents a sample, plotted based on its PC1 and PC2 projections and colored by CTAPs. **h**, Heatmap of pseudo-bulk gene expression of soluble immune mediators across samples, grouped by CTAP. Boxes are colored based on the gene's scaled pseudo-bulk expression across samples. **i**, Bar graph of the proportion of total cells located in high-density and low-density fragments, as captured by histology imaging. Quantitation of total cellular composition demonstrated that fragments with highest cell density (top 50%) contained 86% of total cells and are therefore likely the primary drivers of CTAP classification. **j**, Box plots of the proportion of cells in high-density fragments (N = 76) expressing each marker in histology imaging, stratified by CTAP. Points represent outlier samples (> 1.5 * IQR from median). Box plots show median (vertical bar), 25th and 75th percentiles (lower and upper bounds of the box, respectively) and 1.5 x IQR (or minimum/maximum values; end of whiskers). P-values are calculated with one-way ANOVA tests with Bonferroni correction.

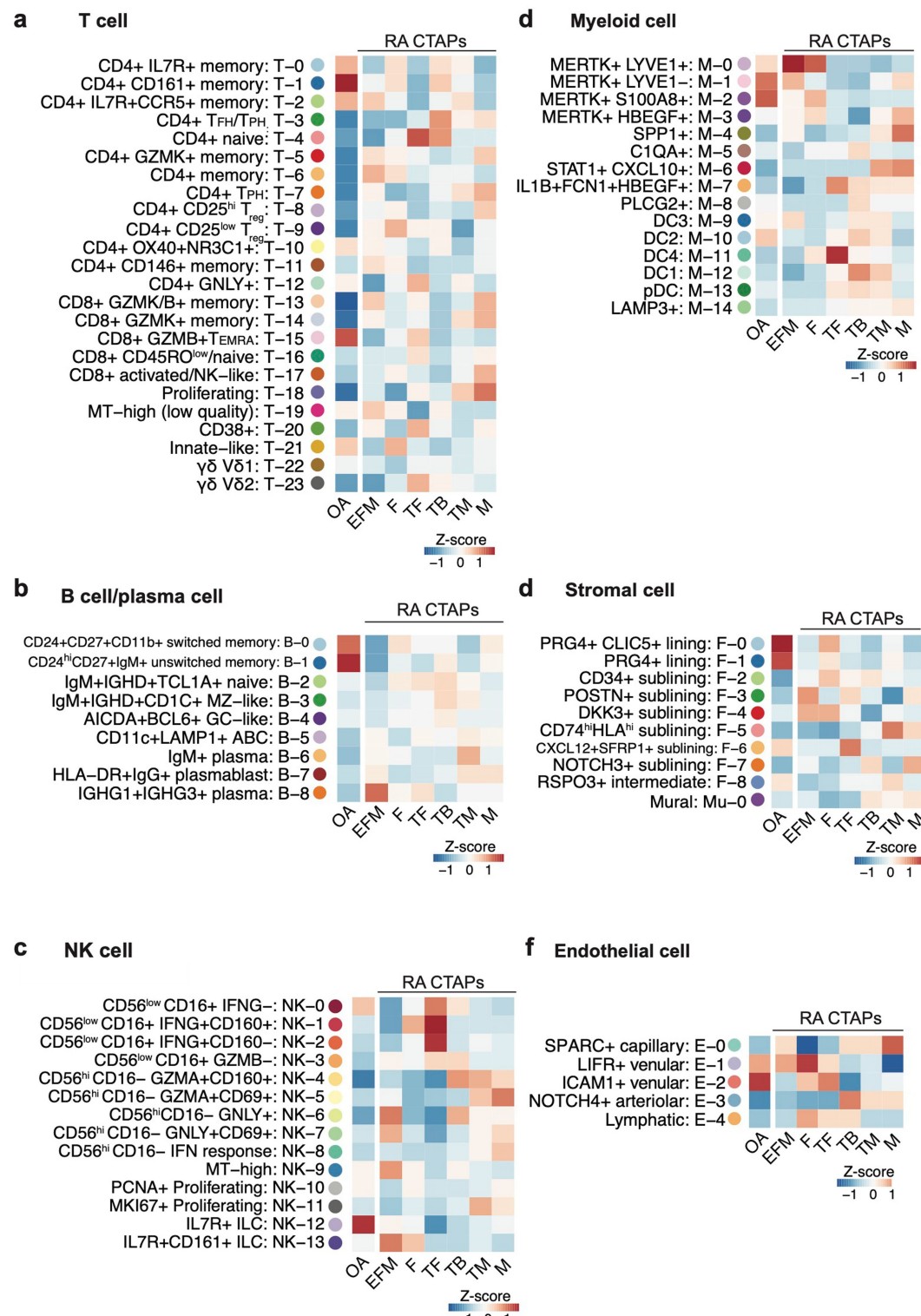

**Extended Data Fig. 2 | Relative enrichment of fine-grain cell clusters across CTAPs and OA. a-f,** Heatmaps show the average proportions of each cluster in the given cell type across patient samples in each RA CTAP and OA, scaled within each cluster.

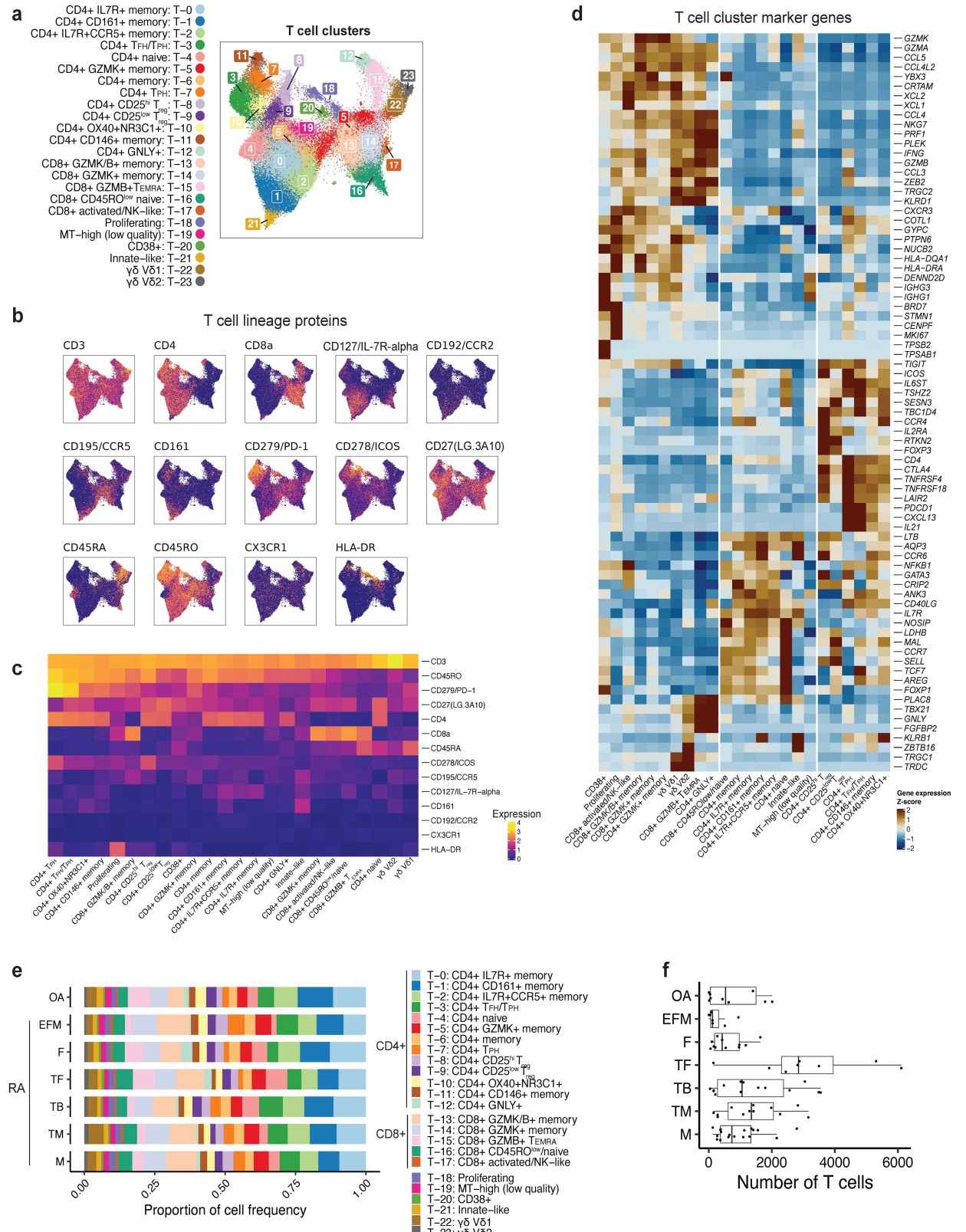

**Extended Data Fig. 3 | T cell-specific analysis. a**, T cell UMAP colored by fine-grained cell-state clusters, **b**, Expression of selected surface proteins among T cells. Cells are colored from blue (low) to yellow (high), **c**, Heatmap of surface protein expression in T cell clusters colored according to the average normalized expression across cells in the cluster, **d**, Heatmap of gene expression in T cell clusters colored according to the average normalized expression across cells in the cluster, scaled for each gene across clusters,

**e**, Distribution of T cells across clusters, stratified by CTAP. The size of each segment of each bar corresponds to the average proportion of cells in that cluster across donors from that CTAP. **f**, Number of T cells per individual, stratified by CTAP. Points represent individuals (N = 82); OA (N = 9), (EFM (N = 7), F (N = 11), TF (N = 8), TB (N = 14), TM (n = 12), M (N = 18). Box plots show median (vertical bar), 25th and 75th percentiles (lower and upper bounds of the box, respectively) and 1.5 x IQR (or minimum/maximum values; end of whiskers).

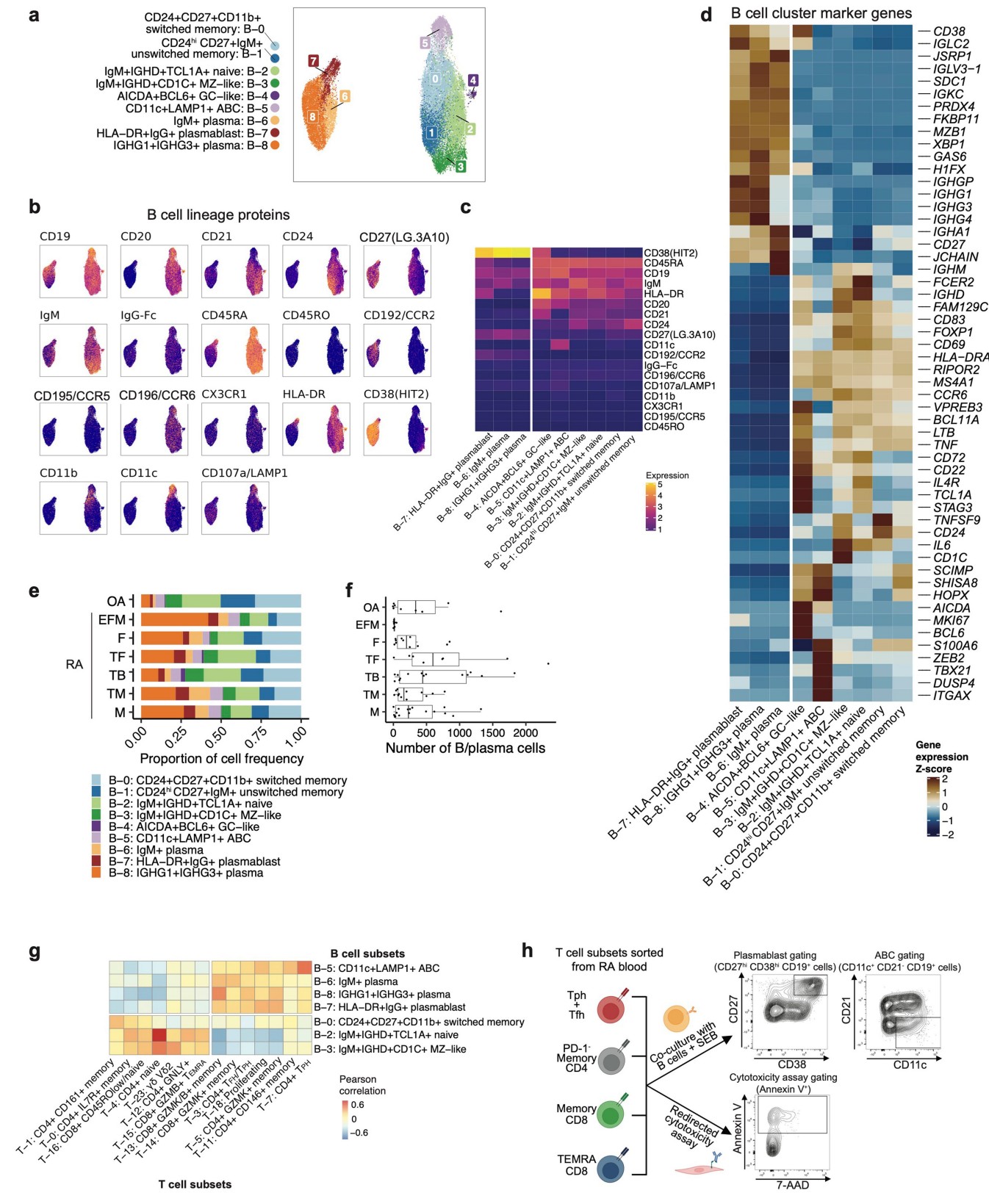

**Extended Data Fig. 4** | See next page for caption.

**Extended Data Fig. 4 | B/plasma cell-specific analysis. a**, B/plasma cell UMAP colored by fine-grained cell state clusters, **b** Expression of selected surface proteins among B/plasma cells. Cells are colored from blue (low) to yellow (high), **c**, Heatmap of surface protein expression in B/plasma cell clusters colored according to the average normalized expression across cells in the cluster, **d**, Heatmap of gene expression in B/plasma cell clusters colored according to the average normalized expression across cells in the cluster, scaled for each gene across clusters, **e**, Distribution of B/plasma cells across clusters, stratified by CTAP. The size of each segment of each bar corresponds to the average proportion of cells in that cluster across donors from that CTAP. **f**, Number of B/plasma cells per individual, stratified by CTAP. Points represent individuals (N = 82); OA (N = 9), EFM (N = 7), F (N = 11), TF (N = 8), TB (N = 14), TM (N = 12), M (N = 18). Box plots show median (vertical bar), 25th and 75th percentiles (lower and upper bounds of the box, respectively) and 1.5 x IQR (or minimum/maximum values; end of whiskers). **g**, Heatmap of correlations between select T and B cell subsets, colored by Pearson correlation between per-donor proportions. **h**, Schematic representation of the experimental design of the T cell functional assays and representative flow cytometry plots showing gating of plasmablasts (CD27$^{hi}$ CD38$^{hi}$ CD19$^+$ cells), ABC B cells (CD11c$^+$ CD21$^-$ CD19$^+$ cells) and dead target cells (Annexin V$^+$). Parts of this schematic were created using BioRender.

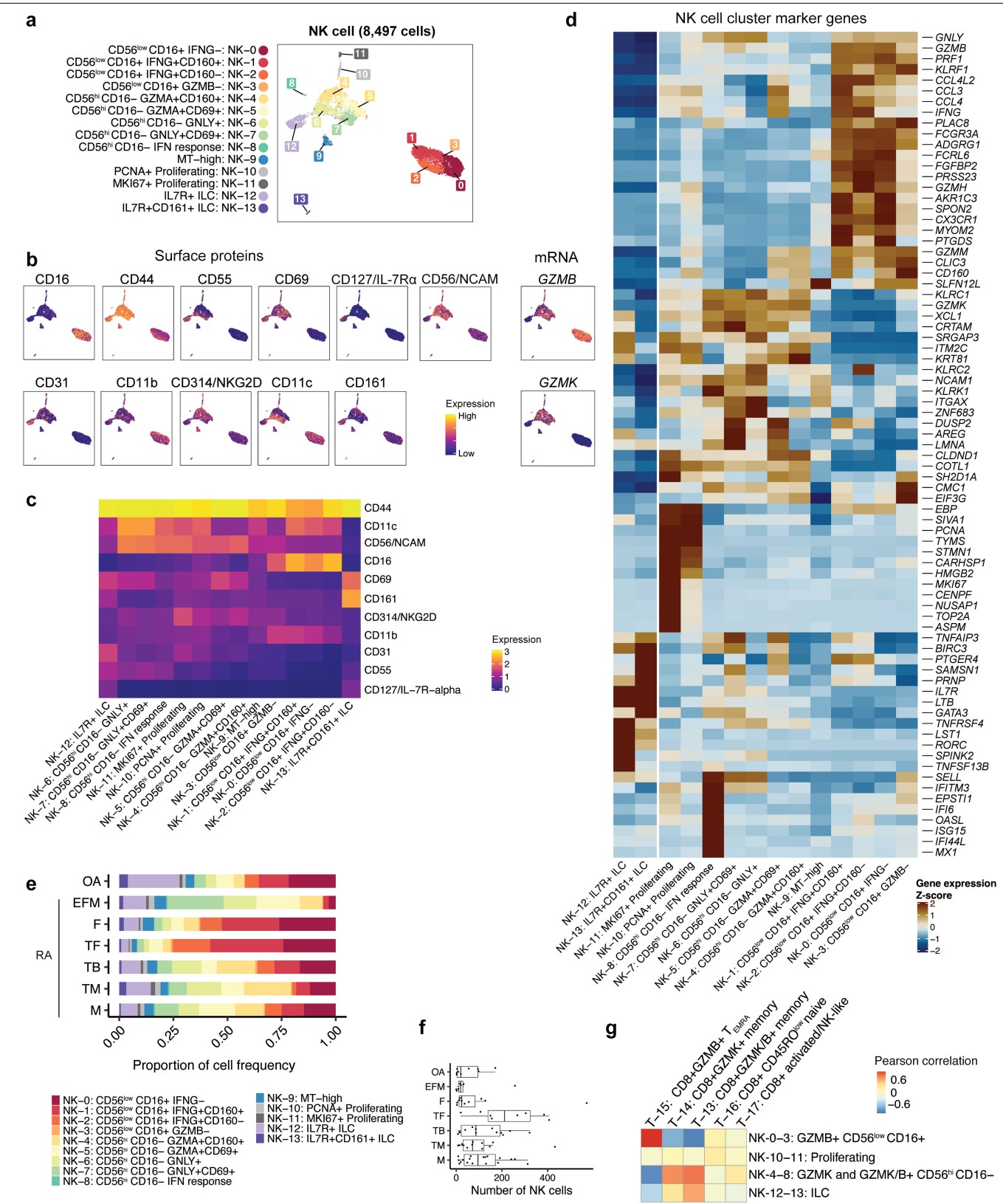

**Extended Data Fig. 5 | NK cell-specific analysis. a**, NK cell UMAP colored by fine-grained cell state clusters, **b**, Expression of selected surface proteins or mRNA transcripts among NK cells colored from blue (low) to yellow (high), **c**, Heatmap of surface protein expression in NK cell clusters colored according to the average normalized expression across cells in the cluster, **d**, Heatmap of gene expression in NK cell clusters colored according to the average normalized expression across cells in the cluster, scaled for each gene across clusters, **e**, Distribution of NK cells across clusters, stratified by CTAP. The size of each segment of each bar corresponds to the average proportion of cells in that cluster across donors from that CTAP. **f**, Number of NK cells per individual, stratified by CTAP. Points represent individuals (N = 82); OA (N = 9), EFM (N = 7), F (N = 11), TF (N = 8), TB (N = 14), TM (N = 12), M (N = 18). Box plots show median (vertical bar), 25th and 75th percentiles (lower and upper bounds of the box, respectively) and 1.5 x IQR (or minimum/maximum values; end of whiskers). **g**, Heatmap colored by Pearson correlation between per-donor CD8+ T cell and NK cell cluster abundances.

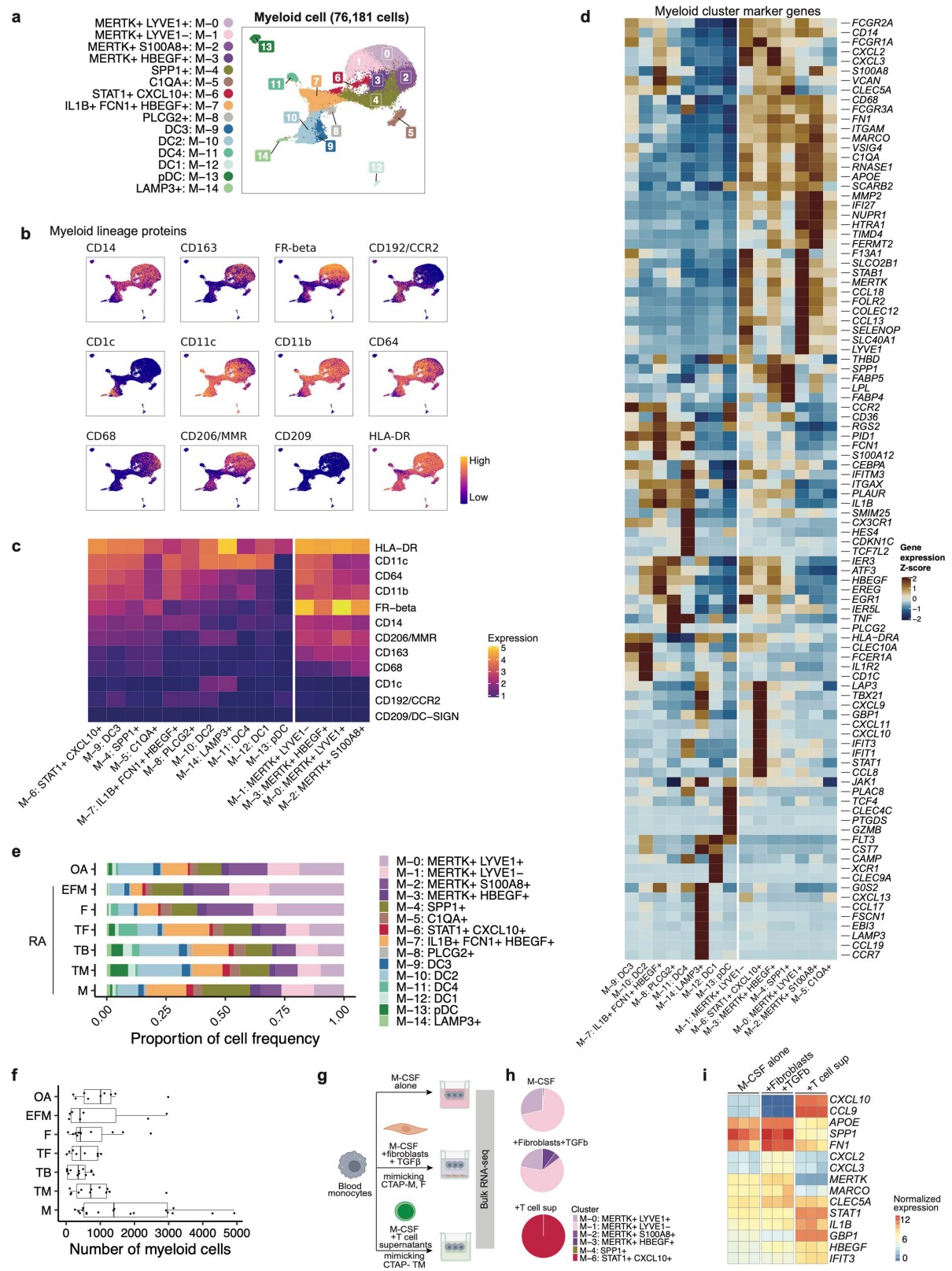

**Extended Data Fig. 6** | See next page for caption.

**Extended Data Fig. 6 | Myeloid cell-specific analysis. a**, Myeloid cell UMAP colored by fine-grained cell state clusters, **b**, Expression of selected surface proteins among myeloid cells colored from blue (low) to yellow (high), **c**, Heatmap of surface protein expression in myeloid cell clusters colored according to the average normalized expression across cells in the cluster, **d**, Heatmap of gene expression in myeloid cell clusters colored according to the average normalized expression across cells in the cluster, scaled for each gene across clusters, **e**, Distribution of myeloid cells across clusters, stratified by CTAP. The size of each segment of each bar corresponds to the average proportion of cells in that cluster across donors from that CTAP. **f**, Number of myeloid cells per individual, stratified by CTAP. Points represent individuals (N = 82); OA (N = 9), EFM (N = 7), F (N = 11), TF (N = 8), TB (N = 14), TM (N = 12), M (N = 18). Box plots show median (vertical bar), 25th and 75th percentiles (lower and upper bounds of the box, respectively) and 1.5 x IQR (or minimum/maximum values; end of whiskers). **g**, Schematic representation of the experimental design of the myeloid cell assays. Parts of this schematic were created using BioRender. **h**, Linear discriminant analysis classification of bulk RNA-seq obtained from myeloid cells cultured in the indicated conditions. Each condition was performed with three biological replicates, and cluster proportions in each pie chart were calculated from the mean of the posterior probability values across replicates. **i**, Heatmap showing expression of selected CTAP-relevant genes in bulk RNA-seq of blood monocytes cultured in the indicated conditions. Columns correspond to three biological replicates for each condition, and boxes are colored by normalized gene expression.

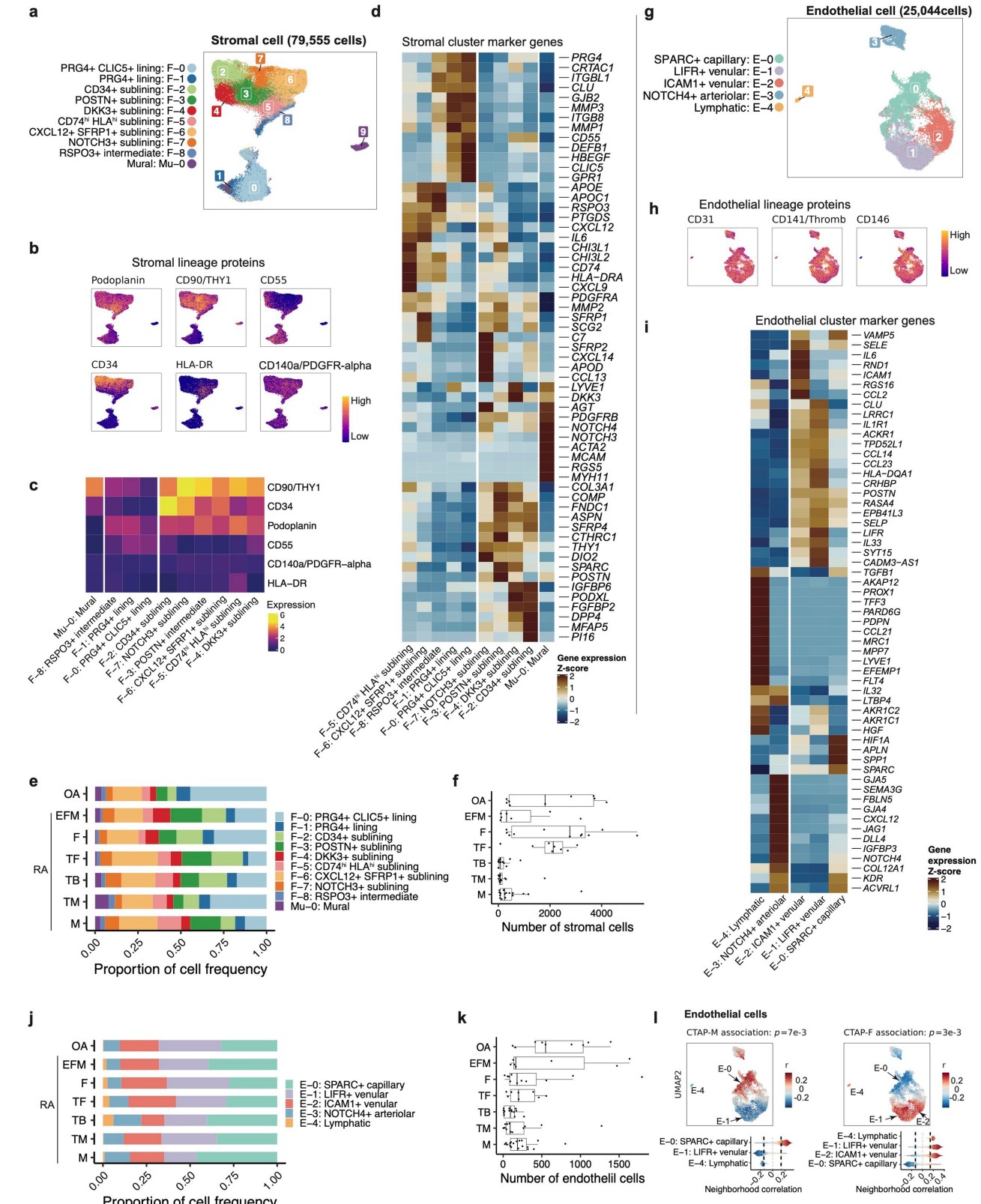

**Extended Data Fig. 7** | See next page for caption.

**Extended Data Fig. 7 | Stromal- and endothelial-specific analysis. a**, Stromal cell UMAP colored by fine-grained cell state clusters, **b**, Expression of selected surface proteins among stromal cells colored from blue (low) to yellow (high), **c**, Heatmap of surface protein expression in stromal cell clusters colored according to the average normalized expression across cells in the cluster, **d**, Heatmap of gene expression in stromal cell clusters colored according to the average normalized expression across cells in the cluster, scaled for each gene across clusters, **e**, Distribution of stromal cells across clusters, stratified by CTAP. The size of each segment of each bar corresponds to the average proportion of cells in that cluster across donors from that CTAP, **f**, Number of stromal cells per individual, stratified by CTAP. Points represent individuals (N = 82); OA (N = 9), (EFM (N = 7), F (N = 11), TF (N = 8), TB (N = 14), TM (N = 12), M (N = 18). Box plots show median (vertical bar), 25th and 75th percentiles (lower and upper bounds of the box, respectively) and 1.5 x IQR (or minimum/maximum values; end of whiskers), **g**, Endothelial cell UMAP colored by fine-grained cell state clusters, **h**, Expression of selected surface proteins among endothelial cells colored from blue (low) to yellow (high), **i**, Heatmap of gene expression in endothelial cell clusters colored according to the average normalized expression across cells in the cluster, scaled for each gene across clusters, **j**, Distribution of endothelial cells across clusters, stratified by CTAP. The size of each segment of each bar corresponds to the average proportion of cells in that cluster across donors from that CTAP. **k**, Number of endothelial cells per individual, stratified by CTAP. Points represent individuals (N = 82); OA (N = 9), EFM (N = 7), F (N = 11), TF (N = 8), TB (N = 14), TM (n = 12), M (N = 18). Box plots show median (vertical bar), 25th and 75th percentiles (lower and upper bounds of the box, respectively) and 1.5 x IQR (or minimum/maximum values; end of whiskers). **l**, Association of endothelial cell neighborhoods with CTAP-M and CTAP-F. For these CNA results, cells in UMAPs are colored in red (positive) or blue (negative) if their neighborhood is significantly associated with the CTAP (FDR < 0.05), and gray otherwise. Distributions of neighborhood correlations are shown for clusters with >50% of neighborhoods correlated with the CTAP at FDR < 0.05; global p-values were obtained based on the permutation testing from the CNA package.

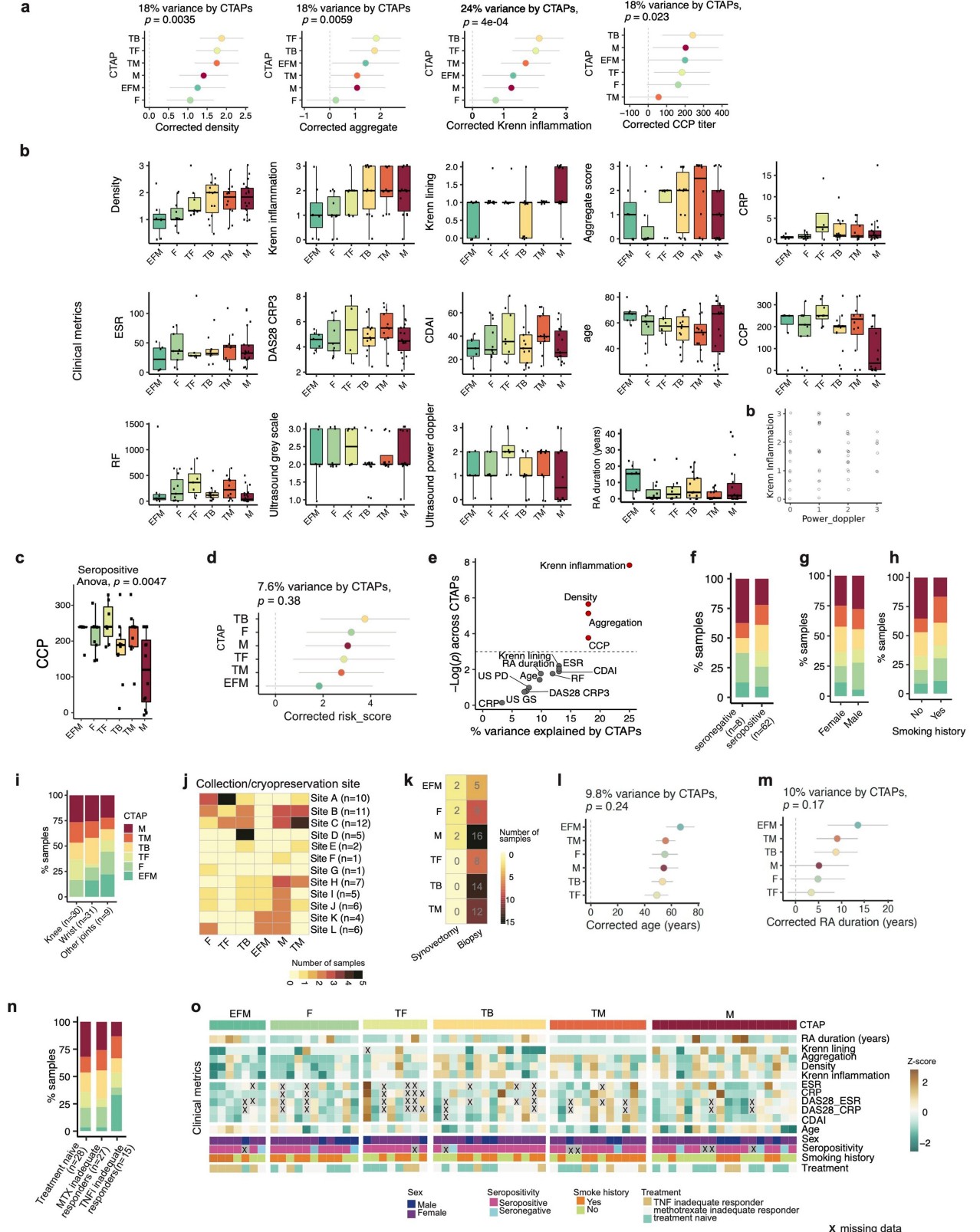

**Extended Data Fig. 8** | See next page for caption.

**Extended Data Fig. 8 | Association of single-cell RA CTAPs with different clinical characteristics. a**, Associations between clinical features and CTAPs (N = 70), adjusting covariates for age, sex, cell number, and clinical collection site. Percentage of variance explained by CTAPs alone and p-value are calculated with ANOVA tests. Points represent odds ratios and bars represent 95% confidence intervals. **b**, Clinical, histologic, and ultrasound parameters of patients in each CTAP. For all box plots, each dot represents an individual (N = 70); boxes show median (vertical bar), 25th and 75th percentiles (lower and upper bounds of the box, respectively) and 1.5 x IQR (or minimum/maximum values; end of whiskers), **c**, Dotplot of Krenn inflammation versus power doppler scores. Each point is a patient. **d**, CCP levels among seropositive patients alone (N = 59). Points represent individuals and box plots show median (vertical bar), 25th and 75th percentiles (lower and upper bounds of the box, respectively) and 1.5 x IQR (or minimum/maximum values; end of whiskers)., **e**, Corrected RA *HLA-DRB1* risk scores and their associations with CTAPs, percent of variance explained by CTAPs only and p-value are calculated with ANOVA test, **f**, Clinical, demographic, and histologic metrics plotted by percentage of variance explained by CTAPs and the ANOVA p-value for its association with CTAPs. Features in red are significant at $p < 0.05$. **g**, CTAP frequency among seropositive (CCP-positive, RF-positive, or both) versus seronegative patients. **h**, CTAP frequency by sex. **i**, CTAP frequency by smoking history, **j**, CTAP frequency by anatomic site of synovial biopsy. **k**, Number of samples per CTAP in each collection/cryopreservation site. **l**, Number of patient samples for each CTAP between biopsy and synovectomy, **m-n**, Association of age and RA duration with CTAPs (N = 70), adjusting covariates for age, sex, cell number, and clinical collection site. Points represent odds ratios and bars represent 95% confidence intervals. Percentage of variance explained by CTAPs alone and p-values are calculated with one-way ANOVA tests. **o**, Sample distributions across CTAPs by recruitment cohort, **p**, Heatmap of clinical variables for patient samples grouped by CTAP. Boxes are colored based on z-score of the metric across samples. "X" represents missing data.

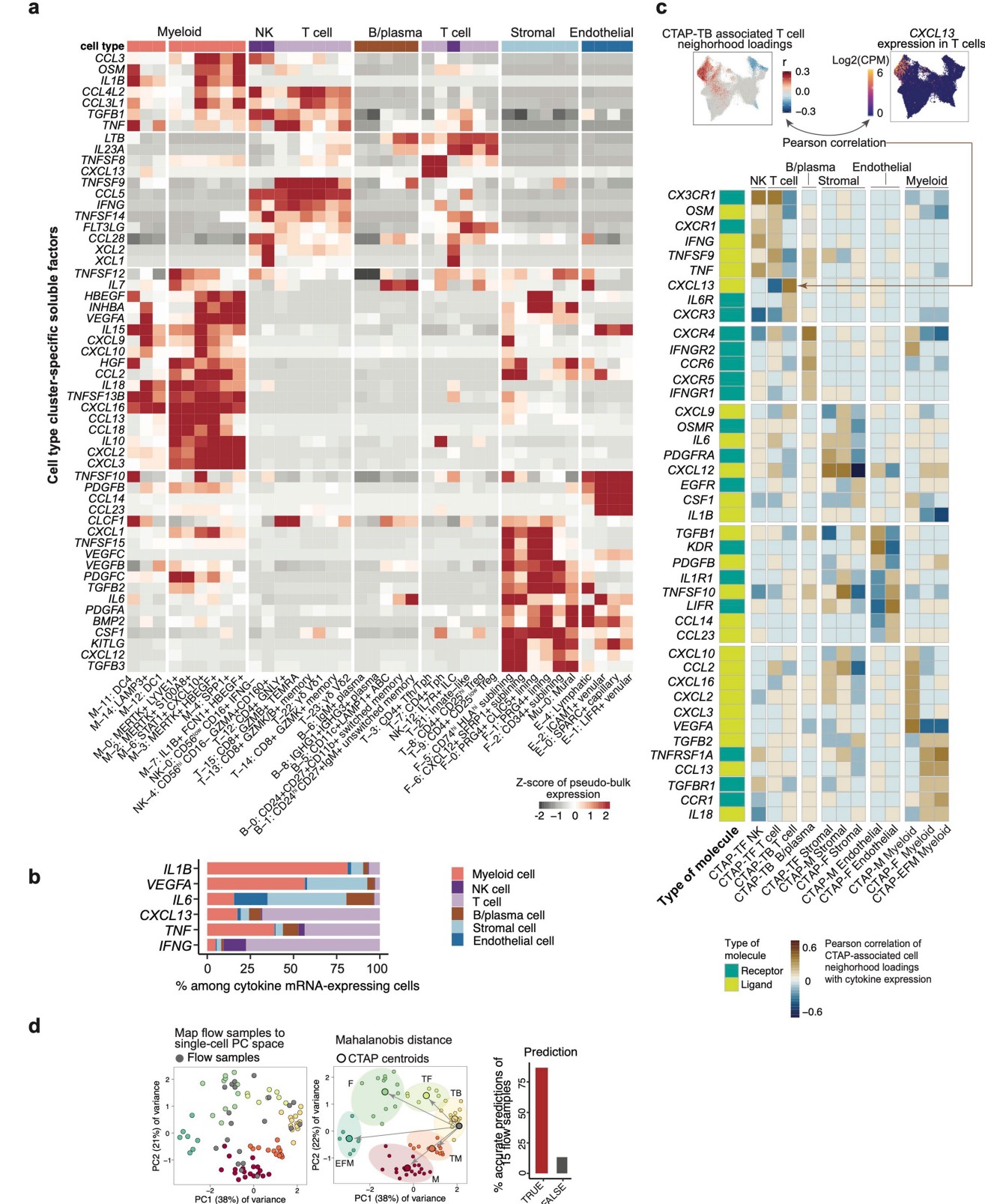

**Extended Data Fig. 9 | Correlations of cytokines/receptors with CTAP-associated cells. a**, Heatmap depicting expression profiles of cell type cluster-specific soluble factors, **b**, Percent contribution among cytokine mRNA-expressing cells from each major cell type, **c**, At top, expression of *CXCL13*, a representative cytokine that is significantly correlated with CTAP-associated cell neighborhoods. Cells in UMAPs of CTAP associations are colored in red (positive) or blue (negative) if their neighborhood is significantly associated with the CTAP (FDR < 0.05), and gray otherwise. Cells in expression UMAPs are colored from blue (low) to yellow (high). Below, an aggregate heatmap visualizing the cytokines and receptors whose expressions are significantly correlated (r > 0.5) with CTAP-associated cells; we then hierarchically clustered them based on cell type-specific CTAPs. Each gene is labeled with receptor/ligand designation. **d**, Pipeline and results to map and classify flow cytometry samples by single-cell RA CTAPs. Bar plot shows accuracy of flow sample classification (i.e., assigned to the same CTAP as a single-cell sample from the same patient).

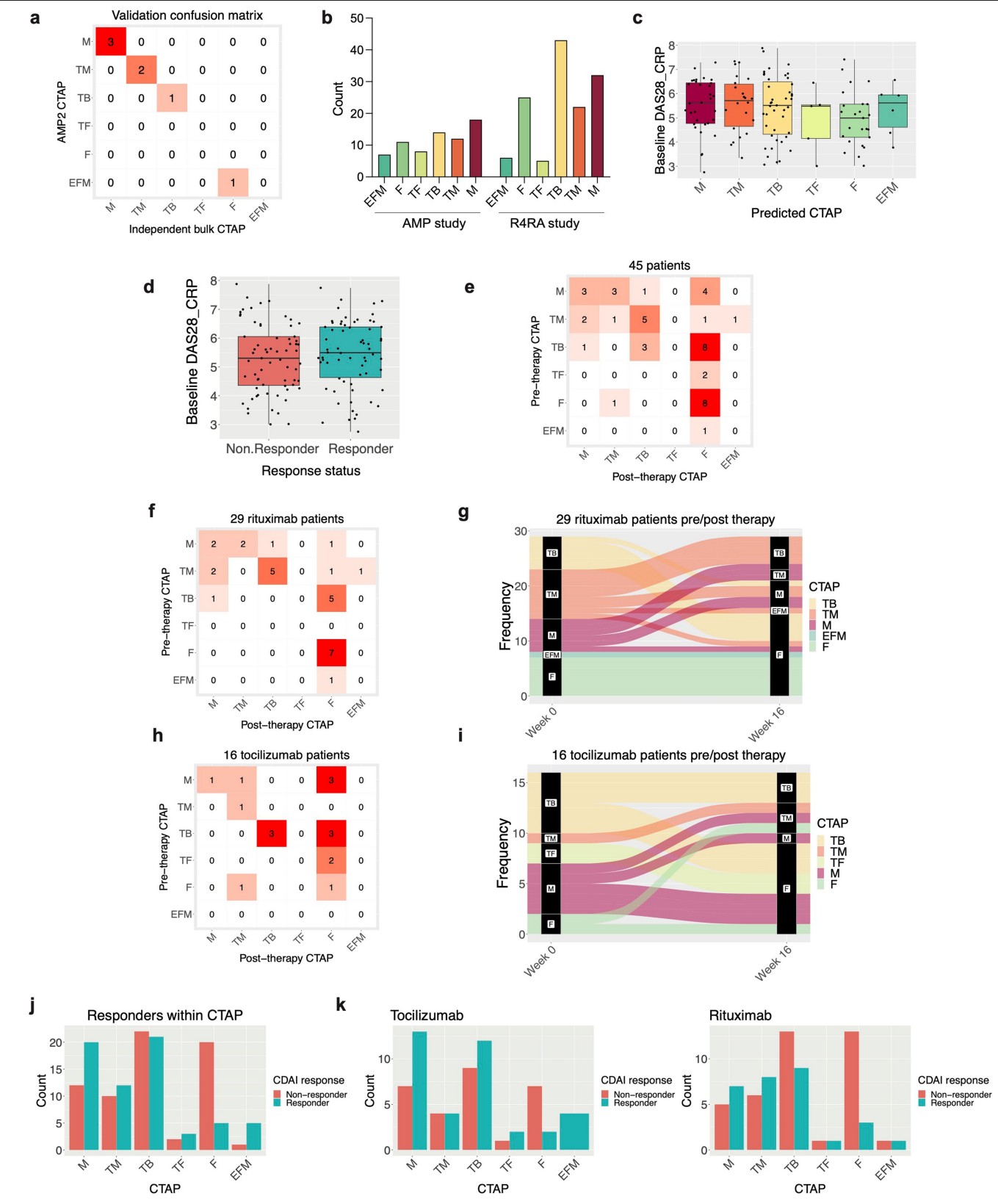

**Extended Data Fig. 10** | See next page for caption.

**Extended Data Fig. 10 | Assigning CTAP labels to bulk RNA-seq samples and clinical association analysis. a**, Confusion matrix showing CTAP assignment by the single-cell CITE-seq panel (gold standard) versus classification of synovial tissue bulk RNA-seq obtained from the same individuals (N = 7). **b**, Patients per CTAP category in the current AMP study, which enrolled a clinically diverse patient cohort, versus the published R4RA study, which restricted enrollment to patients with inadequate response to TNF inhibitor therapies. **c**, Baseline DAS28-CRP scores stratified by predicted CTAP (N = 133 patients). **d**, Baseline DAS28-CRP score stratified by clinical response status ( ≥ 50% improved CDAI after treatment) (N = 133 patients). In **c** and **d**, Points represent individuals and box plots show median (vertical bar), 25th and 75th percentiles (lower and upper bounds of the box, respectively) and 1.5 x IQR (or minimum/maximum values; end of whiskers). **e**, Confusion matrix showing predicted CTAP assignment of pre-treatment (week 0) and post-treatment (week 16) synovial tissue samples obtained from 45 patients. **f-g**, Confusion matrix and alluvial plot showing predicted CTAP assignment before and after treatment with rituximab (N = 29). **h-i**, Confusion matrix and alluvial plot showing predicted CTAP assignment before and after treatment with tocilizumab (N = 16). **j**, Graph of responder and non-responders stratified by CTAP (N = 133) among all patients in the R4RA study. **k**, Graph of responders and non-responders among patients receiving tocilizumab (left, N = 65) or rituximab (right, N = 68), stratified by CTAP.

# Reporting Summary

## Statistics

For all statistical analyses, confirm that the following items are present in the figure legend, table legend, main text, or Methods section.

| n/a | Confirmed | |
|---|---|---|
| ☐ | ☒ | The exact sample size (*n*) for each experimental group/condition, given as a discrete number and unit of measurement |
| ☐ | ☒ | A statement on whether measurements were taken from distinct samples or whether the same sample was measured repeatedly |
| ☐ | ☒ | The statistical test(s) used AND whether they are one- or two-sided *Only common tests should be described solely by name; describe more complex techniques in the Methods section.* |
| ☐ | ☒ | A description of all covariates tested |
| ☐ | ☒ | A description of any assumptions or corrections, such as tests of normality and adjustment for multiple comparisons |
| ☐ | ☒ | A full description of the statistical parameters including central tendency (e.g. means) or other basic estimates (e.g. regression coefficient) AND variation (e.g. standard deviation) or associated estimates of uncertainty (e.g. confidence intervals) |
| ☐ | ☒ | For null hypothesis testing, the test statistic (e.g. *F*, *t*, *r*) with confidence intervals, effect sizes, degrees of freedom and *P* value noted *Give P values as exact values whenever suitable.* |
| ☒ | ☐ | For Bayesian analysis, information on the choice of priors and Markov chain Monte Carlo settings |
| ☒ | ☐ | For hierarchical and complex designs, identification of the appropriate level for tests and full reporting of outcomes |
| ☐ | ☒ | Estimates of effect sizes (e.g. Cohen's *d*, Pearson's *r*), indicating how they were calculated |

*Our web collection on statistics for biologists contains articles on many of the points above.*

## Software and code

Policy information about availability of computer code

| Data collection | Clinical data were collected using REDCap v. 6.9.0 onwards. Flow cytometry data of disaggregate synovial tissue were acquired on a BD FACSAria Fusion using FACSDiva software v. 8.0.1. For functional experiments, T cell and B cell subsets were isolated using a BD FACSAria Fusion sorter, and analytic flow cytometry was performed on a BD Fortessa analyzer (B cell differentiation) or a BD Canto II analyzer (cytotoxicity assays), all using FACSDiva software. |
|---|---|
| Data analysis | Flow cytometry data were analyzed with FlowJo v10.6. Immunofluorescence microscopy images were analyzed with the Visiopharm platform (version 2022.10). Single-cell RNA-seq data were aligned and quantified with Cell Ranger (v. 3.1.0). Bulk RNA-seq data were aligned and quantified with STAR (v. 2.5.3). Other analyses were conducted with R (version 3.6 and 4.0) and Python (version 3.10). Scripts to reproduce analyses are available on GitHub (https://github.com/immunogenomics/RA_Atlas_CITEseq) and Zenodo (https://zenodo.org/record/8118599). |

For manuscripts utilizing custom algorithms or software that are central to the research but not yet described in published literature, software must be made available to editors and reviewers. We strongly encourage code deposition in a community repository (e.g. GitHub). See the Nature Portfolio guidelines for submitting code & software for further information.

## Data

Policy information about [availability of data](availability of data)

All manuscripts must include a [data availability statement](data availability statement). This statement should provide the following information, where applicable:

- Accession codes, unique identifiers, or web links for publicly available datasets
- A description of any restrictions on data availability
- For clinical datasets or third party data, please ensure that the statement adheres to our [policy](policy)

CITE-seq single-cell expression matrices and sequencing, bulk expression matrices, genotyping, and clinical data are available on Synapse (doi:10.7303/syn52297840). A cell browser website https://immunogenomics.io/ampra2/ is available to visualize our data and results. AMP Phase 1 single-cell data from Zhang*, Wei*, Slowikowski*, Rao*, Fonseka*, et al. 2019 are available on Immport (accession: SDY998). PEAC clinical trial RNA-seq data from Lewis, et al. 2019 are available on ArrayExpress (accession: E-MTAB-6141). R4RA clinical trial RNA-seq data from Rivellese*, Surace*, et al. 2022 are available on ArrayExpress (accession: E-MTAB-11611). Single-cell and bulk RNA sequencing data were aligned to GRCh38 (Ensembl 93), available as part of Cell Ranger v. 3.1.0.

# Field-specific reporting

Please select the one below that is the best fit for your research. If you are not sure, read the appropriate sections before making your selection.

☒ Life sciences      ☐ Behavioural & social sciences      ☐ Ecological, evolutionary & environmental sciences

For a reference copy of the document with all sections, see [nature.com/documents/nr-reporting-summary-flat.pdf](nature.com/documents/nr-reporting-summary-flat.pdf)

# Life sciences study design

All studies must disclose on these points even when the disclosure is negative.

| | |
|---|---|
| Sample size | Sample size (n = 82) was based on sample recruitment. No formal power calculations were performed because this was a discovery cohort with multiple advanced technical output measures. Cohort size was determined by approximate estimates of power for identifying differences between the treatment-naive, methotrexate inadequate-responder, and TNF inhibitor inadequate-responder groups. |
| Data exclusions | Biopsies that lacked synovial tissue on histological exam were excluded from the pipeline. Biopsies that yielded <400 live cells by flow cytometric sorting were excluded. Quality control of the sequencing data excluded cells with fever than 500 genes or more than 20% of UMIs from mitochondrial genes. Doublets were removed using Scrublet and a linear-discriminant analysis-based classifier. Three samples with <40% of cells passing QC were excluded from the analysis. |
| Replication | CITE-seq was conducted once per sample, with a total of 82 samples. CITE-seq data were validated by comparing cell type proportions based on CITE-seq to those calculated in the same samples with flow cytometry (n = 18). All attempts at replication were successful.<br>For the T/B cell co-culture experiment, the experiment was conducted independently on three biological replicates.<br>For the myeloid differentiation experiment, samples were randomly assigned to experimental groups (i.e., stimulus conditions). There were three biological replicates for each stimulus condition. |
| Randomization | Samples were randomized into tissue disaggregation processing batches based on treatment group and collection site. Other experiments did not have experimental groups, and there were no treatment interventions provided by this study, so randomization was not otherwise relevant. |
| Blinding | No blinding was performed in this study due to the cross-sectional nature of the study. There were no treatment interventions provided by this study. Cell clustering and CTAP categorization were performed without regard to treatment history or other clinical parameters. |

# Reporting for specific materials, systems and methods

We require information from authors about some types of materials, experimental systems and methods used in many studies. Here, indicate whether each material, system or method listed is relevant to your study. If you are not sure if a list item applies to your research, read the appropriate section before selecting a response.

## Materials & experimental systems

| n/a | Involved in the study |
|---|---|
| ☐ | ☒ Antibodies |
| ☐ | ☒ Eukaryotic cell lines |
| ☒ | ☐ Palaeontology and archaeology |
| ☒ | ☐ Animals and other organisms |
| ☐ | ☒ Human research participants |
| ☒ | ☐ Clinical data |
| ☒ | ☐ Dual use research of concern |

## Methods

| n/a | Involved in the study |
|---|---|
| ☒ | ☐ ChIP-seq |
| ☐ | ☒ Flow cytometry |
| ☒ | ☐ MRI-based neuroimaging |

# Antibodies

| | |
|---|---|
| Antibodies used | An anti-CD235 antibody (1:100, clone 11E4B-7-6 (KC16), 1IM2211U, Beckman Coulter) was included in live cell sorting to exclude red blood cells.

Antibodies used for flow cytometry are listed with clone, dilution and catalog number (as in Supplementary Table 10): CD3 (UCHT1, 1:50, 300460), CD4 (OKT4, 1:50, 317416), CD8A (SK1, 1:100, 344732), CD11c (3.9, 1:50, 301624), CD14 (M5E2, 1:200, 301852), CD19 (HIB19, 1:50, 302240), CD27 (M-T271, 1:100, 356406), CD31 (WM59, 1:200, 303134), CD45 (HI30, 1:200, 304006), CD90 (5E10, 1:200, 328124), CD146 (P1H12, 1:200, 361004), HLA-DR (L243, 1:50, 307644), PD-1 (EH12.2H7, 1:50, 329950), all purchased from BioLegend.

CITE-seq was performed using the following TotalSeq-A antibodies from BioLegend, listed with clone and catalog numbers (barcode sequences, and dilutions in Supplementary Table 2): CD107a/LAMP-1 (H4A3, 328647); CD314/NKG2D (1D11, 320835); CD19 (HIB19, 302259); CD8a (RPA-T8, 301067); CD21 (Bu32, 354915); IgG Fc (M1310G05, 410725); CD209/DC-SIGN (9E9A8, 330119); EGFR (AY13, 352923); CD196/CCR6 (G034E3, 353437); CD1c (L161, 331539); CD309/VEGFR2 (7D4-6, 359919); CD127/IL-7Rα (A019D5, 351352); CD273/B7-DC/PD-L2 (24F.10C12, 329619); CD226/DNAM-1 (TX25, 337111); CD278/ICOS (C398.4A, 313555); CD119/IFN-γ R α chain (GIR-208, 308607); CD274/B7-H1/PD-L1 (29E.2A3, 329743); CD3 (UCHT1, 300475); CD55 (JS11, 311317); IgM (MHM-88, 314541); CD155/PVR (SKII.4, 337623); CD112/Nectin-2 (TX31, 337417); CD4 (SK3, 344649); CD11c (S-HCL-3, 371519); CD34 (581, 343537); CD90/Thy1 (5E10, 328135); CD45RA (HI100, 304157); CD16 (3G8, 302061); CD45RO (UCHL1, 304255); CD20 (2H7, 302359); Podoplanin (NC-08, 337019); CD140a/PDGFRα (16A1, 323509); CD146 (P1H12, 361017); CD195/CCR5 (J418F1, 359135); CD69 (FN50, 310947); CD161 (HP-3G10, 339945); HLA-DR (L243, 307659); CD64 (10.1, 305037); CD24 (ML5, 311137); CD192/CCR2 (K036C2, 357229); CD163 (GHI/61, 333635); CD44 (IM7, 103045); CD141/Thrombomodulin (M80, 344121); CD27 (LG.3A10, 124235); CD206/MMR (15-2, 321143); Folate Receptor β/FR-β (94b/FOLR2, 391707); CD45 (2D1, 368543); CD31 (WM59, 303137); CD11b (ICRF44, 301353); CD68 (Y1/82A, custom conjugate); CD38 (HIT2, 303541); CD144/VE-Cadherin (BV9, 348517); CD304/Neuropilin-1 (12C2, 354525); CD86 (IT2.2, 305443); CD279/PD-1 (EH12.2H7, 329955); CX3CR1 (K0124E1, 355709); CD56/NCAM (QA17A16, custom barcode); CD14 (63D3, custom barcode). Antibodies against CD107a (LAMP-1), CD314 (NKG2D), CD19, CD8a, CD21, IgG Fc, CD209 (DC-SIGN), EGFR, CD196 (CCR6), CD1c, CD309 (VEGFR2), CD127 (IL-7Rα), CD273 (B7-DC, PD-L2), CD226 (DNAM-1), CD278 (ICOS), CD119 (IFN-γ R α chain), CD274 (B7-H1, PD-L1), CD3, CD55, IgM were used at a dilution of 1:250 (0.2 ug per 100 uL staining reaction), whereas the remaining antibodies were used at a dilution of 1:50 (1 ug per 100 uL staining reaction).

Antibodies used for immunofluorescence microscopy studies include CD3 (1:100, Clone M-20, sc-1127, Santa Cruz Biotechnology), CD138 (1:50, PA5-32305, Thermo Fisher Scientific), and CD20 (1:50, Clone L26, GTX29475, GeneTex), CLIC5 (1:50, clone 1E6, SAB1402589, Sigma), CD68 (1:50, clone 514H12, CD68-L-CE, Leica) CD3 (1:25, clone LN10, CD3-565-L-CE, Leica), HLA-DR (1:200, clone EPR3692, ab92511, Abcam), CD34 (1:100, clone QBEnd/10, END-L-CE, Leica) and CD90 (1:200, clone D3V8A, 13801, Cell Signaling Technology). All antibodies are listed in the following format (dilution, clone, catalog number, company).

Antibodies used for sorting T cell subsets for the T cell functional assays include anti-CD4 APC (1:100, RPA-T4, 300537), anti-CD8A BV711 (1:100, RPA-T8, 301044), anti-CD3 APC-Cy7 (1:100, OKT3, 317342), anti-CD14 FITC (1:100, HCD14, 325604), anti-CD45RA BV605 (1:100, HI100, 304134), anti-CCR7 PE-Cy7 (1:100, G043H7, 353226) and anti-PD-1 BV421 (1:100, EH12.2H7, 329920), all from Biolegend. Antibodies used for sorting memory B cells for T cell functional assays include anti-CD19 PE (1:100, HIB19, 302208), anti-CD27 BV421 (1:100, M-T271, 356418), anti-CD3 FITC (1:100, OKT3, 317306) and anti-CD14 APC (1:100, HCD14, 325608) all from Biolegend. Antibodies used to identify B cell subsets at the conclusion of the T-B cell co-cultures include anti-CD3 FITC (1:100, OKT3, 317306), anti-CD20 BV605 (1:100, 2H7, 302334), anti-CD19 APC-Cy7 (1:100, HIB19, 302218), anti-CD27 PE-Cy7 (1:100, M-T271, 356412), anti-CD38 BV785 (1:100, HIT2, 303530), anti-CD11c PE (1:50, Bu15, 337206), and anti-CD21 PerCP-Cy5.5 (1:100, Bu32, 354908), all from Biolegend. Cytotoxicity assays used anti-CD3 antibodies (OKT3, 50 ug/mL, BioXcell) as well as Annexin V (Biolegend). |
| Validation | All antibodies are commercially available and validated for flow cytometry, microscopy, functional assays, or CITE-seq of human cells as stated in the manufacturer's product information, quoted below:
Beckman Coulter flow cytometry: Beckman Coulter tests each lot for consistent performance, as verified on the Certificate of Analysis that accompanies each antibody.
BioLegend flow cytometry: Each lot of this antibody is quality control tested by immunofluorescent staining with flow cytometric analysis.
BioLegend TotalSeq-A: All lots are tested by flow cytometry to make sure they stain the expected cell population and that oligos are attached to the antibodies. This process has been validated by comparison with a traditional two-step flow cytometry staining as shown.
Leica immunofluorescence: Leica performs extensive staining experiments using a diversity of human normal and abnormal tissues to validate their antibodies. The results of these staining QC experiments are described in each Product Detail sheet.
Santa Cruz Biotechnology immunofluorescence: M-20 is a polyclonal goat anti-human CD3 that has been cited in 56 publications dating back over 20 years.
Thermo Fisher Scientific immunofluorescence: PA5-32305 is a polyclonal rabbit anti-human CD138 antibody. It was purified with antigen affinity chromatography and validated by the vendor by staining of human tonsil tissue.
GeneTex immunofluorescence: To optimize the performance of our reagents, we employ various analytic validation strategies to ensure both consistent quality and specificity. These modalities are in line with guidelines described by the International Working Group on Antibody Validation (IWGAV) and have become fundamental components of our quality assurance process:
- KO/KD Validation
- Comparable Abs
- IP/MS Analysis
- Orthogonal Validation
- Protein Overexpression
Sigma immunofluorescence: Clone 1E6 is a mouse monoclonal antibody against GST-tagged human CLIC5. According to the vendor, it is specific for the immunogen by Western blot. According to Novus Bio, which sells the same clone, it is also specific for recombinant CLIC5 without the GST tag by ELISA and Western. The vendor has also performed validation staining of human placenta.
Abcam immunofluorescence: Antibody specificity is confirmed by looking at cells that either do or do not express the target protein |

within the same tissue. Initially, our scientists will review the available literature to determine the best cell lines and tissues to use for validation. We then check the protein expression by IHC/ICC to see if it has the expected cellular localization. If the localization of the signal is as expected, this antibody will pass and is considered suitable for use in IHC/ICC. We use a variety of methods, including staining multi-normal human tissue microarrays (TMAs), multi-tumor human TMAs, and rat or mouse TMAs during antibody development. These high-throughput arrays allow us to check many tissues at the same time, providing uniformly as all tissues are exposed to the exact same conditions.

CST immunofluorescence: All CST™ antibodies that are approved for use in immunofluorescent assays have undergone a rigorous validation process. Validation steps include:
- Cell lines or tissues with known target expression levels are used to verify specificity.
- Appropriate cell lines and tissues are used to verify subcellular localization.
- Antibody performance is assessed on appropriate tissues.
- Cells are subjected to phosphatase treatment to verify phospho-specificity. Target specificity is also verified with the use of known knockout or null cell lines.
- Cells are subjected to siRNA treatment or over-expression of the target protein to verify target specificity.
- Activation state specification, target expression, and translocation are examined using ligands or inhibitors to modulate pathway activity.
- Requirement of threshold signal-to-noise ratio in antibody:isotype comparison and minimum fold-induction for phospho-specific antibodies ensures the greatest possible sensitivity.
- Fixation and permeabilization conditions are optimized; alternative protocols are recommended if necessary.
Stringent testing ensures lot-to-lot consistency.

# Eukaryotic cell lines

Policy information about cell lines

| | |
|---|---|
| Cell line source(s) | CD32-expressing murine fibroblast L cells were kindly gifted to Deepak Rao by Megan Levings (PMID: 36470208) |
| Authentication | The L cells were not recently authenticated. |
| Mycoplasma contamination | L cells were not tested for mycoplasma contamination. |
| Commonly misidentified lines (See ICLAC register) | No commonly misidentified cell lines were used in this study as L cells are not on the list of commonly misidentified cell lines. |

# Human research participants

Policy information about studies involving human research participants

| | |
|---|---|
| Population characteristics | Male and female patients with rheumatoid arthritis according to the ACR 2020 Rheumatoid Arthritis classification criteria. The patients were recruited into three different cohorts: treatment-naive patients (n=28) early in their disease course (mean 2.64 years), methotrexate-inadequate (MTX) responders (n=27), and anti-TNF agent inadequate responders (n=15). The patients were similar in age, sex, disease activity, and other clinical parameters across the three treatment groups. In addition, nine patients with osteoarthritis were recruited. Additional population characteristics detailed in Supplementary Table 1. |
| Recruitment | Participants were recruited by physician referral from 13 clinical sites across the United States and 2 sites in the United Kingdom. Only patients with active disease were recruited. Recruitment occurred mainly at academic medical centers, which may be more likely to see complex cases. Different sites used different techniques for joint biopsies, which may introduce bias. Recruitment site and biopsy method are addressed as potential confounders in the paper. |
| Ethics oversight | The study was performed in accordance with protocols approved by the Institutional Review Board at Stanford University (Protocol ID: 33561). All clinical and experimental sites obtained approval for this study from their Institutional Review Boards. |

Note that full information on the approval of the study protocol must also be provided in the manuscript.

# Flow Cytometry

## Plots

Confirm that:

☒ The axis labels state the marker and fluorochrome used (e.g. CD4-FITC).

☒ The axis scales are clearly visible. Include numbers along axes only for bottom left plot of group (a 'group' is an analysis of identical markers).

☒ All plots are contour plots with outliers or pseudocolor plots.

☒ A numerical value for number of cells or percentage (with statistics) is provided.

## Methodology

| | |
|---|---|
| Sample preparation | Cryopreserved synovial tissue fragments were disaggregated, and live cells were obtained by cell sorting (CD235a- live-dead dye-). The first 60,000 cells were used for CITE-seq studies. The next 50,000 sorted live cells were used for flow cytometry. |

| | |
|---|---|
| Instrument | A BD FACSAria Fusion cell sorter was used for all sorting and analytic flow cytometry except for the outcome analytic flow cytometry of the T-B cell co-culture assay, which was performed on a BD Fortessa analyzer. |
| Software | Data were collected using FACSDiva software 8.0.1 and analysed using FlowJo v10.6. |
| Cell population abundance | During the initial sort of live cells for CITE-seq, live cells represented a mean of 59.2% and median of 61.6% of cells (S.D. 17.9%). Cell populations of interest in the other flow panels ranged from <1% to >50%, depending on the population and sample. |
| Gating strategy | Synovial cell populations were gated as follows: A very large FSC vs SSC gate designed to capture small lymphocytes as well as large fibroblasts and macrophages. After singlet gating, dead cells and red blood cells were gated out using fixable viability dye and anti-CD235a antibodies pooled into the same channel. Cell populations were identified as follows: CD45+CD3-CD14-CD19+ (B cells), CD45-CD31-CD146- (fibroblasts), CD45+CD3-CD14+ (myeloid), CD45+CD3+CD14- (T cells). Gating of these populations is shown in Supplementary Figure 1G.<br>For functional T cell assays, T cells were sorted from live cells (negative for LIVE/DEAD Fixable Aqua Dead Cell Stain) as follows: CD14-CD3+CD4+CD8-CD45RA-PD-1hi (TPH+TFH), CD14-CD3+CD4+CD8-CD45RA-PD-1- (PD-1- Memory CD4), CD14-CD3+CD4-CD8+CD45RA- (Memory CD8), CD14-CD3+CD4-CD8+CD45RA+CCR7- (TEMRA CD8). For these assays, memory B cells were sorted from live cells (negative for LIVE/DEAD Fixable Aqua Dead Cell Stain) as follows: CD19+CD27+CD3-CD14-. At the conclusion of the T-B co-culture, B cell subsets were identified as follows: CD27hi CD38hi CD19+ (plasmablasts) and CD11c+ CD21- CD19+ (ABCs). In the cytotoxicity assay, dead cells were identified as Annexin V+.  Representative gating of the conclusion of the T-B co-culture experiment is shown in Figure 3d. |

☒ Tick this box to confirm that a figure exemplifying the gating strategy is provided in the Supplementary Information.

nature portfolio | reporting summary

March 2021

5