## [Peer Review File · Nature]

Manuscript Title: Deconstruction of rheumatoid arthritis synovium defines inflammatory subtypes

Reviewer Comments & Author Rebuttals

Reviewer Reports on the Initial Version:

Referees' comments:

Referee #1 (Remarks to the Author):

This paper aims to deconstruct the cell states and pathways that characterize pathogenic heterogeneity in RA.

The hypothesis is that specific pathways are present in different patients and that the presence of these pathways determines disease activity. This hypothesis is not tested

This hypothesis is not backed by epidemiological data e.g. the phenotype of a responder/non-responder to anti-TNF is not a stable phenotype. I miss the epidemiological justification for such an hypothesis.

If I look at the distribution patterns in TNF-blocking agents (Clinical responses to tumor necrosis factor alpha antagonists do not show a bimodal distribution: data from the Stockholm tumor necrosis factor alpha follow up registry. van Vollenhoven RF, Klareskog L. Arthritis Rheum. 2003 Jun;48(6):1500-3), I wonder whether there are epidemiological data to support the underlying hypothesis of this paper.

The paper then describes an atlas of the infiltrates studied. The considerations are written below but at the end of the day such an atlas is useful to have, but we don't learn much about the biological meaning. The paper would gain much impact if the authors would be able to generate such data By elegantly analysing surface expression and transcriptome the authors arrive at six different synovial cell-type abundance phenotypes. In the preliminary data of 3 repeated biopsies they assume that these CTAP are stable

From these 6 CTAPs, they subsequently determined the individual cell phenotypes and were able to define 77 cell states.

The type of cells are then presented in an atlas way and make sense to be present in RA.

This is a relatively descriptive study. In figure 5 then CTAPs are correlated with clinical features.

It is known that histology between ACPA-positive and ACPA-negative differs (Differences in synovial tissue infiltrates between anti-cyclic citrullinated peptide-positive rheumatoid arthritis and anti-cyclic citrullinated peptide-negative rheumatoid arthritis. van Oosterhout M, Bajema I, Levarht EW, Toes RE, Huizinga TW, van Laar JM. Arthritis Rheum. 2008 Jan;58(1):53-60.) so I expected to see differences there and indeed this is shown in figure 5B.

It is a bit disappointing that these data are driven by only 9 patients (supplementary table 10, CTAP-M with 50% ACPA negative's. So the statement on line 482 is a little bit too strong.

Then fig 6 is not very convincing. Hardly or not effective drugs like IFN and IL1 are lumped with very effective drugs like TNF or IL6.

Then (line 439-441) to authors state "we expect that CTAPs can be used to systematically query RA heterogeneity across technologies to improve the granularity of clinical studies and trials and

potentially to guide therapy selection.” To me I donot see what the data are to support this statement.

The section of line 489-499 is also a bit too speculative

Referee #2 (Remarks to the Author):

This is an interesting and comprehensive piece of work from arguably the world’s premier collaboration in this domain. It is a necessarily statistical and bioinformatic analysis of a massive dataset. There will inevitably be further complexity beneath the 2/3 cell CTAP definitions proposed.

The work provides an original classification suggesting that RA synovial pathology can be distributed between six ‘high-level’ cell type abundance phenotypes or CTAPs, and that the pathological architecture within these CTAPs is relatively consistent. By further characterisation of the cell types associated with CTAPs and their transcriptional profiles, the authors provide suggestions as to their potential relevance to a precision medicine approach towards treating RA. They also provide some preliminary data suggesting that synovial CTAPs may be predictable from peripheral blood flow cytometry profiles. It is interesting, although perhaps not unexpected, that particular cell types tend to associate with one another within particular CTAPs; perhaps less expected is that, at a deeper level, particular subsets of those cell types tend to co-aggregate. The work will also provide an excellent dataset for workers within this field to study.

An over-riding assumption of the work’s interpretation is that relative cell abundance implies importance. Whilst this feels intuitively possible, the work may overlook critical cell types and their interactions. For example, a CTAP is defined when a cell type is more abundant than its average per cent frequency over all biopsies. As the authors state, however, in numerical terms a cell type can, at times, be more abundant in a particular CTAP than the cells that define it (because it still has a lower per cent frequency than its average across all biopsies). Further, the authors could have decided to adopt an alternative approach, for example classifying biopsies primarily on the basis of immune mediators and then tracing mediator dominance back to cell type. This perhaps would make less biological sense but potentially have more impact from a therapeutic perspective. Nonetheless, the work provides a potentially useful classification and, as above, the dataset is available for other researchers to interrogate and manipulate.

It is perhaps surprising that there are only 6 CTAPs in such a genetically complex condition, as well as the relative constancy of cell subsets within each CTAP. As suggested above, a relatively minor population could still have an important influence over the biological behaviour of tissue if it has important interactions with a more abundant cell type. For example, suppl Fig 6E suggests that, whilst T cell neighborhoods enriched in CTAP-TF (permutation $p=0.036$) mainly comprised cytotoxic CD4+GZMB+ (T-12) and CD8+GZMB+ (T15), these subsets still form a relatively small proportion of overall T-cells. As another example, the work confirms that TFH and PFH associate with ABC and memory B-cells, mainly within the T-B CTAP (Fig 3A, C). IgM+ plasma cells (B-6), plasmablasts (B-7), and ABCs (B-5) were also positively associated with aggregates (permutation $p=0.007$) (suppl Fig

11B, Figure 5A) and yet IgM+ plasma cells (B-6) and plasmablasts (B-7) were NOT associated with T-B CTAP (Fig 3, suppl Fig 7G). In fact Fig 3C suggests that IgM+ plasma cells were relatively depleted in this CTAP. A comparison of Fig 3C and 11B tells us that aggregates do not map accurately to the T-B CTAP, which is perfectly fine but illustrates the complexity of the data and care needed in its interpretation. As mentioned above, there will inevitably be further complexity beneath the 2/3 cell CTAP definitions proposed.

An important element of this work overall are the potential links between CTAPs and other clinical and histological parameters. This information is provided in suppl Fig 12 which, arguably, should be part of the main text. A few features of this Table are worth highlighting. Firstly, it is surprising that Krenn inflammation and US doppler associations do not have more similar associations, particularly with myeloid cell types, given that we believe US Doppler is a marker of synovial inflammation. Secondly, most statistically significant CTAP associations reflect histological parameters which, in some ways, feels rather tautologous, although CCP associations (Fig 12C) are of interest. Having said that, seropositivity overall does not associate with one or other CTAP. Notwithstanding the lack of statistical significance, the presence of the TF CTAP in seropositive but not seronegative patients is perhaps surprising and, again, may reflect deeper heterogeneity within CTAPs. The authors comment: "We argue that CTAPs from biopsies offer independent information from what physician assessments offer". This is definitely the case (eg Krenn associations) but how clinically valuable this additional information will be awaits further investigation.

There is a clear effect of RA duration and/or treatment with CTAP, with longer lasting/more refractory RA clustering within the EFM CTAP, and a disease duration ranging from 2.4 (TM) to 12 years (EFM) across CTAPs (Suppl Table 10). This also links to Fig 12L. More than 70% of TNF inadequate responders fall within the EFM CTAP and, whilst this association is not significant, it may have been appropriate to consider treatment and disease duration as potential confounders. Reinforcing this feeling are data in suppl Table 1 (demographics), where Krenn cell density – which associates with CTAPs - reduces with disease duration/refractoriness. The authors address this in the text: "patients in CTAP-EFM tended to be older and have longer-standing RA than patients in other CTAPs and were mostly TNFi-inadequate responders although these associations were not statistically significant". Given the long-term ambition of using CTAPs as guides to precision medicine, perhaps at least a sensitivity analysis of the data could have been included treating disease duration and treatment as confounders.

Figure 12K is also of interest in terms of precision medicine. The authors comment: "Since CTAPs appear to correlate with known drug targets (Figure 6D) and can be assigned even with flow cytometry, we expect that CTAPs can be used to systematically query RA heterogeneity across technologies to improve the granularity of clinical studies and trials and potentially to guide therapy selection". This is a critical message and it would be of interest to understand how the authors plan to move this work forwards from a precision medicine perspective.

OA tissue was used as a control. Whilst the sample number was small (N=9), is there any message to take home from the OA data regarding consistency vs heterogeneity vs CTAP similarities? Were these samples sufficiently homogeneous to act as a suitable control group?

Three patients were biopsied twice and, whilst the authors comment that, overall, CTAPs appear constant over relatively short timespans (98 to 427 days), this is clearly a critical aspect that requires further work. The cell-type composition of repeat biopsies was similar to the initial biopsy but not identical (mean Mahalanobis distance=1.55, permutation 430 $p=0.073$) (Supplementary Figure 14A-B).

Minor questions/comments:

Why were surface proteins less informative for non-lymphocytes? This is important and perhaps deserves further emphasis/discussion because only transcriptional profiling was used for non-lymphocyte clustering, adding heterogeneity to analyses.

“CD1c 206 + MZ-like B cells (B-3) and other non-plasma B cells were high producers of IL6 and TNF” – I believe the data (suppl 7D) support the former (IL6) but not the latter.

“T cell neighborhoods enriched in CTAP-TF (permutation $p=0.036$) mainly consisted of cytotoxic CD4+GNLY+ (T-12) and CD8+GZMB+ cells (T-15)”. CD4 NAïVE T-cells also appear abundant but are not mentioned here (Fig 3A).

Referee #3 (Remarks to the Author):

This study represents a multi-site ($n=15$) international collaboration involving a strong number of donors (70 RA, 9 osteoarthritis) for which single cell analysis was performed (mostly by biopsy, a few cases of synovectomies) of synovial tissues. 314K cells were analyzed, averaging about 3800 cells per sample. CITE-Seq was performed with 58 protein markers, with application and inference primarily to adaptive immune cells. Ascertainment for the RA patients was by 3 treatment classes: treatment naive, and failing methotrexate or failing anti-TNF. A major result is the classification of RA into 6 CTAPs (cell-type abundance phenotypes), based upon the major cell subsets (T, B, myeloid, stromal, endothelial, NK, B/plasma cells), as opposed to finer-scale clusters. Correlations to histologic measures is performed, with the greatest correlations being with cell density.

This study represents the likely largest cohort ever to be collected, and with full data-sharing, represent a treasure-trove of data to be mined by the immunology and rheumatoid arthritis communities. However, the study does not fulfill the goals stated in the introduction, namely, to define patient subsets predictive of therapeutic responses. While some replicate analyses of selected patients is reported, implying consistency of traits/CTAPs, with the relatively modest sample sizes for clinical correlations, it is unclear whether these subtypes represent basic pathophysiologic subsets, genetic subsets, serologic subsets, or whether these subsets are primarily driven by inflammatory, clinical or other stochastic factors. While the analyses are meticulously reported and cutting-edge, as currently presented, the results (which many similar papers do) often comes across as a laundry-list, lacking clear clinical (recognizing the modest sample sizes with which to do this with) or biologic advances.

General comments

- Clinical significance of subtypes: while the goal of treatment prediction in this manuscript may be out of reach, is there sufficient prospective treatment and response data on this cohort to provide any observational insight? Given that the biopsies were taken at a time of moderate-severe activity, presumably, treatment escalation would commonly have occurred shortly thereafter.
- It is disappointing, but perhaps not surprising that the CTAPs are not associated with treatment class, nor many clinical (including histologic) variables. One exception to this is CCP antibody correlations to TB subset. Given the substantial extant data on TNF and methotrexate (pathways, GEO data), did the investigators attempt to perform pathway or cellular analyses. I wonder if an excessive reliance on the 6 CTAP classes may be limiting discovery potential.
- This effort may be one of the larger and more comprehensive efforts with CITE-Seq, but the results/impact is relatively modest, being confined to classification of adaptive immune subsets. This substantially reflects the much lower number of measures (58) and specific marker selection. I don't believe the correct table was uploaded for Supplementary Table 2. STable 8 does summarize the CITE-Seq results, but restricted by cell type. Might be more informative to include antibody readings for all cells for all 58 markers.
- Given the key roles for cell differentiation, proliferation and trafficking in RA, general application of RNA velocity approaches may provide more dynamic insight into the CTAPs, and possibly even some form of validation of the value of these categories. It would undoubtedly provide insight in the fine-scale clustering (e.g. infiltrating monocytes, tissue macrophages). This might nicely complement the cellular-based CNA analysis
- Were genetics performed on this cohort? If so, is there any evidence that disease subtypes (via cell-specific epigenetics) map to relevant genetics scores?
- This group previously published an elegant spatial mapping focused on stromal cells and the NOTCH pathway. More generally, could spatial approaches validate the 6 categories better? Especially with respect to the inferences nominally inferred to by CNA? The histology scores used clinically may lack the multi-dimensionality of research spatial approaches (multi-dimensional immunofluorescence, spatial transcriptomics, RNAScope). In particular, given the high fraction of highly heterogeneous endothelial cells, spatial approaches may be particularly illuminating.
- By qualitative inspection (Fig 1G), osteoarthritis appears to be similar to EFM RA, yet on finer subclustering, the gene expression patterns are quite distinct. Does this undermine the RA CTAP proposed in any way? Is the diagnostic distinction between OA and RA so marked and definitive that this does not undermine at all the RA CTAP definitions?
- Did the investigators test for batch effects across sites so assure generalizability? The central processing likely substantially mitigates this, but differences between pinch biopsies vs. synovectomies, vs. samples attained at arthroplasty are likely substantial confounders. Joint location?
- Fig 5A involves a large number of comparisons (albeit likely accounted for with permutation testing) against rather general histologic measures (aggregation, density), with only very general, well-known biologic results (T cells correlate with aggregate scores). Much of Fig 5 could be relegated to supplementary materials.
- I am concerned that at least parts of Fig 6 could be misleading. The absence of cytokine expression by the 10X system does not mean that it is not expressed, given its well-known sparse nature. On the contrary, it is quite possible that small amounts of cytokine secretion from selected cells, in close proximity, may be the actually therapeutically critical pathway. 'Line 395-6: Most cytokines produced

by only one cell type'--this is likely biologically not true, to a substantial amount. Spatial expression approaches, as opposed to complete reliance on CNA may provided added support for specific claims. Also consider application of CellPhone db or other similar programs to assist with a more systematic evaluation of source-target cross-talk.

Author Rebuttals to Initial Comments:

We thank the three Reviewers for their helpful comments and suggestions. We were really encouraged by the Reviewers' praise for the scope of this study and its utility to the larger research community. For example, Reviewer #3 states "This study represents the likely largest cohort ever to be collected, and with full data-sharing, represent a treasure-trove of data to be mined by the immunology and rheumatoid arthritis communities."

While the Reviewers noted the tremendous value of the data produced by this project, they also requested additional evidence of the clinical and mechanistic value of the cell type abundance phenotype (CTAP) classification system, which classifies RA synovial tissue into inflammatory phenotypes. Inspired by their feedback, we have added four major new components to the manuscript (including analyses and data). Together these new components markedly strengthen the manuscript and directly address the Reviewers' questions regarding the clinical utility of the CTAP system. Here is a summary of the findings from those analyses. Details are described in the point-by-point response.

- 1) **RA inflammatory synovial tissue phenotypes are dynamic and predict treatment response.** This important analysis demonstrates the clinical significance and future potential of the CTAP classification framework. We recognize that bulk RNA-seq data on synovial tissue biopsies are publicly available and often easier to obtain than single-cell RNAseq in a clinical trial setting. Thus, we developed a classification algorithm that enables investigators to use bulk RNA-seq of RA synovial tissues to infer inflammatory phenotypes that we defined using single-cell data. After confirming the accuracy of this algorithm, we applied it to data from a published clinical trial of rituximab versus tocilizumab for the treatment of RA (R4RA study, Rivellese, Surace, et al, *Nature Medicine* 2022, PMID: 35589854). This analysis yielded two notable findings. First, synovial tissue phenotype prior to treatment is associated with clinical response with these two therapies. Specifically, individuals with CTAP-F were very unlikely to respond to either of these immunomodulatory treatments. Second, inflammatory phenotypes can change with treatment. This analysis strongly supports the potential utility of inflammatory phenotypes in clinical practice in RA and the opportunities for new insights into RA pathophysiology and clinical response if applied in future studies. (See **Reviewer #1, Comment #6**)
- 2) **Functional cell-cell interaction and immune mediator assays validate the biologically distinct environments underlying inflammatory phenotypes.** Using functional assays, we have demonstrated that cell states and soluble factors that characterize specific CTAPs have the predicted effects on myeloid and B-cell phenotypes. In one set of experiments, we examined CD4⁺ and CD8⁺ T cell populations, such as T peripheral and T follicular helper cells (Tph and Tfh cells) and granzyme K-enriched memory CD8 T cells. We have assayed their ability to induce the differentiation of plasmablasts and ABC B cell populations, both of which are highly enriched in CTAP-TB. As predicted, we found that Tph and Tfh cells, which are also enriched in CTAP-TB, efficiently induce the differentiation of these specialized B cell subsets and maintained B cell numbers overall. CD8⁺ T cells subsets associated with CTAP-TF and CTAP-M did not. In a separate set of experiments, we co-cultured blood-derived monocytes with cells and

soluble factors, such as CD8⁺ T cells or TGF β , which characterize specific inflammatory phenotypes. We found that activated CD8⁺ T cells induced a *STAT1*⁺*CXCL10*⁺ macrophage phenotype that is enriched in CTAP-TM, while TGF β and fibroblasts instead promoted differentiation of *MERTK*⁺*HBEGF*⁺ myeloid cells, which are enriched in CTAP-M. These findings not only validate our hypotheses regarding the effects of molecular and cellular environments on synovial cells but also provide a foundation for further study of cellular and molecular interactions in RA (See Reviewer #1, Comment #2)

- 3) **Many RA risk genes are preferentially expressed in CTAP-enriched populations.** We wanted to demonstrate how our data might be used to interpret RA genetic data. We examined genes implicated in rheumatoid arthritis GWAS, many of whose cell-state affinity is unknown. We observed that many of them are often preferentially expressed by cell states that are expanded within specific CTAPs. For example, RA risk allele rs4341355 regulates *IL6R* expression, and we found that *IL6R* is preferentially expressed by Tph populations, which are expanded in CTAP-TB. This finding helps to focus our search for the biological role of RA risk alleles to specific cell states and CTAPs. In this case, for example, IL6 has been shown to increase proliferation of Tph cells and prolongs the production of CXCL13 by CD4⁺ T cells (PMIDs 25681343, 24022618, 35911690). As another example, an RA risk allele is in *PRKCH*, encoding Protein Kinase C (PKC)-eta, a signaling molecule that mediates VEGF-induced endothelial-cell differentiation. *PRKCH* is highly expressed in endothelial cell states expanded in CTAP-M, a CTAP with strong *VEGFA* and VEGF receptor (*KDR*, *FLT1*) expression among myeloid cells and endothelial cells, respectively. These findings demonstrate the potential for these data to shed insight on genetic mechanisms in RA. (See Reviewer #2, Comment #6)

- 4) **Microscopy provides a preliminary look at the architecture of inflammatory phenotypes.** We have added immunofluorescence microscopy imaging of major cell types in synovial tissue fragments from the same cohort of patients as our single-cell CITE-seq study. We imaged 150 synovial tissue fragments from 36 patients stained for eight markers to capture basic synovial cell types. These images provide a qualitative look at the spatial arrangement of major cell types in the CTAPs relative to aggregates and lining/sublining structures. These results suggest that CTAPs are not only different in their cell type frequencies, but may also be morphologically different. In addition, quantitative analysis of the synovial tissue histology demonstrates that the cell frequencies in the most cellular fragments are generally in agreement with the cell types seen in CITE-seq. Together, these new microscopy data reinforce the utility and reliability of synovial tissue biopsies to provide information regarding inflammation in RA and illustrate the potential morphological differences that the CTAPs might represent (See Reviewer #3, Comment #6)

We have also completed many other analyses, which are listed below in a point-by-point response to the Reviewers.

Referee #1, Comment #1:

This paper aims to deconstruct the cell states and pathways that characterize pathogenic heterogeneity in RA. The hypothesis is that specific pathways are present in different patients and that the presence of these pathways determines disease activity. This hypothesis is not tested. This hypothesis is not backed by epidemiological data e.g. the phenotype of a responder/non-responder to anti-TNF is not a stable phenotype. I miss the epidemiological justification for such an hypothesis. If I look at the distribution patterns in TNF-blocking agents (Clinical responses to tumor necrosis factor alpha antagonists do not show a bimodal distribution: data from the Stockholm tumor necrosis factor alpha follow up registry. van Vollenhoven RF, Klareskog L. *Arthritis Rheum.* 2003 Jun;48(6):1500-3), I wonder whether there are epidemiological data to support the underlying hypothesis of this paper.

We thank the reviewer for challenging us to clarify our hypothesis and to better define how our study fits into the context of the RA clinical field. To be clear, our hypothesis is that specific cell states and pathways are most active in different patient subgroups, as shown in **Figures 5 and 6**. Disease activity measures, such as DAS28 or CDAI, are complex composite assessments that reflect pain pathways as well as inflammatory pathways, so biological assessments of their clinical correlates are complex and multifactorial (McWilliams, *BMC Rheumatol*, 2018).

We agree with the reviewer that the relationship between CTAPs, disease activity, and treatment response is a clinically pressing question. In this study, we targeted our enrollment to RA patients with high disease activity. Consequently, our study is not well-powered to test associations between CTAPs in patients and disease activity. As we indicate in **new Extended Data Fig. 8A**, there is no evidence of association between CTAPs and DAS28-CRP (ANOVA $p = 0.53$) or CDAI (ANOVA $p = 0.28$) in this study. An expanded study with individuals with highly variable disease activities would be better powered to investigate variable disease activity across CTAPs.

We have added the following sentence to the main text to clarify this point:

*We did not find a significant association between CTAPs and DAS28-CRP or CDAI (**Extended Data Figure 8A**), although our patient cohort is not ideal for testing such associations since it only includes patients with high disease activity.*

Inspired by the reviewers' comments, we have added new analyses to investigate the relationship between CTAP and disease activity in other studies with bulk RNA-seq data. As described below in **Reviewer #1, Comment #6**, we have developed a strategy to accurately assign CTAPs using bulk RNAseq data from synovial tissue samples. This strategy can now be applied to public bulk RNA-seq synovial tissue data sets. We applied our algorithm to the

largest RA synovial RNA-seq datasets available. Using this approach, we did not see an association between DAS28-CRP and CTAP status in baseline samples. There was also no difference in baseline DAS28-CRP between responders vs. non-responders. Furthermore, as detailed in **Reviewer #1, Comment #6**, we also used this dataset to determine that CTAPs may change over time with treatment. The clinical implications of dynamic CTAP changes will need to be investigated in future studies. However, it is possible that these changes could at least in part explain the instability of the TNF response phenotype in RA patients over time observed by van Vollenhoven et al.

We have added new text to present these new data:

*We then applied our CTAP classification algorithm to the 178 bulk RNA-seq profiles from the R4RA clinical trial comparing rituximab and tocilizumab for the treatment of RA⁹⁶. We first asked whether CTAPs were associated with disease activity as measured by DAS28-CRP in the R4RA cohort. As in our cohort (**Extended Data Figure 8A**), we did not find an association between CTAP assignment and disease activity nor between treatment response and disease activity (**Extended Data Figure 10B-C**). These findings further support our hypothesis that CTAPs reflect distinct inflammatory phenotypes driving arthritis rather than differences in clinical disease activity or stage of disease.*

Extended Data Figure 10B-C. B. Baseline DAS28-CRP3 scores stratified by predicted CTAP. **C.** Baseline DAS28-CRP3 score stratified by clinical response status ($\geq 50\%$ improved CDAI after treatment). Points represent samples. Box plots show median (vertical bar), 25th and 75th percentiles (lower and upper bounds of the box, respectively) and 1.5 x IQR (or minimum/maximum values; end of whiskers).

We have also updated the Discussion of the manuscript to incorporate these points:

While limited by small sample numbers in each group, we found that CTAPs are associated with histologic and serologic (CCP) parameters, in line with published studies¹⁰⁶ that report increased lymphocyte infiltration (suggesting CTAP-TB, -TF, or -TM) in CCP-positive synovium compared with CCP-negative synovium. Our finding that CTAP-M, and not CTAP-F or CTAP-EFM, was associated with CCP-negative status was surprising and warrants further investigation in future studies.

Applying the CTAP classification system to a larger clinical trial in RA⁹⁶, we found that CTAPs can change over time with treatment and that CTAP-F was associated with poor response. The dynamic heterogeneity of RA synovitis may explain the observation that clinical measures of

patients treated with TNF inhibitors do not fall into a bimodal distribution of responders and non-responders¹⁰⁷. We propose that CTAPs from biopsies offer independent information from what physician assessments offer. It is possible that specific CTAPs will be more likely to respond to specific therapies that preferentially target the infiltrating cell types and relevant pathways. We anticipate future longitudinal studies to investigate CTAP changes with treatment effects across a larger array of treatments.

Referee #1, Comment #2:

The paper then describes an atlas of the infiltrates studied. The considerations are written below but at the end of the day such an atlas is useful to have, but we do not learn much about the biological meaning. The paper would gain much impact if the authors would be able to generate such data.

We thank the reviewer for recognizing the utility of a large disease atlas. The major goal of our resource paper was to describe the wide spectrum of cell states and patient phenotypes in RA while also generating biological hypotheses. We have added functional studies to better elucidate the biological meaning of some of our findings investigating cell-cell interactions in T cells and myeloid cells. We also discuss how these data can be used to interpret genetic data (see **Reviewer #2, Comment #6**).

As a first step, we present new data that support the hypothesis that cell-cell communication between the cell types that characterize each CTAP drive the phenotypes of those cells. In one set of experiments, shown in **new Figure 3I-K**, we co-cultured different T cell subsets with B cells in the presence of superantigen (Staphylococcal enterotoxin B, SEB). We describe these findings in the following new text:

*We hypothesized that the finding of preferential enrichment of T_{PH} and T_{FH} cells in CTAP-TB reflected the ability of these subsets to sustain and activate B cells. To test this hypothesis, we sorted T_{PH} and T_{FH} cells and other memory $CD4^+$ T cells, as well as TEMRA $CD8^+$ T cells and $CD45RO^+$ memory $CD8^+$ T cells, which are enriched for $GzmB^+$ and $GzmK^+$ $CD8^+$ T cells, respectively²⁹. We co-cultured the sorted T cell subsets with B cells and SEB (Staphylococcal Enterotoxin B) superantigen for five days and then assessed the phenotypes of the B cells using flow cytometry (**Figure 3I-J**). T_{PH} and T_{FH} cells efficiently induced differentiation of B cells into plasmablast and ABC phenotypes (**Figure 3J**). Interestingly, non- T_{FH}/T_{PH} memory $CD4^+$ T cells were also able to induce ABC differentiation, but not plasmablasts. $CD8^+$ T cells did not induce B cell differentiation despite being functionally potent, as demonstrated by cytotoxicity assays.*

New Figure 3I-K. **I.** Schematic representation of the experimental design of the T cell functional assays. **J.** Representative flow cytometry plots showing gating of plasmablasts ($CD27^{hi} CD38^{hi} CD19^{+}$ cells), ABC B cells ($CD11c^{+} CD21^{+} CD19^{+}$ cells) and dead target cells (Annexin V^{+}). **K.** Box plots of plasmablast count (left), ABC count (center), or Annexin $^{+}$ percentage (right) stratified by co-cultured T cell subset. Points represent samples and shapes correspond to samples from the same donor ($n = 3$). Bar height represents mean, and error bars represent \pm one standard deviation.

In a separate set of experiments shown in **new Figure 4G-I**, we investigated the effect of CTAP-associated cell types and soluble factors on monocytes. We describe these findings in the following new text:

Given their highly plastic nature, we hypothesized that monocytes entering into synovial tissue are distinctly shaped by the unique network of cell types and soluble factors found in each CTAP. For the CTAPs with high myeloid cell proportions (CTAP-M and -TM), we tested this concept by exposing human blood $CD14^{+}$ monocytes to factors enriched in these tissues and then examining which CTAP-associated myeloid state these cells resembled (Figure 4G). We found that activated $CD8^{+}$ T cell factors that mark CTAP-TM induced a set of genes consistent with the $STAT1^{+} CXCL10^{+}$ macrophage state that is enriched in CTAP-TM (Figure 4H-I). Conversely, factors enriched in CTAP-M, including M-CSF, $TGF\beta$, and fibroblasts, drove monocytes towards the $MERTK^{+} HBEGF^{+}$ phenotype that is enriched in CTAP-M.

New Figure 4G-I. **G.** Schematic representation of the experimental design of the myeloid cell assays. **H.** Linear discriminant analysis classification of bulk RNA-seq obtained from myeloid cells cultured in the indicated conditions. Each condition was performed with three biological replicates, and cluster proportions in each pie chart were calculated from the mean of the posterior probability values across replicates. **I.** Heatmap showing expression of

selected CTAP-relevant genes in bulk RNA-seq of blood monocytes cultured in the indicated conditions. Columns correspond to three biological replicates for each condition, and boxes are colored by normalized gene expression.

Referee #1, Comment #3:

By elegantly analysing surface expression and transcriptome the authors arrive at six different synovial cell-type abundance phenotypes. In the preliminary data of 3 repeated biopsies they assume that these CTAP are stable. From these 6 CTAPs, they subsequently determined the individual cell phenotypes and were able to define 77 cell states. The type of cells are then presented in an atlas way and make sense to be present in RA. This is a relatively descriptive study. In figure 5 then CTAPs are correlated with clinical features.

We appreciate the reviewer's acknowledgement of our study's complexity and praise for our analysis.

Referee #1, Comment #4:

It is known that histology between ACPA-positive and ACPA-negative differs (Differences in synovial tissue infiltrates between anti-cyclic citrullinated peptide-positive rheumatoid arthritis and anti-cyclic citrullinated peptide-negative rheumatoid arthritis. van Oosterhout M, Bajema I, Levarht EW, Toes RE, Huizinga TW, van Laar JM. *Arthritis Rheum.* 2008 Jan;58(1):53-60.) so I expected to see differences there and indeed this is shown in figure 5B. It is a bit disappointing that these data are driven by only 9 patients (supplementary table 10, CTAP-M with 50% ACPA negative's. So the statement on line 482 is a little bit too strong.

We thank the reviewer for their recognition of our study's consistency with past studies. We are reassured that our observation of synovial heterogeneity between CCP-positive and -negative samples aligns with van Oosterhout *et al*, but we agree that the number of CCP-negative samples is small and restricted to one CTAP. This is a limitation and needs further examination in future cohorts. We also note that the relationship between histology and serological status may be complex. van Oosterhout, *et al*. studied patients with an average of 9 years of disease electing for therapeutic arthroscopy. It is possible that histological features may be different in early-stage disease, prior to treatment.

To supplement our previous ANOVA analysis associating CCP titer with CTAPs, we performed a logistic regression analysis of CCP status in our AMP2 CITE-seq dataset. We built a univariate logistic regression model for each CTAP, controlling for age and sex, and we tested whether the CTAP is associated with CCP positivity. We found that CTAP-M is statistically associated with CCP-negative status in our data.

We have changed original Line 482 to the text below:

While limited by small sample numbers in each group, we found that CTAPs are associated with histologic and serologic (CCP) parameters, in line with published studies¹⁰⁶ that report increased lymphocyte infiltration (suggesting CTAP-TB, -TF, or -TM) in CCP-positive synovium compared with CCP-negative synovium. Our finding that CTAP-M, and not CTAP-F or CTAP-EFM, was associated with CCP-negative status was surprising and warrants further investigation in future studies.

Referee #1, Comment #5:

Then fig 6 is not very convincing. Hardly or not effective drugs like IFN and IL1 are lumped with very effective drugs like TNF or IL6.

We agree with the reviewer that not all treatments are equally effective, and IL-1 pathway blockers have been approved for treatment, but have seen limited clinical application compared to TNF and IL-6 blockade in treating RA. By annotating RA drug targets in Figure 6D, our intention was to take an all-encompassing view with the unbiased criteria to include any cytokine or receptor targeted by FDA-approved medications for rheumatoid arthritis, even if the drugs are rarely used to treat RA in clinical practice. This included anakinra, which is an IL-1 antagonist approved by the FDA for rheumatoid arthritis. However, we agree that this approach might obfuscate the clinical relevance of these drugs. Therefore, we have removed this annotation, and furthermore, we have de-emphasized this panel by moving it to **Extended Data Figure 9**.

Referee #1, Comment #6:

Then (line 439-441) to authors state "we expect that CTAPs can be used to systematically query RA heterogeneity across technologies to improve the granularity of clinical studies and trials and potentially to guide therapy selection." To me I do not see what the data are to support this statement.

The section of line 489-499 is also a bit too speculative.

We thank the reviewer for identifying this claim as a possible overstatement. The reviewer's comments prompted us to do additional methods development and application to address this issue. We respond to these comments in three parts. First, we argue that CTAPs are generalizable to other studies and can be inferred from single-cell data, flow cytometry data, or bulk RNA-seq data. Second, we demonstrate differences in treatment response across CTAPs. Finally, we modified the language to temper our claims.

Regarding the potential to infer CTAPs from different technologies, we had previously demonstrated the ability to do this using flow cytometry in the original version of the manuscript (original **Figure 6E/new Figure 6C**). Now, we have extended this analysis to demonstrate inference from bulk RNA-seq data and provide code that allows users to do this.

We extended our work to assign bulk tissue RNA-seq profiles to CTAPs. We first used our single cell data to create pseudo-bulk gene expression profiles for each sample by summing read counts for each gene from all single cells. We then reduced the AMP2 pseudo-bulk reference and the query bulk dataset to the union of highly variable genes in each dataset. Then we used canonical correlation analysis (CCA) to align our study's CITE-seq pseudo-bulk tissue samples in a shared transcriptional space with query bulk RNA-seq samples from an external synovial tissue dataset. We inferred CTAP assignments for the external bulk dataset samples by using a 5-nearest neighbor classifier.

To confirm the accuracy of this approach, we analyzed bulk tissue RNA-seq data that was already obtained for seven samples from individuals who were included in our study. These seven samples were sequenced as part of a published dataset of bulk tissue RNA-seq collected in the setting of a clinical trial (PEAC, Lewis et al, 2019). These seven samples gave us the opportunity to assess if our CCA-based classification method was accurate. After excluding these seven samples from our reference dataset (to avoid overfitting), we applied our CTAP classification algorithm to these seven bulk RNA-seq samples. We observed that the CTAP classifications for 6/7 samples agreed with the original AMP CITE-seq CTAPs (**new Extended Data Figure 10A**, shown below). The one discrepant sample was classified as CTAP-EFM in the initial CITE-seq analysis, and CTAP-F in analysis of bulk RNA-seq; these two CTAPs are transcriptionally similar. We observed similar results in sensitivity analyses using variable numbers of nearest-neighbors samples, provided at least 3 nearest neighbors were used.

Validation confusion matrix

AMP2 CTAP	M	TM	TB	TF	F	EFM
M	3	0	0	0	0	0
TM	0	2	0	0	0	0
TB	0	0	1	0	0	0
TF	0	0	0	0	0	0
F	0	0	0	0	0	0
EFM	0	0	0	0	1	0
	M	TM	TB	TF	F	EFM
	Independent bulk CTAP					

Extended Data Figure 10A. Confusion matrix showing CTAP assignment by the single-cell CITE-seq panel (gold standard) versus classification of synovial tissue bulk RNA-seq obtained from the same individuals (N=7).

We describe these results in detail in the **Methods** section and also in new main text of the manuscript, excerpted here:

*Recent clinical trials for RA used bulk RNA-seq from intact synovial tissue to align molecular signatures with treatment outcomes to targeted therapies^{19,96}. To enable the study of CTAPs in clinical trials, we developed a method to classify CTAPs from bulk tissue RNA-seq data. We validated this method using seven patient samples that were split and analyzed by RNA-seq within the PEAC cohort study (bulk)¹⁹ and our single-cell CITE-seq analysis (single-cell). Our CTAP classification algorithm agreed with the original AMP CITE-seq CTAP classification for 6/7 individuals (**Extended Data Figure 10A**). The discrepant sample was classified as CTAP-EFM in the initial CITE-seq analysis, and CTAP-F in analysis of bulk RNA-seq; these two CTAPs are transcriptionally similar and cluster together (**Figure 1G**).*

We then performed analyses to demonstrate the utility of identifying CTAPs to improve the interpretation of clinical trials. We used our CTAP assignment approach to classify bulk tissue

RNA-seq samples from a published, independent clinical trial (R4RA, Rivellese *et al.*, 2022). We then sought to answer two independent sets of questions asked by this and other Reviewers. The R4RA dataset contains baseline pre-treatment bulk samples from 133 unique patients, with 45/133 patients also having post-treatment bulk samples.

First, we asked whether CTAPs are constant or dynamic with treatment (Reviewer #1 Comment #1). To answer this question, we focused on the 45 patients who underwent biopsies at two time points: before therapy and 16 weeks after therapy with either rituximab or tocilizumab. For each patient, we classified their bulk samples before and after therapy.

After 16 weeks of treatment with immune-targeting therapy, 30 out of the 45 patients had changed to a different CTAP, indicating that CTAPs may be dynamic over the course of treatment (**new Figure 6D** and **new Extended Data Figure 10D**). As predicted for immunomodulatory drugs, patients' CTAP classifications tended to become less immune-cell-abundant and resemble CTAP-F (16/30 samples that changed CTAP). We next asked whether CTAPs change in similar ways among patients treated with rituximab versus tocilizumab therapy (**new Extended Data Figure 10E-H**). After 16 weeks of rituximab therapy, most CTAP-TB patients exhibited a change where they now resembled other CTAPs, with 5/6 of the patient's post-therapy bulk samples now being classified as CTAP-F. Conversely, after 16 weeks of tocilizumab, CTAP-TB patients were less likely to change CTAPs, with 3/6 patients being assigned CTAP-TB after therapy.

Interestingly, among the 15 patients (out of 45 total) who retained the same CTAP classification after therapy, 8 patients began as CTAP-F (8/15 samples that did not change CTAP) suggesting that either this CTAP is more resistant to change in response to these treatments (**new Extended Data Figure 10D** and **new Figure 6E**). Notably, the vast majority (7/8) of the patients that started as CTAP-F were clinical non-responders after therapy. We acknowledge that these analyses are limited by small numbers per group, but they provide provocative preliminary data supporting the application of CTAPs in future clinical trials.

We describe these findings in the following new text:

*To investigate whether CTAPs change over time and are associated with treatment response, we applied our CTAP classification algorithm to the 45 R4RA patients who had synovial tissue biopsies before and 16 weeks after starting treatment. We found that CTAPs were dynamic during this period, with 30 out of 45 patients changing to a different CTAP (**Figure 6D** and **Extended Data Figure 10D**). Patients in the tocilizumab and rituximab treatment arms exhibited similar frequencies of change in CTAP (20/29 [69%] patients treated with rituximab and 10/16 [63%] patients treated with tocilizumab) (**Extended Data Figure 10E-H**). Among patients that changed CTAPs, CTAP-F was the most common CTAP at week 16 (16/30 [53%]), consistent with the mechanisms of action of rituximab and tocilizumab targeting inflammatory cells and pathways.*

Extended Data Figure 10D (left): Confusion matrix showing predicted CTAP assignment of pre-treatment (week 0) and post-treatment (week 16) synovial tissue samples obtained from 45 patients.

Figure 6D (right): Alluvial plot showing CTAP classification of samples prior to and at week 16 after starting treatment with either tocilizumab or rituximab ($n = 45$).

Extended Data Figure 10E-F: Confusion matrix and alluvial plot showing predicted CTAP assignment before and after treatment with rituximab ($n = 29$).

Extended Data Figure 10G-H: Confusion matrix and alluvial plot showing predicted CTAP assignment before and after treatment with tocilizumab ($n = 16$).

Second, we asked whether CTAPs prior to treatment can predict treatment response. If an association exists, it would not only help patients find effective treatments more quickly but would also motivate studying CTAPs in which standard treatments do not frequently work. Thus,

we used the 133 pretreatment bulk RNA-seq R4RA samples to classify patients at the beginning of therapy (either rituximab or tocilizumab), and tested whether their initial CTAP classification could predict response, which was quantified by the authors based on change in CDAI. We then built a multivariable logistic model in which we modeled therapy response status as predicted by CTAP. Compared to a null model in which response status was predicted without CTAP, the full model including CTAP significantly improved response prediction (p-value = 0.0105). In this full model, we found that CTAP-F had the most significant odds-ratio (0.25, $p = 0.0056$), suggesting that CTAP-F drives much of the predictive power for response to these treatments (**Figure 6E**). We built another model to account for potential confounding by controlling for sex, age, treatment, and CCP status, and we found that CTAP still significantly predicted treatment response, with CTAP-F exhibiting an even more statistically significant OR (OR = 0.066, p-value = 0.0098; data not shown).

We have added these data as supplementary figures (**Figure 6E** and **Extended Data Figure 10C**) and added the following text to our manuscript.

*To determine whether CTAPs can predict response to these treatments, we used our algorithm to determine the CTAPs of pre-treatment bulk RNA-seq samples from 133 patients in the R4RA study. We then compared the frequencies of responders (defined in the R4RA study as $\geq 50\%$ improvement in CDAI) versus non-responders among the CTAPs. We found that the frequency of responders varied by CTAP ($p = 0.0105$), with patients in CTAP-F having the poorest response, even after controlling for sex, age, treatment and CCP status (OR = 0.2619, $p = 0.0403$, **Figure 6E** and **Extended Data Figure 10I**).*

Extended Data Figure 10I (right): Graph of responder and non-responders stratified by CTAP ($n = 133$). **Figure 6E**. Association between clinical response and CTAPs in the baseline (week 0) samples from the R4RA study. Percentage of variance explained by CTAPs alone and p-value are calculated with ANOVA tests. 95% confidence intervals are shown.

Finally, we have modified the main text language to be less speculative. We have updated our original Line 489-499, which Reviewer #1 felt were too speculative, in the text below:

Targeting the specific cell subsets enriched in a given CTAP may be key in personalized RA treatment. For example, abrogating T-B cell communication with B cell-depleting antibodies (e.g. rituximab) or blocking costimulation (e.g. abatacept) in CTAP-TB may break the pathogenic mechanisms that drive inflammation in these patients^{18,82}. Conversely, patients with CTAP-TF and CTAP-M feature fibroblast populations with high IL6, an established target of current FDA-approved treatments of RA (e.g. tocilizumab). CTAP-TF and CTAP-M feature abundant IFNG-

expressing cells or IFN-associated gene signatures, suggesting that these patients may respond effectively to JAK inhibitors (e.g. tofacitinib, upadacitinib). Lastly, other CTAPs, such as CTAP-EFM and CTAP-F, have no obvious targets of currently available treatments and warrant further focused study.

Referee #2, Comment #1:

This is an interesting and comprehensive piece of work from arguably the world's premier collaboration in this domain. It is a necessarily statistical and bioinformatic analysis of a massive dataset. There will inevitably be further complexity beneath the 2/3 cell CTAP definitions proposed.

The work provides an original classification suggesting that RA synovial pathology can be distributed between six 'high-level' cell type abundance phenotypes or CTAPs, and that the pathological architecture within these CTAPs is relatively consistent. By further characterisation of the cell types associated with CTAPs and their transcriptional profiles, the authors provide suggestions as to their potential relevance to a precision medicine approach towards treating RA. They also provide some preliminary data suggesting that synovial CTAPs may be predictable from peripheral blood flow cytometry profiles. It is interesting, although perhaps not unexpected, that particular cell types tend to associate with one another within particular CTAPs; perhaps less expected is that, at a deeper level, particular subsets of those cell types tend to co-aggregate. The work will also provide an excellent dataset for workers within this field to study.

We thank the reviewer for their praise of the utility of our study and the highly collaborative team behind it. We fully agree that the idea that specific cell types and cell states co-occur together is a finding that is both important and surprising.

Referee #2, Comment #2:

An over-riding assumption of the work's interpretation is that relative cell abundance implies importance. Whilst this feels intuitively possible, the work may overlook critical cell types and their interactions. For example, a CTAP is defined when a cell type is more abundant than its average per cent frequency over all biopsies. As the authors state, however, in numerical terms a cell type can, at times, be more abundant in a particular CTAP than the cells that define it (because it still has a lower per cent frequency than its average across all biopsies). Further, the authors could have decided to adopt an alternative approach, for example classifying biopsies primarily on the basis of immune mediators and then tracing mediator dominance back to cell type. This perhaps would make less biological sense but potentially have more impact from a therapeutic perspective. Nonetheless, the work provides a potentially useful classification

and, as above, the dataset is available for other researchers to interrogate and manipulate.

We appreciate the reviewer's recognition of the complexity of CTAPs and comparing cell abundance across patients. One notable distinction is that we do not *define* a CTAP based on a cell type being more abundant than its average. Rather, we cluster samples into CTAPs based on cell-type abundances, and we *name* each CTAP based on a cell type being more abundant than its average. In our analysis, the CTAP classification is a launching point for an analysis of all cell subsets. All six cell types analyzed in this study influence CTAP definition, and all cell types are subsequently tested for associations with clinical and histological variables. Even if a cell type is not in the name of a CTAP, this association testing may still find critical substates of that cell type that are associated with a CTAP, inflammation metric, or another variable. For example, some non-myeloid populations such as capillary endothelial cells and mural cells are both associated with CTAP-M (**Figure 4**). Thus, CTAPs should not be the cause of overlooking critical cell types or states. The names of the CTAPs are bestowed based on the broad cell types but is not meant to imply that only the cell types mentioned in the name are important.

We have clarified this in the main text:

We named the CTAPs based on relatively enriched cell type(s): 1) endothelial, fibroblast, and myeloid cells (EFM), 2) fibroblasts (F), 3) T cells and fibroblasts (TF), 4) T and B cells (TB), 5) T and myeloid cells (TM), and 6) myeloid cells (M) (Figure 1H, Supplementary Table 4, Methods).

...

We next quantified how the composition of fine-grained cell states differed between CTAPs. Although CTAPs were named for the most relatively enriched cell types, we tested all cell types for associations with all CTAPs, recognizing that even less enriched cell types may contain subsets that have critical functions in a CTAP.

We agree that an immune mediator-focused approach might be an intriguing alternative to classify patients. Encouraged by the reviewers' comments, we assessed how effective this approach may be. In order to pursue this analytically, we first collapsed our single-cell data into 70 pseudo-bulk samples. We focused our analyses on 55 soluble factors from the highly relevant KEGG pathway "Cytokine-Cytokine receptor interaction." To classify the samples based on these immune mediators, we restricted the genes to those that encode these cytokines and soluble factors (n = 55 genes). We applied Principal Component Analysis (PCA) to these pseudo-bulk profiles. We observed that even when clustered based on immune mediators alone (i.e., with no cell type abundance information), the pseudo-bulk samples effectively grouped together by the cell-type-determined CTAP classification (**Extended Data Figure 1E-F**). This finding indicates that the CTAP definition is biologically meaningful and reflects immune mediator-based classification as suggested by the reviewer.

Extended Data Figure 1E-F. E. PCA of samples based on pseudobulk gene expression of 55 soluble immune mediators. Each dot represents a sample, plotted based on its PC1 and PC2 projections and colored by CTAPs. **F.** Heatmap of pseudobulk gene expression of soluble immune mediators across samples, grouped by CTAP. Boxes are colored based on the gene's scaled pseudobulk expression across samples.

We described this analysis in the main text of the manuscript:

Given the importance of cytokines and other soluble immune mediators in the pathogenesis of RA, we next asked whether classifying RA samples based on soluble immune mediators would replicate the cell type-defined CTAPs. Indeed, categorization based on pseudo-bulk expression of 55 cytokines, chemokines, and growth factors distributed the tissues similarly to the cell-lineage-based CTAP categorization (Extended Data Figure 1E-F, Methods).

We also note that there is a broad interest in cell-cell interactions, and as such our broader consortium is investing heavily in spatial transcriptomic assays to identify colocalization of cells with each other. The AMP-AIM program is a multi-year and multi-institutional effort that is now just commencing (<https://www.niams.nih.gov/grants-funding/niams-supported-research-programs/accelerating-medicines-partnership-amp>).

Referee #2, Comment #3:

It is perhaps surprising that there are only 6 CTAPs in such a genetically complex condition, as well as the relative constancy of cell subsets within each CTAP. As suggested above, a relatively minor population could still have an important influence over the biological behaviour of tissue if it has important interactions with a more abundant cell type. For example, suppl Fig 6E suggests that, whilst T cell neighborhoods enriched in CTAP-TF (permutation $p=0.036$) mainly comprised cytotoxic CD4+GNLY+ (T-12) and CD8+GZMB+ (T15), these subsets still form a relatively small proportion of overall T-cells. As another example, the work confirms that TFH and PFH associate with ABC and memory B-cells, mainly within the T-B CTAP (Fig 3A, C). IgM+ plasma cells (B-6), plasmablasts (B-7), and ABCs (B-5) were also positively associated with aggregates (permutation $p=0.007$) (suppl Fig

11B, Figure 5A) and yet IgM+ plasma cells (B-6) and plasmablasts (B-7) were NOT associated with T-B CTAP (Fig3, suppl Fig 7G). In fact Fig 3C suggests that IgM+ plasma cells were relatively depleted in this CTAP. A comparison of Fig 3C and 11B tells us that aggregates do not map accurately to the T-B CTAP, which is perfectly fine but illustrates the complexity of the data and care needed in its interpretation. As mentioned above, there will inevitably be further complexity beneath the 2/3 cell CTAP definitions proposed.

We thank the reviewer for noting the difficulty of classifying patients with a genetically complex disease and the care that must be taken in interpretation. In a cohort of 82 patients, settling on 6 reproducible patient phenotypes helped stratify patients into distinct subsets while maintaining some power and within-group stability for statistical analysis (see **Extended Data Figure 1A** showing Jaccard analysis and bootstrapping to support 6 groups). A larger patient cohort and finer-grained clustering of patients may produce a higher number of patient subsets that is consistent with the reviewer's expectations.

We agree with the reviewer's assessment that **Figure 3C** and **new Supplementary Figure 7B** (**old Supplementary Figure 11B**) show that aggregates do not perfectly map to CTAP-TB. We, too, were initially surprised by this finding. Upon further consideration, we realized that the assumption that aggregates should necessarily associate with TB-rich samples is false for two main reasons. First, our histologic aggregate score focuses on the single most advanced aggregate in the sample. This means that a tissue sample with a single moderately organized aggregate will have the same aggregate score as a different sample that has ten similarly organized aggregates. In other words, the aggregate score does not indicate total aggregate burden in the sample. Secondly, prior studies of synovial tissue have documented different kinds of aggregates composed of primarily T cells, primarily plasma cells, or T cells and B cells together (e.g. Scheel et al, PMID 20882667; Young et al, PMID 6197977, Schroder et al, PMID 8552609). The aggregate score does not distinguish between these different types of aggregates, and we cannot directly address this in our study, which primarily used disaggregated tissue and therefore cannot distinguish which cells came from inside aggregates versus outside aggregates. We hope this question will be studied further with quantitative analysis of high-dimensional microscopy and spatial transcriptomic data. Indeed, this is one of the goals of the AMP-AIM research initiative in which several of the co-authors are involved.

We edited the main text to make this nuance clearer:

Cell states and CTAPs are associated with histology and clinical metrics

*In addition to association with CTAPs, we used CNA to test for cell neighborhoods associated with histologic features of RA synovium (**Figure 5A**)⁸¹⁻⁸³. We scored samples for Krenn histologic inflammation and lining layer domains, in addition to discrete histologic cell density and aggregate scores that reflect inflammatory cell infiltration and organization, respectively (**Supplementary Figure 7A**). Many T cell states were associated with aggregate scores (permutation $p=0.0088$, adjusted for age and sex), including T cell neighborhoods in $CD4^+ T_{FH}/T_{PH}$ (T-3), consistent with their role in organizing lymphoid follicles^{84,85}, as well as $GZMK^+CD8^+$ T cells and some memory $CD4^+$ T cell populations (**Figure 5A, Supplementary Figure 7B**). Among NK cell neighborhoods,*

a GZMK⁺ NK cell cluster, NK-4, was also positively associated with both density and aggregate scores (permutation $p = 3e-04$ and $1e-04$, respectively) (**Supplementary Figure 7B**). Inflammatory myeloid neighborhoods within STAT1⁺CXCL10⁺ (M-6), SPP1⁺ (M-4) and inflammatory DC3 (M-9) (**Figure 5A, Supplementary Figure 7B**) were associated with both aggregate and density scores (permutation $p = 0.006$ and $p = 0.005$, respectively). Among B cells, IgM⁺ plasma cells (B-6), plasmablasts (B-7), and ABCs (B-5) were positively associated with aggregate scores (permutation $p = 0.007$) (**Figure 5A, Supplementary Figure 7B**) despite this population not being associated with CTAP-TB (**Figure 3C**). These disparate cell state associations with aggregate scores likely represent the diversity of aggregates in synovial tissue, which can be T cell-dominant, plasma cell-dominant, or T-B follicles^{55,86-88}.

We wanted to understand if histologic and clinical measures are explained by CTAPs, taking age, sex, cell count, and clinical collection site into account (**Methods**). CTAPs account for 18% of variance of histologic density ($p = 0.0035$) and 18% of variance for aggregates ($p = 0.0059$), with CTAP-TB and CTAP-TF having the highest scores for both (**Figure 5B, Extended Data Figure 8A**). Consistent with these observations, CTAPs are associated with Krenn inflammation scores ($p = 4e-04$), but not with Krenn lining scores ($p = 0.11$) (**Figure 5B, Extended Data Figure 8A**). CTAP-F, CTAP-EFM, and CTAP-M have the lowest corrected scores for all histologic parameters (**Figure 5B, Extended Data Figure 8A**). Ultrasound measurements in the biopsied joint (gray scale or power doppler scores) did not vary by CTAP (**Extended Data Figure 8A**). In our dataset, we did not observe an association between Krenn inflammation and power doppler scores, consistent with some prior studies⁸⁹⁻⁹¹ (**Extended Data Figure 8B**)

Referee #2, Comment #4 :

An important element of this work overall are the potential links between CTAPs and other clinical and histological parameters. This information is provided in suppl Fig 12 which, arguably, should be part of the main text. A few features of this Table are worth highlighting. Firstly, it is surprising that Krenn inflammation and US doppler associations do not have more similar associations, particularly with myeloid cell types, given that we believe US Doppler is a marker of synovial inflammation. Secondly, most statistically significant CTAP associations reflect histological parameters which, in some ways, feels rather tautologous, although CCP associations (Fig 12C) are of interest. Having said that, seropositivity overall does not associate with one or other CTAP. Notwithstanding the lack of statistical significance, the presence of the TF CTAP in seropositive but not seronegative patients is perhaps surprising and, again, may reflect deeper heterogeneity within CTAPs. The authors comment: "We argue that CTAPs from biopsies offer independent information from what physician assessments offer". This is definitely the case (eg Krenn associations) but how clinically valuable this additional information will be awaits further investigation.

We thank the reviewer for emphasizing the importance of clinically relevant results in our study. We agree that the information in **old Supplementary Figure 12** is valuable. Consequently, we

have moved this figure to **Extended Data Figure 8** to better highlight the insights that CTAPs can offer.

We also agree that some of our results are surprising, such as Krenn and US doppler not having more similar associations. On the other hand, other studies have documented the lack of consistent associations between US Doppler and Krenn inflammation (e.g., Just et al, PMID 31413866; Ramao et al, PMID 32531503; and Andersen et al, PMID 23475981). We plotted the relation between US doppler score and Krenn inflammation in our data (**new Extended Data Figure 8B**). There is very little correlation between US Power Doppler scores and Krenn inflammation in our dataset, likely because our study recruited only patients passing a specific threshold for inflammation, and we obtained biopsies from swollen joints (e.g., clinically tender and swollen and with an US gray scale ≥ 2 for QuickCore or ≥ 1 for portal and forceps biopsies). Thus, our associations are analyzing the spectrum of high disease activity RA patients and may minimize the strength of associations with Krenn inflammation.

We have updated the text to read as follows:

*We wanted to understand if histologic and clinical measures vary by CTAPs, taking age, sex, cell count, and clinical collection site into account (**Methods**). CTAPs account for only 18% of variance of histologic density ($p = 0.0035$) and 18% of variance for aggregates ($p = 0.0059$), with CTAP-TB and CTAP-TF having the highest scores for both (**Figure 5B, Extended Data Figure 8A**). Consistent with these observations, CTAPs are associated with Krenn inflammation scores ($p = 4e-04$), but not with Krenn lining scores ($p = 0.11$) (**Figure 5B, Extended Data Figure 8A**). CTAP-F, CTAP-EFM, and CTAP-M have the lowest corrected scores for all histological parameters (**Figure 5B, Extended Data Figure 8A**). Neither DAS28-CRP disease measures nor ultrasound gray scale or power doppler scores varied by CTAP (**Extended Data Figure 8A**). In our dataset, we did not see an association between Krenn inflammation and power doppler scores, consistent with some prior studies⁸⁷⁻⁸⁹ (**Extended Data Figure 8B**).*

Extended Data Figure 8B. Dot plot of Krenn inflammation versus power doppler scores. Each point is a sample.

We have delved further into the question of CCP associations in **Reviewer #1 Comments #4 and #6**. We agree with the Referee that seropositivity versus seronegativity may reflect an additional layer of heterogeneity within each CTAP. For example, given that CCP antibodies tend to develop years prior to RA diagnosis while CTAPs are potentially dynamic over time, it is possible that seropositive (or seronegative) patients all have the same CTAP at disease initiation and then CTAPs may evolve over time with treatment. However, we did not find any correlation between CTAPs and treatment-naive status (**Extended Data Figure 8M**), though we

may have lacked power given the relatively small number of samples in this analysis. We hope that this will be explored in better-powered future studies.

The new analysis of tissue samples from a published clinical trial supports the use of CTAPs as clinically valuable measures, as patients with CTAP-F have very poor treatment responses to both tocilizumab and rituximab (see **Reviewer #1 Comment #6**). We hope that the CTAP classification system is utilized in future clinical studies to help further investigate the link between CTAPs and response to specific treatments.

Referee #2, Comment #5:

There is a clear effect of RA duration and/or treatment with CTAP, with longer lasting/more refractory RA clustering within the EFM CTAP, and a disease duration ranging from 2.4 (TM) to 12 years (EFM) across CTAPs (Suppl Table 10). This also links to Fig 12L. More than 70% of TNF inadequate responders fall within the EFM CTAP and, whilst this association is not significant, it may have been appropriate to consider treatment and disease duration as potential confounders. Reinforcing this feeling are data in supp Table 1 (demographics), where Krenn cell density - which associates with CTAPs - reduces with disease duration/refractoriness. The authors address this in the text: "patients in CTAP-EFM tended to be older and have longer-standing RA than patients in other CTAPs and were mostly TNFi-inadequate responders although these associations were not statistically significant". Given the long-term ambition of using CTAPs as guides to precision medicine, perhaps at least a sensitivity analysis of the data could have been included treating disease duration and treatment as confounders.

We thank the reviewer for this thoughtful comment. We agree with the reviewer that we are underpowered to see a statistically significant association between TNFi-inadequate responders and RA CTAPs. We removed the following text from the manuscript: "patients in CTAP-EFM tended to be older and have longer-standing RA than patients in other CTAPs and were mostly TNFi-inadequate responders although these associations were not statistically significant". To fully answer these questions, it would likely be better suited with a longitudinal study and better design.

We agree with the reviewer that disease duration and treatment could be confounders for association tests. To address this, we repeated the cell type abundance analysis after adjusting for disease duration and treatment, and we found that the CTAPs remained largely unaffected (**new Extended Data Figure 1C**). The main differences occurred in samples in CTAP-M and CTAM-TM, which were mainly driven by differences in disease duration (see **Extended Data Figure 8K**). We have added a sentence to the main text to present this new analysis:

Adjusting for treatment and disease duration had little effect on clustering of samples (**Extended Data Figure 1C**).

Extended Data Figure 1C. PCA of samples based on cell type abundances, adjusting for disease duration and treatment. Each dot represents a sample, plotted based on its PC1 and PC2 projections and colored by CTAPs.

Referee #2, Comment #6:

Figure 12K is also of interest in terms of precision medicine. The authors comment: "Since CTAPs appear to correlate with known drug targets (Figure 6D) and can be assigned even with flow cytometry, we expect that CTAPs can be used to systematically query RA heterogeneity across technologies to improve the granularity of clinical studies and trials and potentially to guide therapy selection". This is a critical message and it would be of interest to understand how the authors plan to move this work forwards from a precision medicine perspective.

We thank the reviewer for this comment. We agree that the CTAP classification system has the potential to bring precision medicine closer to reality in rheumatoid arthritis. We make two points. First, we demonstrate how CTAPs can be used in a clinical trial context to predict treatment response. Second, we demonstrate how genetic risk factors can be interrogated in the context of CTAPs.

The new analysis detailed in **Reviewer #1 Comment #6** takes a step closer to this goal by applying the CTAPs to a clinical trial. This proof-of-concept analysis demonstrates the utility of CTAPs in clinical trials and argues for the ability to use less costly technologies, such as flow cytometry and bulk RNA-seq data to classify samples. The CTAP classification system can move our field closer to precision medicine by identifying which patients are most likely to respond to a given treatment as well as highlight CTAPs (e.g., CTAP-F) that require further study to identify new treatment targets.

We have edited the text of the **Discussion** to outline these future opportunities:

The CTAP paradigm provides a tissue classification system that captures coarse cell-type and fine cell-state heterogeneity. CTAPs use global cell-type frequencies and are thereby an accessible tool to categorize heterogeneity of tissue inflammation using multiple technologies. The model presented here may serve as a powerful prototype to classify other types of tissue

inflammation, including in other immune-mediated diseases. A deeper understanding of the heterogeneity of tissue inflammation in RA and other autoimmune diseases may shed new light on disease pathogenesis and reveal new treatment targets, key elements of precision medicine.

We also now demonstrate through analyses how CTAPs can be used to infer the mechanism of genes from RA genetic susceptibility studies. We argue that the genes from genetic studies may represent important potential drug targets. If we can identify key cell types and cell states that they are acting in, we might predict which CTAPs may be particularly responsive to therapeutics targeting these states. We describe this analysis in the following new text:

Cell states associated with CTAPs express RA-associated genes

We next tested whether genes implicated by RA genetic studies are preferentially expressed by cell states associated with specific CTAPs, which may help identify states that they act in to drive tissue inflammation in RA. Using data from a recent genetic study of RA with > 250,000 individuals from five genetic ancestry groups⁹⁷ we identified 71 genes that were likely to be causal (Methods) (Supplementary Table 6). We detected expression of all 71 genes in one or more cell types in our dataset (Supplementary Figure 11A). For RA genes expressed in a cell type, we tested if loadings from CNA for specific CTAPs were correlated with expression. A positive correlation indicates that a cell state expanded within a CTAP specifically expresses an RA risk gene (Figure 6F).

We identified 48/71 genes with expression patterns that were significantly positively correlated with one or more CTAPs for a cell type ($p < 0.05$, controlling for expression level, Methods), which is significantly higher than predicted by chance (median = 34, permutation $p < 0.01$) (Supplementary Figure 11B-C). HLA-DRB1 expression was correlated with CTAP-associated cell states in several cell types (Figure 6F). Some cell types expressed RA genes in different subsets of cells. For example, in T cells, LEF1 was more highly expressed in the naive states expanded in CTAP-TF, while IL6R was more highly expressed in T_{FH}/T_{PH} states expanded in CTAP-TB (Figure 3A, Figure 6F, and Supplementary Figure 11D). This potentially indicates a role for RA-associated IL6R regulatory variants in CTAP-TB, but since IL6R is also broadly expressed by all myeloid cells in our study (Supplementary Figure 11H), risk alleles may play a role in RA pathogenesis across CTAPs as well.

Some genes underscored the importance of signaling pathways that may be important in a specific CTAP, such as VEGF in CTAP-M (Extended Data Figure 9). For example, the rs146492555 SNP is associated with PRKCH, which encodes Protein Kinase C (PKC)- η , a mediator of VEGF-induced endothelial-cell differentiation⁹⁸. PRKCH and VEGF receptor genes KDR and FLT1 are highly expressed in endothelial cell states expanded in CTAP-M (Figure 4C, Figure 6C, Supplementary Figure 11E). VEGFA, a key signaling factor in neoangiogenesis and endothelial cell proliferation, is highly expressed in myeloid cell states expanded in CTAP-M; nearly 30% of all myeloid cells in CTAP-M express VEGFA compared to 5-10% among other CTAPs (Figure 4E, Figure 6E, Supplementary Figure 11F-G).

Figure 6F. Significance of correlations between RA risk gene expression and CTAP-associated cells. Significance levels are shown in red ($p < 0.01$), yellow ($0.01 < p < 0.05$), and white ($p > 0.05$). Genes with low counts (UMI>1 among < 5% of cells with a given cell type) were not analyzed in that cell type (gray boxes). Below the graph are UMAPs displaying normalized expression levels of selected genes in T cells (IL6R, LEF1) or endothelial cells (PRKCH). Genes with low counts (UMI>1 among < 5% of cells with a given cell type) were not analyzed in that cell type (gray boxes).

Supplementary Figure 11. Expression of RA GWAS-implicated genes. **A.** Heatmap of pseudobulk normalized expression of genes implicated in RA GWAS studies in cells from each indicated cluster and CTAP combination. Color scale ranges from brown (high) to turquoise (low) and is scaled across cluster+CTAP combinations shown. **B.** Statistical strategy to correlate RA-associated genes with CTAP-associated cells. Briefly, we measured the Pearson correlation coefficient between an RA GWAS gene's

normalized expression and the cell-neighborhood correlations with a CTAP (left). To define significance, we compared this correlation to the correlations computed between other genes of similar expression level and the same CTAP (right). **C.** Histogram of number of RA genes significantly correlated with CTAP-associated cells in at least one CTAP/cell type pair in null simulations. Significance threshold for correlation is $p < 0.05$ (left) or $p < 0.01$ (right). **D-F, H.** UMAPs colored by normalized expression of selected RA-associated genes in T cells (**D**), endothelial cells (**E**), or myeloid cells (**F, H**). **G.** Box plot of percent of myeloid cells expressing VEGFA in each CTAP. Points represent samples. Box plots show median (vertical bar), 25th and 75th percentiles (lower and upper bounds of the box, respectively) and $1.5 \times$ IQR (or minimum/maximum values; end of whiskers).

Referee #2, Comment #7:

OA tissue was used as a control. Whilst the sample number was small (N=9), is there any message to take home from the OA data regarding consistency vs heterogeneity vs CTAP similarities? Were these samples sufficiently homogeneous to act as a suitable control group?

We thank the reviewer for these questions about OA samples. We included these samples for comparison but not strictly as a control. In Figure 1G, we note these samples don't appear homogeneous. When we map the OA samples onto the PCA of our CITE-seq RA CTAPs, they appear most similar to CTAP-EFM and CTAP-F (**new Extended Data Figure 1D**). We also note that in Figure 2, there are noticeable key populations that are differentially abundant between OA and RA, which reinforces that OA, CTAP-EFM, and CTAP-F are different. We add the following to the main text of the manuscript:

Post-hoc mapping of the OA samples to the PCA demonstrates that the OA samples most resemble CTAP-EFM and CTAP-F (Extended Data Figure 1D).

Extended Data 1D. Projection of OA samples onto PCA of samples based on cell-type abundances from Figure 1J. OA samples are marked with gray points; RA samples are colored based on CTAP (left) or in blue (right).

Referee #2, Comment #8:

Three patients were biopsied twice and, whilst the authors comment that, overall, CTAPs appear constant over relatively short timespans (98 to 427 days), this is clearly a critical aspect that

requires further work. The cell-type composition of repeat biopsies was similar to the initial biopsy but not identical (mean Mahalanobis distance=1.55, permutation 430 p=0.073) (Supplementary Figure 14A-B).

We thank the reviewer for emphasizing the importance of tracing changes in CTAPs over time, especially with treatment. We agree that further study may offer additional insights into CTAPs' relationship with disease. While we have limited access to repeat biopsies in this cohort and follow-up studies are beyond the scope of this manuscript, we have reviewed publicly available synovial tissue bulk RNA-seq samples from a clinical trial with longitudinal transcriptomic profiling to attempt to address how CTAPs change over time. In **Referee #1, Comment #6**, we present a method to classify published longitudinal bulk RNA-seq samples into CTAPs defined in our study and observe that treatment may alter CTAPs even in the short time span of 6 months.

Referee #2, Comment #9:

Minor questions/comments: Why were surface proteins less informative for non-lymphocytes? This is important and perhaps deserves further emphasis/discussion because only transcriptional profiling was used for non-lymphocyte clustering, adding heterogeneity to analyses.

We thank the reviewer for raising this important point. It is not necessarily true in general that surface proteins must be less informative for non-lymphocytes. However, the surface proteins included in our panel (**Supplementary Table 2**) were largely targeted towards distinguishing lymphocyte subtypes/functions. When we developed the panel, we were guided by prominent markers in the literature, few of which are informative for endothelial cells and fibroblasts. Although we included broad non-lymphocyte markers, we did not have enough to differentiate between more granular non-lymphocyte subtypes. The K-L divergence of each surface protein marker is shown in **Supplementary Figure 4**. It is possible that surface proteins can be informative for non-lymphocytes, but it likely requires a more comprehensive panel.

We have modified the main text to make note of this issue:

*We defined finer-grained cell states and quantified cluster abundances within cell types (**Figure 2**). Surface proteins were informative for cell-state delineation in T and B cells (**Supplementary Figure 4A-C**), so we clustered cells on CCA canonical variates (CVs) capturing both RNA and protein data (**Supplementary Figure 4D-F, Supplementary Figure 5, Methods**). Myeloid, stromal, and endothelial cell states were defined by the mRNA component alone since surface proteins in this panel were less informative for these cell types.*

Referee #2, Comment #10:

"CD1c 206 + MZ-like B cells (B-3) and other non-plasma B cells were high producers of IL6 and TNF" - I believe the data (suppl 7D) support the former (IL6) but not the latter.

We agree that this statement warrants clarification. We have revised the text to the following:

CD1c⁺ MZ-like B cells (B-3) and other non-plasma B cells produce IL6 and TNF.

Referee #2, Comment #11:

"T cell neighborhoods enriched in CTAP-TF (permutation $p=0.036$) mainly consisted of cytotoxic CD4+GNLY⁺ (T-12) and CD8+GZMB⁺ cells (T-15)". CD4 NAÏVE T-cells also appear abundant but are not mentioned here (Fig 3A).

We thank the reviewer for this astute observation. We have revised the text to the following:

T cell neighborhoods enriched in CTAP-TF (permutation $p=0.036$) mainly consisted of cytotoxic CD4⁺GNLY⁺ (T-12) and CD8⁺GZMB⁺ cells (T-15) as well as naive CD4⁺ and CD8⁺ T cells (T-4 and T-16).

Reviewer #3, Comment #1:

This study represents the likely largest cohort ever to be collected, and with full data-sharing, represent a treasure-trove of data to be mined by the immunology and rheumatoid arthritis communities. However, the study does not fulfill the goals stated in the introduction, namely, to define patient subsets predictive of therapeutic responses. While some replicate analyses of selected patients is reported, implying consistency of traits/CTAPs, with the relatively modest sample sizes for clinical correlations, it is unclear whether these subtypes represent basic pathophysiologic subsets, genetic subsets, serologic subsets, or whether these subsets are primarily driven by inflammatory, clinical or other stochastic factors. While the analyses are meticulously reported and cutting-edge, as currently presented, the results (which many similar papers do) often comes across as a laundry-list, lacking clear clinical (recognizing the modest sample sizes with which to do this with) or biologic advances.

We thank the Reviewer for praising the rigor and utility of our dataset while also recognizing the challenges of working with necessarily limited patient numbers. We have addressed the Reviewer's concerns in several ways.

First, as outlined in **Reviewer #1, Comment #6**, we now show that CTAPs can change over the course of treatment and that certain CTAPs are more resistant to common therapies. These findings demonstrate the potential that CTAPs may have clinical value, and specifically that they may be predictive of a patient's clinical response (or lack thereof) to certain drugs and motivate the further study of CTAPs in future clinical trials.

Second, we have investigated the potential for this data to help interpret the role of genes implicated by GWAS. As detailed in **Reviewer #2, Comment #6**, we find that many RA-associated genes are more highly expressed in one or more CTAP-associated cell populations, more than expected by chance. These findings demonstrate the power of this data to define the function of RA disease genes in the context of cell states in the inflamed synovium.

Third, we have reworded the Introduction to make it clear that the CTAPs will require further testing in prospective studies to fully explore associations with treatment outcomes, pathophysiological pathways, and other aspects of RA heterogeneity. Our descriptions of methods to determine CTAPs from flow cytometry and now from bulk tissue RNA-seq put further study of CTAPs within easier reach. The reworded portion of the introduction is recounted here:

A more granular understanding of tissue inflammation and cell states may reveal synovial phenotypes that could inform prognosis, predict responses to targeted therapies, and potentially identify new treatment targets.

Finally, we note that this is the single largest resource of its kind. While there are other data sets applying bulk RNA-seq or histology to inflammatory synovial tissues, there are no single-cell synovial data sets of the size reported here. In addition, the ancillary data on surface protein marker staining, clinical data, flow cytometry data, and other molecular data types is unprecedented.

Reviewer #3, Comment #2:

It is disappointing, but perhaps not surprising that the CTAPs are not associated with treatment class, nor many clinical (including histologic) variables. One exception to this is CCP antibody correlations to TB subset. Given the substantial extant data on TNF and methotrexate (pathways, GEO data), did the investigators attempt to perform pathway or cellular analyses. I wonder if an excessive reliance on the 6 CTAP classes may be limiting discovery potential.

We understand the Reviewer's concern. Overall, we believe the CTAP classification system to be robust, informative, and flexible to new discoveries, as demonstrated by the following new analyses. In summary, CTAPs capture broadly cellular heterogeneity in synovial tissue samples, and alternative analysis strategies largely define the same populations.

As detailed in **Reviewer #2 Comment #2**, using Principal Component Analysis on pseudo-bulk data from our cohort, we classified the samples based on expression of 55 key soluble immune mediators (cytokines, chemokines, etc). These sample clusters are essentially identical to the cell-type-based CTAPs. We prefer the CTAP classification scheme, since it is relatively easy to implement with other technologies (e.g., flow cytometry).

Second, we have performed *in vitro* experiments to explore the effects of cells and soluble factors on blood-derived monocytes and on B cells, demonstrating that CTAP-associated cells and factors can shape myeloid populations in the directions predicted by the CTAPs. These analyses are described in **Reviewer #1 Comment #2**.

The reviewer registered disappointment that the CTAPs do not more cleanly align with treatment groups. Of note, while we recruited patients into three treatment groups, their treatment histories were heterogeneous. That is, the patients were not treated uniformly before or after recruitment. We also note that past failure of one treatment is a poor predictor of future success with a different treatment, as evident in the lack of a recommendation of one biologic agent over another in the EULAR and ACR treatment criteria. This suggests that there is significant molecular heterogeneity within treatment failure cohorts, as we indeed see in **(Extended Data Figure 8M)**.

However, our new results demonstrate the potential for CTAP to predict treatment response in a properly structured study. Using a classification algorithm to assign CTAPs to synovial tissue

bulk RNA-seq data, we compared the response of CTAPs to tocilizumab versus rituximab, i.e. biological therapies targeting two distinct and relevant pathogenic pathways in RA. We found that CTAP-F was less likely to respond to either treatment compared to the other CTAPs. See **Reviewer #1 Comment #6** for full details on how this analysis was performed.

We also were inspired by the Reviewer's comment to directly measure TNF-associated gene programs among fibroblasts, a cell type that is known to be highly sensitive to TNF stimulation. In this analysis, we defined a fibroblast-specific list of TNF-induced genes using a dataset generated by Slowikowski et al (PNAS 2020, PMID 32079724). We then quantified expression of these genes across fibroblasts in our dataset. Interestingly, these TNF-induced genes were most highly expressed lining fibroblasts, largely driven by matrix metalloproteinases, known lining products. There were no differences by treatment groups (e.g., no difference in patients who failed TNF blocker therapy). Perhaps this approach would be more effective in a structured clinical study.

Reviewer #3 Response Figure 1. Expression of TNF-induced genes among fibroblasts. The gene list was obtained by identifying genes induced by TNF alone (dark green) or TNF + IL-17A (light green) in a study of cultured RA synovial fibroblasts stimulated in vitro and analyzed by bulk RNA-seq (Slowikowski et al, PMID 32079724). Treatment groups: (1) untreated; (2) methotrexate inadequate-responder; (3) TNFi inadequate responder; (Repeat) three samples of repeat biopsies several weeks after baseline biopsies.

We agree with the reviewer that different approaches to analyzing these data may yield additional new discoveries and insights. We look forward to seeing how other investigators use these data in years to come, and we have added the following text to the Discussion:

Targeting the specific cell subsets enriched in a given CTAP may be key in personalized RA treatment. For example, abrogating T-B cell communication with B cell-depleting antibodies (e.g., rituximab) or blocking co-stimulation (e.g., abatacept) in CTAP-TB may break the pathogenic mechanisms that drive inflammation in these patients^{18,82}. Conversely, patients with CTAP-TF

and CTAP-M feature fibroblast populations with high IL-6, an established target of current FDA-approved treatments of RA (e.g., tocilizumab). CTAP-TF and CTAP-M feature abundant IFNG-expressing cells or IFN-associated gene signatures, suggesting that these patients may respond effectively to JAK inhibitors (e.g., tofacitinib, upadacitinib). Lastly, other CTAPs, such as CTAP-EFM and CTAP-F, have no obvious targets of currently available treatments and warrant further focused study. CTAP-F in particular appears to be unresponsive to certain immune-targeting treatment strategies.

Reviewer #3, Comment #3:

This effort may be one of the larger and more comprehensive efforts with CITE-Seq, but the results/impact is relatively modest, being confined to classification of adaptive immune subsets. This substantially reflects the much lower number of measures (58) and specific marker selection. I don't believe the correct table was uploaded for Supplementary Table 2. STable 8 does summarize the CITE-Seq results, but restricted by cell type. Might be more informative to include antibody readings for all cells for all 58 markers.

We thank the reviewer for identifying potential difficulty in accessing the correct tables and figures in our manuscript. We have uploaded the correct table for **Supplementary Table 2** so that it contains the list of protein markers included in the CITE-seq antibody panel. We do agree that robust classification of non-immune subsets (such as fibroblasts) based on protein markers requires specific markers catered to those cell types. At the time the panel was developed, we did not have access to sufficient antibodies against such markers to incorporate into the panel. As requested by the Reviewer, we have added heatmaps of antibody readings for all protein markers across all cell types (**Supplementary Figure 2H**). The antibody expression readings for all cells of each marker are available on our cell browser website (<https://immunogenomics.io/ampra2/> (account: ampra2, password: synovium2021)). To view protein expression, the user needs to search for the marker followed by "prot" (e.g., CD45RA_prot or CD4_prot).

A similar issue was also raised by **Reviewer #2 (Comment #9)**. To address both Reviewers' concerns,

we have modified the main text as follows:

*We defined finer-grained cell states and quantified cluster abundances within cell types (**Figure 2**). Surface proteins were informative for cell-state delineation in T and B cells (**Supplementary Figure 4A-C**), so we clustered cells on CCA canonical variates (CVs) capturing both RNA and protein data (**Supplementary Figure 4D-F, Supplementary Figure 5, Methods**). Myeloid, stromal, and endothelial cell states were defined by the mRNA component alone since surface proteins in this panel were less informative for these cell types.*

Reviewer #3, Comment #4:

Given the key roles for cell differentiation, proliferation and trafficking in RA, general application of RNA velocity approaches may provide more dynamic insight into the CTAPs, and possibly even some form of validation of the value of these categories. It would undoubtedly provide insight in the fine-scale clustering (e.g. infiltrating monocytes, tissue macrophages). This might nicely complement the cellular-based CNA analysis

We thank the reviewer for suggesting potentially relevant computational analyses. We have experience in our group with RNA velocity. However, we and others have encountered problems with the formulation, interpretability, and stability of RNA velocity results, especially when the data contain many patients and batches that confound trajectories. An excellent summary and presentation of these open problems of RNA velocity can be found in a recent publication from Dr. Lior Pachter's group (PLOS Computational Biology 2022, PMID 36094956).

Instead of resorting to computational techniques, we have used *in vitro* studies to investigate the relationships between different cell subsets in our study, as detailed in **Reviewer #1 Comment #2**.

The differentiation pathways connecting cell subsets certainly warrants further investigation, and we hope our dataset will be useful for other investigators performing dedicated studies of this issue. For T cells and B cells, TCR and BCR repertoire studies shed valuable light on connections between subsets and are being explored elsewhere (Dunlap et al, <https://www.biorxiv.org/content/10.1101/2023.03.18.533282v1>).

Reviewer #3, Comment #5:

Were genetics performed on this cohort? If so, is there any evidence that disease subtypes (via cell-specific epigenetics) map to relevant genetics scores?

We thank the reviewer for suggesting we include a genetic component to our study. The participants in the CITE-seq cohort were genotyped, which provides an interesting opportunity to associate CTAPs with relevant genetic scores. In our study, we observed that CCP is associated with CTAPs, while previous data suggests that CCP is associated with specific HLA alleles. Thus, we first performed HLA imputation on the patients of our CITE-seq cohort to determine which relevant HLA alleles our patient has. Second, we obtained the RA risk odds ratio for each HLA allele with a focus on HLA-DRB1 from our previously published HLA genetics paper (Buhm Han, ..., Raychaudhuri, AJHG, 2014). Third, we calculated an HLA risk score for each patient from this study by multiplying each variant dosage with disease risk odds ratio based on the 46 overlapped alleles on the HLA-DRB1 region. Lastly, we associated HLA imputation risk scores with identified CTAPs controlling age and sex. We present our results in

the plot below, where we show the corrected risk score for each CTAP (**new Extended Data Figure 8D**). This result suggested that CTAP-TB patients present the highest risk score, while CTAP-EFM patients have the lowest risk score. We noted the sample number is relatively small, so the association is not statistically significant. (For additional genetic association studies beyond the HLA-DR locus, please see **Reviewer #2, Comment #6**.) We have added the following text to introduce this new finding:

HLA-DRB1 is the strongest genetic RA risk factor for seropositive disease, so we stratified our patient cohort by HLA-DRB1 genotype. We did not find that the risk alleles were associated with a particular CTAP, although there was a trend toward the strongest association with CTAP-TB (Extended Data Figure 8D).

Extended Data Figure 8D. Corrected RA HLA-DRB1 risk scores and their associations with CTAPs, Percent of variance explained by CTAPs only and p-value are calculated with ANOVA test

We are also very interested in using these data and related data sets to understand the link between genetics, epigenetics and cell states. This is a topic that has been of high interest to us for some time. These vast and important questions require additional data capturing the epigenetic landscape of single cell states. We are exploring this topic in great detail elsewhere (Weinand et al, preprint available at <https://www.biorxiv.org/content/10.1101/2023.04.07.536026v1>; Sakaue et al, preprint available at <https://www.medrxiv.org/content/10.1101/2022.10.27.22281574v2>; Gupta et al, preprint available at <https://www.medrxiv.org/content/10.1101/2023.02.24.23286364v1>).

Reviewer #3, Comment #6:

This group previously published an elegant spatial mapping focused on stromal cells and the NOTCH pathway. More generally, could spatial approaches validate the 6 categories better? Especially with respect to the inferences nominally inferred to by CNA? The histology scores used clinically may lack the multi-dimensionality of research spatial approaches (multi-dimensional immunofluorescence, spatial transcriptomics, RNAScope). In particular, given the high fraction of highly heterogeneous endothelial cells, spatial approaches may be particularly illuminating.

We thank the reviewer for their positive comments regarding previous spatial studies from our group as well as suggesting its potential utility for studying CTAPs. In our revised manuscript, we present new immunofluorescence microscopy data in both pictorial and quantitative forms. We are also very enthusiastic about the use of spatial transcriptomics to understand tissue heterogeneity in RA. As described in **Reviewer #2, Comment #2**, this effort is a major undertaking of the AMP AIM consortium, which has been developed specifically to address this issue. Currently that consortium is developing protocols and methods, and we hope that exciting results will be forthcoming.

Here we have augmented the manuscript to include immunofluorescence microscopy data. In new **Figure 1J** (below) and **Supplementary Figure 3** (a 6-page figure not shown here due to space constraints), we show images of representative synovial tissue fragments from each CTAP, stained with H&E or stained for stromal/myeloid markers (CD90, CLIC5, CD68, CD34, HLA-DR, CD3) or lymphocyte markers (CD3, CD19, CD138). This is a subset of the 150 analyzed tissue fragment samples that we include now as part of this resource paper. The immunofluorescent staining panels suggest interesting patterns, such as a localized vs diffuse arrangement of macrophages in CTAP-TM vs CTAP-M, that warrant further investigation with dedicated, rigorous spatial analysis that is beyond the scope of what we can perform for this manuscript. In **Extended Data Figure 2**, we quantified cell types in the largely unmanipulated synovial tissue sections used for immunofluorescent staining to validate the CITE-seq-based cell quantitation performed on disaggregated synovial tissue cells. Indeed, IF-based quantification of cell types from higher-density fragments resembled the patterns predicted by each patient's CITE-seq-based CTAP assignment.

Spatial mapping is certainly of interest, and the synovial tissue samples in this cohort will be used for a separate imaging mass cytometry pipeline with 30+ markers. That study is now being pursued with the knowledge of CTAPs to develop a marker panel of interest. Given that it is of large scale and will not be ready for publication for some time, we believe it would be best to report those results as a separate manuscript down the line.

Here is the new text describing the microscopy findings:

Cell frequencies in synovial tissue sections follows CITE-seq-based CTAP patterns

To examine how robust CTAP classification was across multiple biopsy fragments from the same joint, we performed immunofluorescence microscopy staining on independent synovial tissue biopsy fragments from each patient (n = 36) (Figure 1J and Supplementary Figure 3). As predicted by the CITE-seq-based CTAP classifications, synovial tissue from patients with CTAP-TB displayed large numbers of B and T cells, often in well-organized aggregates. At the other extreme, synovial tissue from patients with CTAP-EFM and CTAP-F contained few B cells, T cells, or plasma cells.

We next quantified these cell-type proportions in each individual biopsy fragment for comparison with the disaggregated CITE-seq-based cell frequencies. Quantitation of total cellular composition demonstrated that fragments with highest cell density (top 50%) contained 86% of total cells and are therefore likely the primary drivers of CTAP classification (Extended Data Figure 2A). When we measured the cell-type proportions from these high-density fragments, they followed the patterns of cell frequencies predicted by the CITE-seq-based CTAP assignment (Extended Data

Figure 2B). For example, CD20⁺ (i.e., non-plasma) B cells were most frequent in CTAP-TB, whereas CD68⁺ myeloid cells were most frequent in CTAP-M and -TM (**Extended Data Figure 2B**). These data suggest that CTAPs are stable across synovial tissue in a particular joint.

Figure 1J. Representative synovial tissue fragments from each of the CTAPs. Top row shows hematoxylin and eosin (H&E) staining. Middle row shows immunofluorescence microscopy for CD3 (cyan), CD34 (red), CD68 (violet), CD90 (gray), CLIC5 (yellow), HLA-DR (green), and DAPI (blue). Bottom row shows immunofluorescence microscopy for CD3 (cyan), CD20 (magenta), CD138 (green), and DAPI (blue). Single-color images of each fragment are found in Supplemental Figure 3.

Supplemental Figure 3 not shown here due to space constraints.

Extended Data Figure 2A. Bar graph of proportion of total cells located in high-density and low-density fragments, as captured by histology imaging.

Extended Data Figure 2B. Box plots of the proportion of cells in high-density fragments expressing each marker in histology imaging, stratified by CTAP. Points represent outlier samples ($> 1.5 \times$ IQR from median). Box plots show median (vertical bar), 25th and 75th percentiles (lower and upper bounds of the box, respectively) and $1.5 \times$ IQR (or minimum/maximum values; end of whiskers).

Reviewer #3, Comment #7:

By qualitative inspection (Fig 1G), osteoarthritis appears to be similar to EFM RA, yet on finer subclustering, the gene expression patterns are quite distinct. Does this undermine the RA CTAP proposed in any way? Is the diagnostic distinction between OA and RA so marked and definitive that this does not undermine at all the RA CTAP definitions?

We thank the reviewer for noting the potential importance of the OA samples included in this study. In **Figure 1G**, we note these samples don't necessarily appear homogeneous. We have mapped the OA samples onto our CITE-seq RA CTAPs, and they appear to most resemble CTAP-EFM and CTAP-F (see **Reviewer #2, Comment #7**). We also note that in **Figure 2**, there are noticeable key populations that are differentially abundant between OA and RA. These observations suggest that, while OA and CTAP-EFM are similar in certain aspects, they are distinct in both gene expression patterns and the cell substates that are present.

Classically, the diagnostic distinction between OA and RA involves the presence of higher inflammation in RA patients (though we now know that OA can be inflammatory, as recently reviewed by Sanchez-Lopez et al, Nature Reviews Rheumatology, 2022, PMID 35165404). We note that a prerequisite for RA patients to be enrolled in this study was passing a specific threshold for inflammation both in terms of CDAI score as well as ultrasound grayscale score. Thus, our associations are analyzing the spectrum of high disease activity RA patients that are likely to be distinct from OA, which reinforces the robustness of our CTAP definitions.

Reviewer #3, Comment #8:

Did the investigators test for batch effects across sites to assure generalizability? The central processing likely substantially mitigates this, but differences between pinch biopsies vs.

synovectomies, vs. samples attained at arthroplasty are likely substantial confounders. Joint location?

We thank the reviewer for suggesting potential confounding factors that should be accounted for in our tests. The RA tissue samples in our CITE-seq cohort were all recovered from biopsies or synovectomies. We did not include any RA samples from arthroplasties, so this was not a factor in this analysis. The breakdown of CTAPs by tissue collection procedure (biopsy vs synovectomy) and by joint are found in **Extended Data Figure 8H, J**.

In our CNA correlation analyses, we did not include site as a covariate. Instead, we randomized samples by site (and treatment group) when determining the batches of samples for the tissue disaggregation and sequencing pipeline. We then included technical batch as a covariate in the analysis of the results. In practice, we have observed that technical batch is often the largest confounder. With these precautions, we felt that the technical effect of site should be minimal.

We were reluctant to include site as a covariate since the different sites had different patient populations. For example, some clinical sites recruited mainly newly diagnosed, treatment-naive RA patients, while others recruited primarily TNF inadequate responders. Most clinical sites performed biopsies of only one kind (portal-and-forceps vs needle vs synovectomy), which determines the joints amenability for biopsy (e.g., one cannot do a portal and forceps biopsy of an MCP joint). We worried that using site as a covariate would reduce our ability to see biological differences stemming from these factors.

Reviewer #3, Comment #9:

Fig 5A involves a large number of comparisons (albeit likely accounted for with permutation testing) against rather general histologic measures (aggregation, density), with only very general, well-known biologic results (T cells correlate with aggregate scores). Much of Fig 5 could be relegated to supplementary materials.

We thank the reviewer for suggesting an alternative style for organizing the data in our figures and results. Our preference would be to keep this figure in the main text. We believe that, while several of the biologic results presented in some of our figures are well-known, readers are highly interested in evaluating how our CTAP definitions and analyses agree with previously published results and literature. In particular, with **Figure 5A**, we show the association of fine-grain clusters, such as Tph cells (T-7) and TemRA CD8 T cells (T-15), with histologic measures, which is new. Indeed, the finding that TemRA CD8 T cells, the stereotypic CD8 T cell effector cell type, are negatively correlated with aggregates might be somewhat of a surprise given studies showing that CD8 T cells are necessary for proper organization of lymphoid follicles in synovium (Wagner et al, JI 1998, PMID 9834130; Kang et al, JEM 2022, PMID 12021312). Similarly, while it may not be surprising that the STAT1⁺CXCL10⁺ myeloid cell cluster correlates with aggregates, it is interesting that the IL1B⁺FCN⁺ macrophage population (which seems to have a strong TNF signature) is negatively correlated with aggregates. **Figure 5A** does indeed

involve a large number of comparisons, and we feel that this is its strength - it will help readers generate hypotheses for future studies. Similarly, **Figure 5B** shows clinical correlations, an area of strong interest for many RA investigators, that achieved statistical significance, with a concisely presented summary of all clinical factors presented in **Figure 5C**. We therefore respectfully request to keep these panels in the main figures.

Reviewer #3, Comment #10:

I am concerned that at least parts of Fig 6 could be misleading. The absence of cytokine expression by the 10X system does not mean that it is not expressed, given its well-known sparse nature. On the contrary, it is quite possible that small amounts of cytokine secretion from selected cells, in close proximity, may be the actually therapeutically critical pathway.

We thank the reviewer for identifying potential areas in which we can provide further clarity to how we present results. We agree that the absence of cytokine expression from sparse 10x data does not necessarily mean it is not expressed. To avoid making conclusions on cytokines that were poorly expressed in our data, we directed our focus specifically to those cytokines that were relatively well-expressed. To avoid confusion, we have edited the text by replacing “expressed” with “detected” to clarify we are evaluating cytokines that we were able to detect. In addition, we have added a sentence to specifically caution readers to interpret the data in **Figure 6A** with the sparsity of single-cell RNA-seq data in mind:

Cytokines, chemokines, and their receptors are key effector molecules and potential treatment targets in RA. We therefore analyzed their transcript levels across cell states, keeping in mind that these transcripts are often sparse in single-cell RNA-seq data (Supplementary Figure 8). Most cytokines and chemokines are detected predominantly in one cell type (Figure 6A). For key cytokines produced by multiple cell types, we quantified the relative contributions of each cell type. For example, we detected TNF in roughly equal numbers of T cells and myeloid cells while fibroblasts, endothelial cells, and B cells dominated among cells with detectable IL-6 (Figure 6B).

Reviewer #3, Comment #11:

'Line 395-6: Most cytokines produced by only one cell type'--this is likely biologically not true, to a substantial amount. Spatial expression approaches, as opposed to complete reliance on CNA may provided added support for specific claims. Also consider application of CellPhone db or other similar programs to assist with a more systematic evaluation of source-target cross-talk.

We thank the reviewer for suggesting potentially interesting cell-cell interaction analyses. To provide a systematic evaluation of cell-cell cross-talk, we have applied a robust method, CellChat (Jin et al, Nature Communications 2021, PMID 33597522), to our dataset, thereby querying more than 2,000 ligand-receptor pairs (compared to approximately 900 ligand-receptor

pairs in CellPhoneDB). We found a large number of potential molecular interactions between different clusters that reinforce our findings from CNA and other approaches described in the manuscript, including pathways involving VEGF, TNF, SPP1, Notch, and MHC class II. We included ligand-receptor pairs with a permutation p-value < 0.05 in **new Supplementary Figure 9** and have added the following text to the manuscript:

We complemented this analysis with a cell-cell communication analysis⁹³, which confirmed several of these pathways (Supplementary Figure 9).

In addition to the computation method described above, we also pursued functional assays to evaluate potential interactions that could potentially occur within CTAPs. These experiments are described in **Reviewer #1, Comment #2**.

Supplementary Figure 9. Outgoing (left) and incoming (right) cell-state interaction patterns identified with CellChat. Boxes are colored based on the relative strength of the signaling molecule in that cluster.

Reviewer Reports on the First Revision:

Referees' comments:

Referee #1 (Remarks to the Author):

In my view the authors have adequately dealt with the reviewer's comments. I was reviewer 1 and I am pleased to see the additional data

Referee #2 (Remarks to the Author):

I thank the authors for comprehensively addressing the points that I raised regarding their original manuscript, including clarification of my interpretation of some of the fundamental features of a CTAP and explanation of some potential inconsistencies, for example surrounding synovial aggregates. They have also demonstrated that an alternative classification algorithm, based on immune mediators rather than cells, leads to a very similar categorisation of samples.

The authors have also performed three additional pieces of work. In the first, they demonstrate the potential functional relevance of CTAP cell neighbourhoods, using in vitro assays focussed on B-cells and monocytes. In a second, they link potentially causal genetic SNPs to specific cells within CTAPs, which could contribute to our understanding of the functional relevance of these SNPs (CTAPs are now shown not to be fixed across time but this does not negate the potential relevance of these associations). Lastly, they have developed a technique to reclassify tissues subjected to bulk RNAseq into CTAPs and apply this to the R4RA trial.

Comments.

The reclassification of the R4RA trial is probably the most relevant aspect of the revised manuscript, in terms of a precision medicine message. Whilst the F CTAP may ultimately prove more informative than classifying tissue purely according to cellular dominance, at present I am not convinced that we are much further forward. The Rivellesse Nature Medicine paper has already associated a fibroblast-rich (pauci-immune) pathotype with refractoriness to treatment. Furthermore, in that post hoc analysis, a B-cell poor and myeloid rich synovium, identified by more standard deconvolution of bulk RNAseq, associated with a 77% likelihood of response to tocilizumab and only 14% response to rituximab. A B-cell rich synovium did not predict response to rituximab, consistent with the T-B CTAP also not predicting response to RTX.

In the current reanalysis of R4RA the authors clearly demonstrate that CTAPs change with treatment, which is a new and important observation. Curiously, however, 5/9 TM CTAPs become T-B after rituximab, which seems counter-intuitive and is not discussed. As mentioned above, the only CTAP that predicted (non)-response was CTAP F. Despite significant IL6 expression, neither CTAP TF nor M predicted response to TCZ, despite the data from the Rivellesse paper that a myeloid rich synovium (regardless of B-cell infiltrate) associated with a 70% response to TCZ. I recognise that, by definition, we cannot equate M-CTAP with myeloid-rich synovium by deconvoluted bulk sequencing (or indeed F-CTAP with fibroblast-rich synovium) but, at least according to this reanalysis of R4RA,

CTAPs do not appear to provide an improved theragnostic prediction.

I note that the relatively rare CTAP EFM seems to include a high (?5/6) frequency of responders but presumably in a non-discriminatory manner. This, of course, could be 'noise' due to small numbers, and is not statistically significant, but feels contrary to the statement: "other CTAPs, such as CTAP EFM and CTAP-F, have no obvious targets of currently available treatments and warrant further focused study". Almost two thirds of M CTAP also respond to either RTX or TCZ.

CTAPs TB and TM, both highly cellular CTAPs, segregate more or less equally between responders and non-responders (not quite in keeping with the statement: "abrogating T-B cell communication with B cell-depleting antibodies (e.g. rituximab) or blocking costimulation (e.g. abatacept) in CTAP-TB may break the pathogenic mechanisms that drive inflammation in these patients").

In summary, this work remains outstanding and world-leading in its depth and breadth. The additional information provided further increases our understanding of this novel synovium classification system. Continued evaluation is likely to further our understanding of RA pathogenesis.

Referee #3 (Remarks to the Author):

This resubmission is substantially revised from the prior submission. The authors have been moderately responsive to the prior review, with a comprehensive re-writing of the discussion and results from 2-3 of 6 of their major figures. The anchor for the overall manuscript continues to be their 6 high level CTAP subtypes, namely TB, TF, TM, M, F, EFM. Four major additions/new work products are provided:

1. The 6 CTAP subtypes are dynamic with treatment (Surace et al., Nature Medicine 2022). Mahalanobis classifiers were used to place patients with serial bulk RNASeq (45 are serial, pre- and post-16 week treatment with either rituximab or tocilizumab, R4RA trial) into CTAP categories.
2. Cell-cell interactions in vitro.
 1. SEB: superantigens of TB cross-talk
 2. Monocyte subsets with supernatant
3. Genetics added (Fig 6). The 71 high confidence rheumatoid arthritis genes were projected onto the single cell data, which is very insightful for the field.
4. Microscopy. IF from 36 linked joints; SFig 3.

The authors have streamlined items that were off topic/off the mainstream of the currently most effective RA medications (e.g. deleting data directed toward IL1 blockade), in order to focus on the most effective current agents. They have added, in response to the prior review, many pertinent negatives in supplementary materials.

Major comments

1. The 6 CTAP subtypes being the anchor for this manuscript. Fig 1 from the original submission was based on moderate to highly active RA, of 3 major groups; a) no prior treatment, b) methotrexate treatment, and c) anti-TNF non-responders. They provide additional support for the 6 major CTAPs,

including secreted factor analyses and immunofluorescence from the same joints. However, this last, important new data on 36 linked joint samples (new Fig 1J, SFig 3) supports a more dichotomous distinction between CTAP-TB vs. CTAP-F & CTAP-EFM; it generally confirms that multiple samples from the same joint are of the similar general subtype. I continue to have concerns regarding the overall meaning and impact of their 6 subtype classification for the following reasons:

- Their new data from the Surace et al clinical trial (Fig 6) pre-treatment demonstrates markedly lower pre-treatment frequencies for two of the six subtypes, CTAP-TF and CTAP-EFM. Surace et al. includes only anti-TNF non-responders, so represents a substantially different cohort compared to their single cell cohort.

- The high level lumping of B cells and plasma cells in the present CTAP definitions. Given the key role for B cells in RA pathogenesis overall, this seems misguided. The exclusion of one of their 6 major cell cluster types (NK cells--were there CITE-Seq markers for NK cells) excludes important secreted mediators (e.g. IFNG). Their more fine-scale transcript clustering (24 T cell clusters, 9 B/plasma cell clusters; 15 myeloid; 5 DC markers) provides much more refined gene selection than can be achieved with their protein (limited measures)-RNA multimodal CTAP classifier.

- That the present 6 CTAPs have two (TM, TF, TB), one (M, F) and 3 (EFM) major cell subtypes. I am concerned that this reflects an arbitrary reflection of their reliance on protein-RNA multimodal classifiers at the beginning of their study. Given the present addition of the very important Surace data to close the manuscript, I would favor a more transcript-focused classifier (esp. for Fig 6), relying on CITE-Seq for selected validation of key refined subclustering.

2. The additional studies (Fig 3) focused on the CTAP-TB nicely justifies the TB subset.

3. The additional monocyte focused studies details nicely their enormous heterogeneity, as does the serial analyses of the Surace data (alluvial plots). Many patients originally CTAP-M spread to multiple subsequent subtypes after 16 weeks of therapy.

4. CCP antibody levels: are they available in the Surace cohort? In the Darrah et al. review, 28937414, the source of citrullination is from innate immune cells, so not necessarily surprising that CTAP-TB have lower CCP antibodies.

5. The imperfect mapping (only 6 of 7 patients) of CTAP classifiers with bulk in Figure 6 further underscores the questionable value and overly heavy reliance on this classification. A gene-based classifier selected from their 77 fine-scale clusters might well be more insightful and reproducible.

Minor

- Differences between their original cohort and R4RA trial should be briefly mentioned in the discussion. What implications might this have for differences in cell clusters?

- Expression of high confidence associated genes: consider adding bulk data for non-detected genes from single cell

- HLA expression & B cells: Can a reference be added to the discussion with respect to the cellular contributions of MHC class II alleles

Author Rebuttals to First Revision:

We are pleased by the positive responses of the reviewers to the substantial revisions of our manuscript. We have now addressed the additional comments below.

Referees' comments:**Referee #1 (Remarks to the Author):**

In my view the authors have adequately dealt with the reviewer's comments. I was reviewer 1 and I am pleased to see the additional data

We thank this reviewer for their very thoughtful comments, and we are happy that they consider the revised manuscript a thorough response to the comments.

Referee #2 (Remarks to the Author):

I thank the authors for comprehensively addressing the points that I raised regarding their original manuscript, including clarification of my interpretation of some of the fundamental features of a CTAP and explanation of some potential inconsistencies, for example surrounding synovial aggregates. They have also demonstrated that an alternative classification algorithm, based on immune mediators rather than cells, leads to a very similar categorisation of samples. The authors have also performed three additional pieces of work. In the first, they demonstrate the potential functional relevance of CTAP cell neighbourhoods, using in vitro assays focussed on B-cells and monocytes. In a second, they link potentially causal genetic SNPs to specific cells within CTAPS, which could contribute to our understanding of the functional relevance of these SNPs (CTAPs are now shown not to be fixed across time but this does not negate the potential relevance of these associations). Lastly, they have developed a technique to reclassify tissues subjected to bulk RNAseq into CTAPs and apply this to the R4RA trial.

We thank this reviewer for noting the additional contributions in our revised manuscript.

Comments.

The reclassification of the R4RA trial is probably the most relevant aspect of the revised manuscript, in terms of a precision medicine message. Whilst the F CTAP may ultimately prove more informative than classifying tissue purely according to cellular dominance, at present I am not convinced that we are much further forward. The Rivellese Nature Medicine paper has already associated a fibroblast-rich (pauci-immune) pathotype with refractoriness to treatment. Furthermore, in that post hoc analysis, a B-cell poor and myeloid rich synovium,

identified by more standard deconvolution of bulk RNAseq, associated with a 77% likelihood of response to tocilizumab and only 14% response to rituximab. A B-cell rich synovium did not predict response to rituximab, consistent with the T-B CTAP also not predicting response to RTX.

Rivellese*, Surace*, et al. showed an important association between histologic or mixed cell RNAseq data and response to a specific treatment. However, in our study, by more precisely classifying RA subgroups using cell type proportions, we take an additional step toward unraveling the underlying molecular pathophysiology of RA synovitis that is not possible with bulk techniques. The CTAPs provide a formalized framework, linked to specific cell states and genes, in which to investigate these questions. The CTAP framework offers several advantages over histologic classifications and cell-rich/poor stratifications.

1. Our study has put forth putative newly defined RA disease endotypes that are defined by major cell types but characterized by the distinct arrangement of finer-grained cell states, in this case 77 cell clusters across the six major cell types. Our recategorization of the R4RA samples is notable as we now can define in significantly more detail what the underlying cell types are that confer treatment response or the lack thereof. For example, while the R4RA study found that $6/12 = 50\%$ of patients with the pauci-immune pathotype responded to rituximab, we found that only $6/22 = 27\%$ of patients in CTAP-F responded, suggesting that CTAP-F more cleanly captures treatment-resistant patients. Finer-grain post-hoc subsetting is how Rivellese, Surace, et al identified the B-cell-poor and macro/mDC-rich group with a 77% likelihood of response to tocilizumab. The CTAPs allow for prospective classification of synovial tissue in future clinical studies to allow investigators to link their clinical outcomes with fine-grained cellular and molecular characterizations.
2. The CTAPs can be easily generalized to other studies that use scRNA-seq, bulk RNA-seq, or flow cytometry, where differences in response, prognosis and other clinical features can be examined. The CTAP algorithm does not rely on the diversity of samples within a cohort, unlike the B-cell-rich/poor etc framework used by R4RA, which splits samples into two equal groups and therefore inherently assumes a uniform distribution of samples in all RA cohorts.
3. The CTAPs are associated with specific cell subsets and molecular pathways that can be investigated further in future studies. For example, in **Figures 3d-e and 4d-f**, we were able to utilize the CTAP framework to identify cell states and soluble factors characteristic of specific CTAPs, and design functional experiments to characterize the interactions between these elements in different types of RA synovium.

In short, by combining molecular and cellular insights with an RA classification algorithm that any investigator can apply to their scRNA-seq, bulk RNA-seq, or flow cytometry data, the CTAP framework can enable progress on both clinical and research fronts.

In the current reanalysis of R4RA the authors clearly demonstrate that CTAPs change with treatment, which is a new and important observation.

Curiously, however, 5/9 TM CTAPs become T-B after rituximab, which seems counter-intuitive and is not discussed. As mentioned above, the only CTAP that predicted (non)-response was CTAP F. Despite significant IL6 expression, neither CTAP TF nor M predicted response to TCZ, despite the data from the Rivellese paper that a myeloid rich synovium (regardless of B-cell infiltrate) associated with a 70% response to TCZ. I recognise that, by definition, we cannot equate M-CTAP with myeloid-rich synovium by deconvoluted bulk sequencing (or indeed F-CTAP with fibroblast-rich synovium) but, at least according to this reanalysis of R4RA, CTAPs do not appear to provide an improved theragnostic prediction.

We thank the reviewer for acknowledging the importance of our new analyses and observations. We agree that it is hard to equate certain CTAPs with a pathotype classification, given that CTAPs are derived from a more granular assessment of direct abundances of multiple cell states, while pathotypes are histologically based. We also note that the small sample size of certain CTAPs in this analysis makes it difficult to make conclusions about particular CTAPs.

The observation that several patients in CTAP-TM convert to CTAP-TB after treatment with rituximab (**Extended Data 10f**) is indeed very interesting and warrants further investigation in future studies, as does the association, or lack thereof, between IL-6 expression in tissues and response to treatments with tocilizumab or other medications targeting the IL-6 pathway. These are exactly the kind of provocative and potentially important new questions that we hoped to inspire with this discovery study.

In the current manuscript, we present treatment responses for tocilizumab and rituximab pooled together because of limited sample size in the R4RA cohort. However, clinical response data stratified by treatment reveals interesting trends that point to the opportunities presented by the CTAP framework (**New Extended Data Figure 10k**). For example, we find that 13/20 = 65% of patients in CTAP-M had a clinical response to tocilizumab compared to only 4/8 = 50% of the patients in CTAP-TM, suggesting that tocilizumab is more effective in patients with the molecular environment of CTAP-M compared to CTAP-TM. Taken together with the cell states we find to be associated with CTAP-M versus CTAP-TM (e.g. SPP1+ macrophages, mural fibroblasts, etc), this clinical response observation opens up new avenues of translational investigation, the central goal of our discovery study.

We have revised the text to reflect these points:

*We found that CTAPs were dynamic during this period, with 30 out of 45 patients changing to a different CTAP (**Figure 5d, Extended Data Figure 10e**). Patients in the tocilizumab and rituximab treatment arms exhibited similar frequencies of change in CTAP (20/29 [69%] patients treated with rituximab and 10/16 [63%] patients treated with tocilizumab) (**Extended Data Figure 10f-i**). Among patients treated with rituximab, all patients with CTAP-TB converted to a different CTAP, while some patients in CTAP-TM converted to CTAP-TB (**Extended Data Figure 10f**). Among patients that changed*

CTAPs, CTAP-F was the most common CTAP at week 16 (16/30 [53%]), consistent with the mechanisms of action of rituximab and tocilizumab targeting inflammatory cells and pathways.

New Extended Data Figure 10k. Clinical response in the R4RA trial, stratified by treatment. Bar graphs show R4RA participants in each CTAP with clinical response (teal) (50% improvement in CDAI over 16 weeks) vs non-response (red) to tocilizumab (left) or rituximab (right).

I note that the relatively rare CTAP EFM seems to include a high (?5/6) frequency of responders but presumably in a non-discriminatory manner. This, of course, could be 'noise' due to small numbers, and is not statistically significant, but feels contrary to the statement: "other CTAPs, such as CTAP EFM and CTAP-F, have no obvious targets of currently available treatments and warrant further focused study". Almost two thirds of M CTAP also respond to either RTX or TCZ.

We thank the reviewer, and we have revised our text. To avoid confusion, we have removed the quoted text from the Discussion.

Regarding CTAP-M, as discussed above (**Extended Data Figure 10k**), CTAP-M displays a trend toward stronger response to tocilizumab than to rituximab, similar to the observation regarding myeloid-rich samples reported by Rivellese, Surace, et al, although numbers are too small to be certain. We have chosen not to remark on this finding in the manuscript given the limited sample size in this subgroup analysis.

CTAPs TB and TM, both highly cellular CTAPs, segregate more or less equally between responders and non-responders (not quite in keeping with the statement: "abrogating T-B cell communication with B cell-depleting antibodies (e.g. rituximab) or blocking costimulation (e.g. abatacept) in CTAP-TB may break the pathogenic mechanisms that drive inflammation in these patients").

We thank the reviewer for this comment. To clarify, the treatment analysis shown in **Figure 5e** and **Extended Data Figure 10j** pooled individuals in the two treatment groups (tocilizumab and rituximab) together. To avoid confusion, we have removed the statement quoted by the Reviewer from the Discussion.

In summary, this work remains outstanding and world-leading in its depth and breadth. The additional information provided further increases our understanding of this novel synovium classification system. Continued evaluation is likely to further our understanding of RA pathogenesis.

We thank the reviewer for their insightful comments and generating important discussion. We agree that continued evaluation is needed to further our understanding of RA pathogenesis, and hope that the CTAP framework is instrumental in motivating more hypotheses and experiments that contribute to this understanding.

Referee #3 (Remarks to the Author):

This resubmission is substantially revised from the prior submission. The authors have been moderately responsive to the prior review, with a comprehensive re-writing of the discussion and results from 2-3 of 6 of their major figures. The anchor for the overall manuscript continues to be their 6 high level CTAP subtypes, namely TB, TF, TM, M, F, EFM. Four major additions/new work products are provided:

1. The 6 CTAP subtypes are dynamic with treatment (Surace et al., Nature Medicine 2022). Mahalanobis classifiers were used to place patients with serial bulk RNASeq (45 are serial, pre- and post-16 week treatment with either rituximab or tocilizumab, R4RA trial) into CTAP categories.
2. Cell-cell interactions in vitro.
3. SEB: superantigens of TB cross-talk
4. Monocyte subsets with supernatant
5. Genetics added (Fig 6). The 71 high confidence rheumatoid arthritis genes were projected onto the single cell data, which is very insightful for the field.
6. Microscopy. IF from 36 linked joints; SFig 3.

The authors have streamlined items that were off topic/off the mainstream of the currently most effective RA medications (e.g. deleting data directed toward IL1 blockade), in order to focus on the most effective current agents. They have added, in response to the prior review, many pertinent negatives in supplementary materials.

We thank the Reviewer for appreciating the new analyses and discussion in the revised manuscript.

Major comments

1. The 6 CTAP subtypes being the anchor for this manuscript. Fig 1 from the original submission was based on moderate to highly active RA, of 3 major groups; a) no prior treatment, b) methotrexate treatment, and c) anti-TNF non-responders. They provide additional support for the 6 major CTAPs, including secreted factor analyses and immunofluorescence from the same joints. However, this last, important new data on 36 linked joint samples (new Fig 1J, SFig 3) supports a more dichotomous distinction between CTAP-TB vs. CTAP-F & CTAP-EFM; it generally confirms that multiple samples from the same joint are of the similar general subtype. I continue to have concerns regarding the overall meaning and impact of their 6 subtype classification for the following reasons:

- Their new data from the Surace et al clinical trial (Fig 6) pre-treatment demonstrates markedly lower pre-treatment frequencies for two of the six subtypes, CTAP-TF and CTAP-EFM. Surace et al. includes only anti-TNF non-responders, so represents a substantially different cohort compared to their single cell cohort.

We agree that the pre-treatment frequencies are dissimilar between Ravellese, Surace et al and our AMP discovery dataset (**New Extended Data Figure 10b**). They are indeed different cohorts. The patients in our study were recruited to reflect a diversity of patient phenotypes. In contrast, R4RA (Ravellese, Surace et al) is a clinical trial with specific recruitment criteria, which by design restricts patient diversity.

We have revised the text to reflect these points:

We note that our cohort is more diverse than the R4RA cohort, which included only patients with an inadequate response to TNF inhibitor therapy. This is reflected in the distribution of CTAPs within these datasets (Extended Data Figure 10b).

New Extended Data Figure 10b. Distribution of CTAPs among patients in the AMP and R4RA study cohorts. The AMP cohort represents a diverse clinical cohort of RA patients whereas recruitment to the R4RA clinical trial was limited to patients with an inadequate response to TNF inhibitors.

- The high level lumping of B cells and plasma cells in the present CTAP definitions. Given the key role for B cells in RA pathogenesis overall, this seems misguided. The exclusion of one of their 6 major cell cluster types (NK cells--were there CITE-Seq markers for NK cells) excludes important secreted mediators (e.g. IFNG). Their more fine-scale transcript clustering (24 T cell clusters, 9 B/plasma cell clusters; 15 myeloid; 5 DC markers) provides much more refined gene selection than can be achieved with their protein (limited measures)-RNA multimodal CTAP classifier.

The reviewer has encouraged us to consider B cells and plasma cells separately in our CTAP definitions. We have evaluated the effect of separating B cells and plasma cells into two populations. We find almost no difference in defining CTAPs when B cells are separated into plasma cells and non-plasma B cells compared to pooling them together, as we did in the original CTAP analysis (**New Supplementary Figure 3e-g**).

New Supplementary Figure 3e-g. Separating plasma cells and non-plasma B cells has minimal effects on CTAP classifications of synovial tissue samples. *e*, PCA plot showing the distribution of samples in the AMP cohort by abundance of the frequencies of plasma cells (P), non-plasma B cells (B), T cells (T), NK cells (NK), endothelial cells (E), fibroblasts (F), and myeloid cells (M). *f*, Confusion matrix demonstrates high concordance in CTAP classifications when B cells are separated into plasma cells versus non-plasma B cells (7-population CTAPs) versus pooled together (original 6-population CTAPs). *g*, Heatmap depicting average proportions of each of the seven major cell types among samples in each of the original CTAPs.

We also note that NK cells were included in the CTAP determination; we clustered on frequencies of all cells, including NK cell frequencies (**Figure 1e-g**). However, NK cells are not included in the names of the CTAPs since they are always a minority of cell states (see **Figure 1f**), per our naming convention. We note that evaluation of synovial NK cells is prominently featured in our manuscript (**Figures 3f-g, 5a, Extended Data Figures 5a-f, 9a-c, and Supplementary Figures 2h, 8b, 9, 10, 12a**), as is IFN γ (**Extended Data Figures 1h, 3d, 5d, 9a-c; Supplementary Figure 9**)

- That the present 6 CTAPs have two (TM, TF, TB), one (M, F) and 3 (EFM) major cell subtypes. I am concerned that this reflects an arbitrary reflection of their reliance on protein-RNA multimodal classifiers at the beginning of their study. Given the present addition of the very important Surace data to close the manuscript, I would favor a more transcript-focused classifier (esp. for Fig 6), relying on CITE-Seq for selected validation of key refined subclustering.

We thank the Reviewer for this comment. To address these concerns, we have performed pseudo-bulk analysis of the single-cell RNAseq data from our CITE-seq dataset (i.e. RNA only, no protein markers). Specifically, we summed and normalized the counts across all genes and then performed principal component and confusion matrix analyses. We find that the classification of samples based on pseudo-bulk largely reproduces the original CTAP groups (**New Supplementary Figure 3a-b**).

New Supplementary Figure 3a-b. Classification of samples based on pseudo-bulk RNAseq analysis largely recreates the original CTAPs. a, PCA plot showing the distribution of samples in the AMP cohort based on pseudo-bulk transcriptomic analysis. b, Confusion matrix demonstrates high concordance of transcriptomic pseudo-bulk classifications with the original CTAP designations.

2. The additional studies (Fig 3) focused on the CTAP-TB nicely justifies the TB subset.

Thank you for this positive comment.

3. The additional monocyte focused studies details nicely their enormous heterogeneity, as does the serial analyses of the Surace data (alluvial plots). Many patients originally CTAP-M spread to multiple subsequent subtypes after 16 weeks of therapy.

We thank the reviewer for this observation linking together the functional experiments and the observations from the R4RA clinical trial.

4. CCP antibody levels: are they available in the Surace cohort? In the Darrah et al. review, 28937414, the source of citrullination is from innate immune cells, so not necessarily surprising that CTAP-TB have lower CCP antibodies.

In our cohort, we found that CCP negative status was associated with CTAP-M (**Extended Data 8c**), while CTAP-TM exhibited the lowest CCP titers (**Figure 5b**). In the R4RA cohort, we only have seropositive vs seronegative status data, and unfortunately do not have titer values. Among R4RA patients, a larger proportion of CCP-negative patients were CTAP-M. Patients in CTAP-TB were more frequent among CCP-positive patients in the AMP cohort and approximately equally frequent among CCP-positive and -negative patients in the R4RA cohort. Both studies are underpowered for this analysis since both studies include such a small number of seronegative patients. Further studies are necessary to determine the association of CCP antibody status with CTAPs and the mechanistic underpinnings of these associations.

Review Response Figure 1. CTAPs of patients in the AMP cohort (left) and R4RA cohort (right) stratified by CCP antibody status.

5. The imperfect mapping (only 6 of 7 patients) of CTAP classifiers with bulk in Figure 6 further underscores the questionable value and overly heavy reliance on this classification. A gene-based classifier selected from their 77 fine-scale clusters might well be more insightful and reproducible.

From our point of view, the assignment of 6/7 in a 6-way classification demonstrates relatively high accuracy in a difficult task. Indeed, the one misclassification was between CTAP-F and CTAP-EFM, which are two of the more transcriptionally similar CTAPs. Generally, deconvoluting bulk RNA-seq profiles into cell type or cell state frequencies is a difficult task, and we felt that these results validate the efficacy of our approach.

The Reviewer also asks about a gene-based classifier using the 77 fine-grain clusters. We analyzed the 70 RA samples using the abundances of cells in the 77 clusters, and the pattern

essentially reproduced the original CTAPs, as shown in **New Supplemental Figure 3c-d**, shown below.

New Supplemental Figure 3c-d. Clustering samples based on frequencies of 77 fine-grain clusters replicates the CTAPs. **c**, PCA of the abundance analysis using the fine-grained cell states for the 70 RA samples, with samples colored by original CTAP classification. **d**, Heatmap visualizing concordance of the fine-grained cell states-driven classification (columns) and original CTAP classification (rows) for each sample,

Minor

- Differences between their original cohort and R4RA trial should be briefly mentioned in the discussion. What implications might this have for differences in cell clusters?

We have added a sentence to the Discussion to reiterate the differences in clinical diversity among our cohort and the R4RA clinical trial:

*We applied the CTAP classification system to a clinical trial in RA.⁹⁶ **Even within the more limited clinical diversity of the R4RA cohort, we found that CTAPs can change over time with treatment and that CTAP-F was associated with poor clinical response.***

- Expression of high confidence associated genes: consider adding bulk data for non-detected genes from single cell

We agree that bulk RNA-seq offers greater depth of sequencing than currently available single-cell technologies. Bulk RNA-seq was the primary technology in the first phase of the AMP project, published as Zhang et al, Nature Immunology 2019, PMID 31061532. This study includes bulk RNAseq from 36 patients with RA and 15 with osteoarthritis as well as single-cell RNAseq and mass cytometry from subsets of these patients. These data are available on an easy-to-use website, openly accessible at <https://immunogenomics.io/ampra/>, to allow readers to query genes of interest in bulk RNA-seq data.

- HLA expression & B cells: Can a reference be added to the discussion with respect to the cellular contributions of MHC class II alleles

HLA class II expression by B cells is indeed an interesting topic. We have added a recent review to the discussion to refer readers to a more in-depth description of the current science. Here is our updated wording in the text:

...HLA-DRB1 expression was correlated with CTAP-associated cell states in several cell types, including B cells, which may play a dual role in RA as antigen-presenting cells in addition to antibody-producing cells (Figure 5f) (New reference Ghosh et al, Current Opinion in Immunology 2021; PMID 34242927).

Reviewer Reports on the Second Revision:

Referees' comments:

Referee #3 (Remarks to the Author):

The authors have responded completely to major points raised in the prior reviews.

1. Clarification wrt roles of B vs. plasma cells, NK cells
2. Confusion matrix analyses confirming classification validity (now Supp Fig 3) between CTAPs and the 77 fine-scale clusters
3. Additional data (review letter only) on anti-citrulline antibodies. It appears that, for future classification protocols in clinical trials, this information will provide orthogonal information to gene expression data.
4. Agree with authors that the present data and classification distinctions does provide an important advance over Rivellesse, Surace et al data.
5. The treatment ordering and response heterogeneity AMP vs. R2RA datasets continue to plague the field, highlighting the enormous complexity of these traits.